# Dimensionality reduction: theoretical perspective on practical measures

**Yair Bartal**[*]
Department of Computer Science
Hebrew University of Jerusalem
Jerusalem, Israel
yair@cs.huji.ac.il

**Nova Fandina**
Department of Computer Science
Hebrew University of Jerusalem
Jerusalem, Israel
fandina@cs.huji.ac.il

**Ofer Neiman**
Department of Computer Science
Ben Gurion University of the Negev
Beer-Sheva, Israel
neimano@cs.bgu.ac.il

## Abstract

Dimensionality reduction plays a central role in real world applications for Machine Learning, among many fields. In particular, *metric dimensionality reduction*, where data from a general metric is mapped into low dimensional space, is often used as a first step before applying machine learning algorithms. In almost all these applications the quality of the embedding is measured by various *average case* criteria. Metric dimensionality reduction has also been studied in Math and TCS, within the extremely fruitful and influential field of *metric embedding*. Yet, the vast majority of theoretical research has been devoted to analyzing the *worst case* behavior of embeddings, and therefore has little relevance to practical settings. The goal of this paper is to bridge the gap between theory and practice view-points of *metric dimensionality reduction*, laying the foundation for a theoretical study of more practically oriented analysis.

This paper can be viewed as providing a comprehensive theoretical framework for analyzing different distortion measurement criteria, with the lens of practical applicability, and in particular for Machine Learning. The need for this line of research was recently raised by Chennuru Vankadara and von Luxburg in (13)[NeurIPS' 18], who emphasized the importance of pursuing it from both theoretical and practical perspectives.

We consider some important and vastly used average case criteria, some of which originated within the well-known *Multi-Dimensional Scaling* framework. While often studied in practice, no theoretical studies have thus far attempted at providing rigorous analysis of these criteria. In this paper we provide the first analysis of these, as well as the new distortion measure developed in (13) designed to posses Machine Learning desired properties. Moreover, we show that all measures considered can be adapted to posses similar qualities. The main consequences of our work are nearly tight bounds on the absolute values of all distortion criteria, as well as *first* approximation algorithms with provable guarantees.

All our theoretical results are backed by empirical experiments.

---

[*]Author names are ordered alphabetically.

# 1 Introduction

*Metric Embedding* plays an important role in a vast range of application areas such as machine learning, computer vision, computational biology, networking, statistics, data mining, neuroscience and mathematical psychology, to name a few. Perhaps the most significant is the application of *metric dimensionality reduction* for large data sets, where the data is represented by points in a metric space. It is desirable to efficiently embed the data into low dimensional space which would allow compact representation, efficient use of resources, efficient access and interpretation, and enable operations to be carried out significantly faster.

In machine learning, this task is often used as a preliminary step before applying various machine learning algorithms, and sometimes refereed to as unsupervised metric dimensionality reduction. Some studies of dimensionality reduction within ML include (10; 11; 13; 42; 49). Moreover, there are numerous practical studies of metric embedding and dimensionality reduction appearing in a plethora of papers ranging in a wide scope of research areas including work on Internet coordinate systems, feature extraction, similarity search, visual recognition, and computational biology applications; the papers (25; 41; 24; 5; 21; 17; 45; 48; 51; 52; 44; 32; 12; 11; 47) are just a small sample.

In nearly all practical applications of metric embedding and dimensionality reduction methods, the fundamental criterion for measuring the quality of the embedding is its *average* performance over all pairs, where the measure of quality per pair is often the distortion, the square distortion and similar related notions. Such experimental results often indicate that the quality of metric embeddings and dimensionality reduction techniques behave very well in practice.

In contrast, the classic theory of metric embedding has mostly failed to address this phenomenon. Developed over the past few decades by both mathematicians and theoretical computer scientists (see (26; 34; 27) for surveys), it has been extremely fruitful in analyzing the *worst case* distortion of embeddings. However, worst case analysis results often exhibit extremely high lower bounds. Indeed, in most cases, the worst case bounds are growing, in terms of both distortion and dimension, as a function of the size of the space. Such bounds are often irrelevant in practical terms.

These concerns were recently raised in the context of Machine Learning in (13) (NeurIPS'18), stressing the desire for embeddings into *constant* dimension with *constant* distortion. The authors of (13) state the necessity for a systematic study of different average distortion measures. Their main motivation is to examine the relevance of these measures for machine learning applications. Here, the first step is made to tackle this challenge.

The goal of this paper is to bridge between theory and practice outlook on metric embedding and dimensionality reduction. In particular, providing the first comprehensive rigorous analysis of the most basic practically oriented *average* case quality measurement criteria, using methods and techniques developed within the classic theory of metric embedding, thereby providing new insights for both theory and practice.

We focus on some of the most basic and commonly used average distortion measurement criteria:

**Moments analysis: moments of distortion and Relative Error.** The most basic average case performance criterion is the *average distortion*. More generally, one could study all *q-moments* of the distortion for every $1 \leq q \leq \infty$. This notion was first studied in (1). For a *non-contractive* embedding $f$, whose distortion for a pair of points $u, v$ is denoted $dist_f(u, v)$:

**Definition 1** ($\ell_q$-distortion). *Let $(X, d_X)$ and $(Y, d_Y)$ be any metric spaces, and $f : X \to Y$ be an embedding. For any distribution $\Pi$ over $\binom{X}{2}$ and $q \geq 1$, the $\ell_q$-distortion of $f$ with respect to $\Pi$ is defined by: $\ell_q\text{-}dist^{(\Pi)}(f) = \left(E_\Pi\left[(dist_f(u, v))^q\right]\right)^{\frac{1}{q}}$, $\ell_\infty\text{-}dist^{(\Pi)}(f) = \sup_{\Pi(u,v)\neq 0}\{dist_f(u, v)\}$.*

The most natural case is where $\Pi$ is the uniform distribution (and will be omitted from the notation). In order for this definition to extend to handle embeddings in their full generality and address important applications such as dimensionality reduction, it turns out that one should remove the assumption that the embedding is non-contractive.

We therefore naturally extend the above definition to deal with *arbitrary* embeddings by letting $dist_f(u, v) = \max\{expans_f(u, v), contr_f(u, v)\}$, where $expans_f(u, v) = \frac{d_Y(f(u), f(v))}{d_X(u, v)}$,

$contr_f(u,v) = \frac{d_X(u,v)}{d_Y(f(u),f(v))}$. In the full version (in supplementary materials) we provide justification of the necessity of this definition. Observe that this definition is not scale invariant[2].

In many practical cases, where we may expect a near isometry for most pairs, the moments of distortion may not be sensitive enough and more delicate measures of quality, which examine directly the pairwise additive error, may be desired. The *relative error measure* (REM), commonly used in network applications (45; 44; 15) is the most natural choice. It turns out that this measure can be viewed as the moment of distortion about 1. This gives rise to the following generalization of Definition 1:

**Definition 2** ($\ell_q$**-distortion about** $c$**, REM**). *For $c \geq 0$, the $\ell_q$-distortion of $f$ about $c$ is given by:*

$$\ell_q\text{-}dist_{(c)}^{(\Pi)}(f) = (E_\Pi\left[|dist_f(u,v) - c|^q\right])^{\frac{1}{q}}, \quad REM_q^{(\Pi)}(f) = \ell_q\text{-}dist_{(1)}^{(\Pi)}(f).$$

**Additive distortion measures: Stress and Energy.** *Multi Dimensional Scaling* (see (16; 8)) is a well-established methodology aiming at embedding a metric representing the relations between objects into (usually Euclidean) low-dimensional space, to allow feature extraction often used for indexing, clustering, nearest neighbor searching and visualization in many application areas, including machine learning (42). Several average additive error criteria for the embedding's quality have been suggested in the context of MDS over the years. Perhaps the most popular is the *stress* measure going back to (30). For $d_{uv} = d_X(u,v)$ and $\hat{d}_{uv} = d_Y(f(u),f(v))$, for normalized nonnegative weights $\Pi(u,v)$ (or distribution) we define the following natural generalizations, which include the classic Kruskal stress $Stress^*_2(f)$ and normalized stress $Stress_2(f)$ measures, as well as other common variants in the literature (e.g. (23; 46; 20; 9; 50; 11)): $Stress_q^{(\Pi)}(f) = \left(\frac{E_\Pi\left[|\hat{d}_{uv} - d_{uv}|^q\right]}{E_\Pi\left[(d_{uv})^q\right]}\right)^{1/q}$, and

$Stress^*{}_q^{(\Pi)}(f) = \left(\frac{E_\Pi\left[|\hat{d}_{uv} - d_{uv}|^q\right]}{E_\Pi\left[(\hat{d}_{uv})^q\right]}\right)^{1/q}$. Another popular and widely used additive error measure is *energy* and its special case, *Sammon* cost (see e.g. (43; 7; 14; 36; 37; 12)). We define the following generalizations, which include some common variants (e.g. (41; 45; 44; 33)): $Energy_q^{(\Pi)}(f) = \left(E_\Pi\left[\left(\frac{|\hat{d}_{uv} - d_{uv}|}{d_{uv}}\right)^q\right]\right)^{1/q}$, and $REM_q^{(\Pi)}(f) = \left(E_\Pi\left[\left(\frac{|\hat{d}_{uv} - d_{uv}|^q}{\min\{\hat{d}_{uv}, d_{uv}\}}\right)^q\right]\right)^{1/q}$.

It immediately follows from the definitions that: $Energy_q^{(\Pi)}(f) \leq REM_q^{(\Pi)}(f) \leq \ell_q\text{-}dist^{(\Pi)}(f)$. Also it's not hard to observe that $Stress_q^{(\Pi)}$ and $Energy_q^{(\Pi')}(f)$ are equivalent via a simple transformation of weights.

**ML motivated measure: $\sigma$-Distortion.** Recently published paper (13) studies various existing and commonly used quality criteria in terms of their relevance in machine learning applications. Particularly, the authors suggest a new measure, $\sigma$- *distortion*, which is claimed to possess all the necessary properties for machine learning applications. We consider a generalized version of $\sigma$-distortion[3]. Let $\ell_r\text{-}expans(f) = \left(\binom{n}{2}^{-1}\sum_{u \neq v}(expans_f(u,v))^r\right)^{1/r}$. For a distribution $\Pi$ over $\binom{X}{2}$, let $\Phi_{\sigma,q,r}^{(\Pi)}(f) = \left(E_\Pi\left[\left|\frac{expans_f(u,v)}{\ell_r\text{-}expans(f)} - 1\right|^q\right]\right)^{1/q}$ (for $q = 2$, $r = 1$ this is the square root of the measure defined by (13)). We show that the tools we develop in this paper can be applied to $\sigma$-distortion to obtain theoretical bounds on its value.

We further show (Section 7), generalizing (13), that all other average distortion measures considered here can be easily adapted to satisfy similar ML motivated properties.

A basic contribution of our paper is showing deeper tight relations between these different objective functions, and further developing properties and tools for analyzing embeddings for these measures. While these measures have been extensively studied from a practical point of view, and many heuristics are known in the literature, almost nothing is known in terms of rigorous analysis and absolute bounds. Moreover, many real-world misconceptions exist about what dimension may be necessary for good embeddings. In this paper we present the *first theoretical analysis* of all these

measures providing absolute bounds that shed light on these questions. We exhibit approximation algorithms for optimizing these measures, and further applications.

In this paper we focus only on analyzing objective measures that attempt to preserve metric structure. As a result, some popular objective measures used in applied settings are beyond the scope of this paper, this includes the widely used t-SNE heuristic (which aims at reflecting the cluster structure of the data, and generally does not preserve metric structure), and various heuristics with local structure objectives. When validating our theoretical findings experimentally (Section 6), we chose to compare our results with the most common in practice heuristics PCA/classical-MDS and Isomap amongst the various methods that appear in the literature.

**Moment analysis of dimensionality reduction.** The main theoretical question our paper studies is:

**Problem 1 ((k, q)-Dimension Reduction).** *Given a dimension bound $k$ and $1 \leq q \leq \infty$, what is the least $\alpha(k, q)$ such that every finite subset of Euclidean space embeds into $k$ dimensions with $\mathrm{Measure}_q \leq \alpha(k, q)$ ?*

This question can be phrased for each $\mathrm{Measure}_q$ of practical importance. A stronger demand would be to require a single embedding to *simultaneously* achieve best possible bounds for all values of $q$.

We answer Problem 1 by providing (almost tight for most of the values of $k$ and $q$) upper and lower bounds on $\alpha(k, q)$. In particular we prove that the Johnson-Lindenstrauss (JL) dimensionality reduction achieves bounds in terms of $q$ and $k$ that dramatically outperform a widely used in practice PCA algorithm. Moreover, our experiments show that the same holds for the Isomap and classical MDS methods.

The bounds we obtain provide several interesting conclusions regarding the expected behavior of dimensionality reduction methods. As expected, the bound for the JL method is improving as $k$ grows, confirming the intuition expressed in (13). Yet, countering their intuition, the bound *does not* increase as a function of the original dimension $d$. A phase transition, exhibited in our bounds, provides guidance on how to choose the target dimension $k$.

Another consequence arises by combining our result with the embedding of (1), by composing it with the JL: we obtain an embedding of general spaces into a *constant* dimensional Euclidean space with *constant* distortion, for all discussed measures (presented in the full version). Here, the dimension is constant even if the original space is not doubling, improving on the result obtained in (13).

**Approximation algorithms.** The bounds achieved for the Euclidean $(k, q)$-dimension reduction are then applied to provide the *first* approximation algorithms for embedding general metric spaces into low dimensional Euclidean space, for all the various distortion criteria. This is based on composing convex programming with the JL-transform. It should be stressed that such a composition may not necessarily work in general, however, we are able to show that this yields efficient approximation algorithms for all the criteria considered in this paper.

The results on the JL transform yield bounds on distance oracles. In the full version, we provide additional applications, including *metric hyper sketching*, a generalization of standard sketching.

**Empirical Experiments.** We validate our theoretical findings experimentally on various randomly generated Euclidean and non-Euclidean metric spaces, in Section 6. In particular, as predicted by our lower bounds, the phase transition is clearly seen in the JL, PCA and Isomap embeddings for all the measurement criteria. Moreover, in our simulations the JL based approximation algorithm (as well as the JL itself, when applied on Euclidean metrics) has shown dramatically better performance than the PCA and Isomap heuristics for all distortion measures, indicating that the JL-based approximation algorithm is a preferable choice when the preservation of metric properties is desirable.

**Related work.** For Euclidean embedding, it was shown in (35) that using SDP one can obtain arbitrarily good approximation of the distortion. However, such a result is impossible when restricting the target dimension to $k$, as in (39) it was shown that unless P=NP, the approximation factor must be $n^{\Omega(1/k)}$. Of all the measures studied in this paper, only $Stress_q$ was previously studied. In (10), it was shown that computing an embedding into $\mathbb{R}^1$ with optimal $Stress_q$ is NP-hard, for any given $q$. The only approximation algorithms known for this problem are the following: a 2-approximation to $Stress_\infty$ for embedding into $\mathbb{R}^1$ (22); an $O(\log^{1/q} n)$-approximation to $Stress_q$ for embedding into $\mathbb{R}^1$ (19); an $O(1)$-approximation to $Stress_\infty$ for embedding into $\ell_1^2$ (6).

All proofs are omitted from this version and appear in the full version (in supplemental material).

## 2 On the limitations of classical MDS

Practitioners have developed various heuristics to cope with dimensionality reduction (see (49) for a comprehensive overview). Most of the suggested methods are based on iterative improvement of various objectives. All these strategies do not provide theoretical guarantees on convergence to the global minimum and most of them even do not necessarily converge. Furthermore, classical MDS or PCA, one of the widely used heuristics, is usually referred to as the method that computes the optimal solution for minimizing $Stress_2$. We show that this is in fact false: PCA can produce an embedding with $Stress_q$ value being far from optimum, even for the space that can be efficiently embedded into a line.[4] Consider the following subset of $\mathbb{R}^d$. For any $\alpha < 1/2$, for all $i \in [1, d]$, for any $q \geq 1$, let $s_i = 1/(\alpha^i)^q$. Let $X_i \subset \ell_2^d$ be the (multi) set of size $2s_i$ that contains $s_i$ copies of the vector $\alpha^i \cdot e_i$, denoted by $X_i^+$, and $s_i$ copies of the antipodal vector $-\alpha_i \cdot e_i$, denoted by $X_i^-$, where $e_i$ is the standard basis vector of $\mathbb{R}^d$. Define $X$ as the union of all $X_i$. In the full version of the paper, we show that $X$ can be embedded into a line with $Stress_q/Energy_q(f) = O(\alpha/d^{1/q})$, for any $q \geq 1$. Yet, for the PCA algorithm applied on $X$, into $k \leq \beta \cdot d$ dimensions ($\beta < 1$), it holds that $Stress_q/Energy_q(F) = \Omega(1)$, and $\ell_q\text{-}dist/REM_q(F) = \infty$.

Moreover, our empirical experiments show that the PCA and Isomap methods have significantly worse performance than the JL on a variety of randomly generated families of metric spaces.

## 3 Euclidean dimension reduction: moment analysis of the JL transform

From a theoretical perspective, dimensionality reduction is known to be possible in Euclidean space via the Johnson-Lindenstrauss Lemma (29), a cornerstone of Banach space analysis and metric embedding theory, playing a central role in a plethora of applications. The lemma states that every $n$ point subset of Euclidean space can be embedded in $O(\epsilon^{-2} \log n)$ dimensions with *worst case* $1 + \epsilon$ distortion. The dimension bound is shown to be tight in (31) (improving upon (4)). When applied in a fixed dimension $k$, the worst case distortion becomes as bad as $O(n^{2/k}\sqrt{\log n})$. Moreover, in (40) a lower bound of $n^{\Omega(1/k)}$ on the worst case distortion of *any* embedding in $k$ dimensions was proven. However, as explained above, in many practical instances it is desirable to replace the demand for worst case with average case guarantees. It should be noted yet that the *JL transform* does have good properties, even when applied in $k$ dimensions. The JL lemma in fact implies that in dimension $k$ for every pair there is some constant probability ($\approx \exp(-\epsilon^2 k)$) that a $1 + \epsilon$ distortion is achieved. While in itself an appealing property, it should be stressed that standard tail bounds arguments *cannot* imply that the average (or higher moments) distortion is bounded. Indeed, we show that for certain specialized implementations of the JL embedding, such as those of (2) (e.g., using Rademacher entries matrix), (3) (fast JL), and (18) (sparse JL), the $\ell_q\text{-}dist$ and $REM_q$ are *unbounded*.

**Observation 1.** *Let $k \geq 1$, and $d > k$. Let $E_d = \{e_i\}_{1 \leq i \leq d} \subseteq \ell_2^d$ be the set of standard basis vectors. Assume that a linear map $f : \ell_2^d \to \ell_2^k$ is given by a transformation matrix $P_{k \times d}$, such that for all $i, j$, $P[i, j] \in U$ for some finite set $U \subset \mathbb{R}$. If $|U| < d^{\frac{1}{k}}$ then for the set $E_d$, for all $q \geq 1$, $\ell_q\text{-}dist(f), REM_q(f) = \infty$.*

The proof follows by volume argument: for matrix $P$ the set $f(E_d) = \{Pe_i\}_{1 \leq i \leq d}$ is exactly the set of columns of $P$. Since the entries of $P$ belong to $U$, there can be at most $|U|^k < d$ different columns in the set $f(E_d)$. Therefore, there is at least one pair of vectors in $E_d$ that will be mapped into the same image by $f$. This implies the observation as $\ell_q$-distortion and $REM_q$ measures depend on the inverse of the embedded distance.

Yet, focusing on the Gaussian entries implementation by (28) we show that it behaves dramatically better. Let $X \subset \ell_2^d$ be an $n$-point set, and $k \geq 1$ be an integer. The *JL transform of dimension $k$*, $f : X \to \ell_2^k$ is defined by generating a random matrix $T$ of size $k \times d$, with i.i.d. standard normal entries, and setting $f(x) = \frac{1}{\sqrt{k}}Tx$, for all $x \in X$.

**Theorem 1.** *Let $X \subset \ell_2^d$ be an $n$-point set, and let $k \geq 1$. Given any distribution $\Pi$ over $\binom{X}{2}$, the JL transform $f : X \to \ell_2^k$ is s.t. with probability at least $1/2$, $\ell_q\text{-}dist^{(\Pi)}(f)$ is bounded by:*

| $1 \leq q < \sqrt{k}$ | $\sqrt{k} \leq q \leq \frac{k}{4}$ | $\frac{k}{4} \lesssim q \lesssim k$ | $q = k$ | $k \lesssim q \leq \infty$ |
|---|---|---|---|---|
| $1 + O\left(\frac{1}{\sqrt{k}}\right)$ | $1 + O\left(\frac{q}{k-q}\right)$ | $\left(\frac{k}{k-q}\right)^{O(1/q)}$ | $(\log n)^{O(1/k)}$ | $n^{O\left(\frac{1}{k} - \frac{1}{q}\right)}$ |

*The bounds are asymptotically tight for most values of $k$ and $q$ when the embedding is required to maintain all bounds simultaneously. For fixed $q$ tightness holds for most values of $q \geq \sqrt{k}$.*

Note that for large $q$ our theorem shows that a phase transition emerges around $q = k$. The necessity of this phenomenon is implied by nearly tight lower bounds given in Section 4.1.

**Additive distortion and $\sigma$-distortion measures analysis.** The following theorem provides tight upper bounds for all the additive distortion measures and for $\sigma$-distortion, for $1 \leq q \leq k - 1$. This follows from analyzing the $REM$ (via similar approach to the raw moments analysis):

**Theorem 2.** *Given a finite set $X \subset \ell_2^d$ and an integer $k \geq 2$, let $f : X \to \ell_2^k$ be the JL transform of dimension $k$. For any distribution $\Pi$ over $\binom{X}{2}$, with constant probability, for all $1 \leq r \leq q \leq k - 1$:*
$$REM_q^{(\Pi)}(f), Energy_q^{(\Pi)}(f), \Phi_{\sigma,q,r}^{(\Pi)}(f), Stress_q^{(\Pi)}(f), Stress^{*(\Pi)}_q(f) = O\left(\sqrt{q/k}\right).$$

The more challenging part of the analysis is figuring out how good are the JL performance bounds. Therefore our main goal is the task of establishing lower bounds for Problem 1.

# 4  Partially tight lower bounds: $q < k$

In the full version we show that JL is essentially optimal when *simultaneous* guarantees are required. If that requirement is removed, it is still the case for most of the ranges of $q$. Providing lower bounds for each range requires a different technique. One of the most interesting cases, is the proof of the lower bound of $1 + \Omega(q/(k - q))$ for the range $1 \leq q \leq k - 1$. For $q \leq \sqrt{k}$, this turns out to be a consequence of the tightness for the additive distortion measures and $\sigma$-distortion, shown to be tight for $q \geq 2$. The proof is based on a delicate application of the technique of (4). We show that the analysis of the JL transform for the additive measures and $\sigma$-distortion, provides tight bounds for all values of $2 \leq q \leq k$. Due to tight relations between the additive measures, the lower bounds for all measures follow from Energy measure. Let $E_n$ denote an $n$-point equilateral metric space.

**Claim 3.** *For all $k \geq 2$, $k \geq q \geq 2$, and $n \geq 4 \left(9 \cdot \frac{k}{q}\right)^{q/2}$, for any embedding $f : E_n \to \ell_2^k$ it holds that $Energy_q(f) = \Omega(\sqrt{\frac{q}{k}})$.*

**Claim 4.** *For all $k \geq 1$, $1 \leq q < 2$, and $n \geq 18k$, for all $f : E_n \to \ell_2^k$, $Energy_q(f) = \Omega\left(\frac{1}{k^{1/q}}\right)$.*

A more involved argument shows that Claim 3 implies

**Corollary 1.** *For any $k \geq 1$ and any $n \geq 18k$, for any embedding $f : E_n \to \ell_2^k$ it holds that $\ell_q\text{-}dist(f) = 1 + \Omega\left(\frac{q}{k}\right)$, for all $1 \leq q \leq \sqrt{k}$.*

Based on (31), we also prove

**Theorem 5.** *For all $k \geq 16$, for all $N$ large enough, there is a metric space $Z \subseteq \ell_2$ on $N$ points, such that for any $F : Z \to \ell_2^k$ it holds that $\ell_q\text{-}dist(F) \geq 1 + \Omega\left(\frac{q}{k-q}\right)$, for all $q = \Omega\left(\sqrt{k \log k}\right)$.*

## 4.1  Phase transition: moment analysis lower bounds for $q \geq k$

An important consequence of our analysis is that the $q$-moments of the distortion (including $REM_q$), exhibit an impressive *phase transition* phenomenon occurring around $q = k$. This follows from lower bounds for $q \geq k$. The case $q = k$ (and $\approx k$) is of special interest where we obtain a tight bound:

**Theorem 6.** *Any embedding $f : E_n \to \ell_2^k$ has $\ell_k\text{-}dist(f) = \Omega((\sqrt{\log n})^{1/k}/k^{1/4})$, for any $k \geq 1$.*

Hence, for any $q$, the theorem tells that only $k \geq 1.01q$ may be suitable for dimensionality reduction. This new consequence may serve an important guide for practical considerations, that seems to be missing prior to our work. We also prove the following claim for large values of $q$:

**Claim 7.** *For any embedding $f : E_n \to \ell_2^k$, for all $k \geq 1$, for all $q > k$, $\ell_q\text{-}dist(f) = \Omega(\max\{n^{\left(\frac{1}{2\lceil k/2 \rceil} - \frac{2}{q}\right)}, n^{\frac{1}{2k} - \frac{1}{2q}}\})$.*

# 5 Approximate optimal embedding of general metrics

Perhaps the most basic goal in dimensionality reduction theory and essentially, the main problem of MDS, is: Given an *arbitrary* metric space compute an embedding into $k$ dimensional Euclidean space which approximates the best possible embedding, in terms of minimizing a particular distortion measure objective. Except for some very special cases no such approximation algorithms were known prior to this work. Applying our moment analysis bounds for JL we are able to obtain the *first* general approximation guarantees to all the discussed measures. The bounds are obtained via convex programming combined with the JL-transform. While the basic idea is quite simple, it is not obvious that it can actually go through. The main obstacle is that all $q$-moment measures are *not* associative. In fact, this is not generally the case that combining two embeddings results in a good final embedding. However, as we show, this is indeed true specifically for JL-type embeddings.

Let $OBJ_q^{(\Pi)} = \{\ell_q\text{-}dist^{(\Pi)}, REM_q^{(\Pi)}, Energy_q^{(\Pi)}, \Phi_{\sigma,q,2}^{(\Pi)}, Stress_q^{(\Pi)}, Stress^{*(\Pi)}_q\}$ denote the set of the objective measures. For $Obj_q^{(\Pi)} \in OBJ_q^{(\Pi)}$, denote $OPT^{(n)} = \inf_{f:X \to \ell_2^n} \left\{ Obj_q^{(\Pi)}(f) \right\}$, and $OPT = \inf_{h:X \to \ell_2^k} \left\{ Obj_q^{(\Pi)}(h) \right\}$. Note that $OPT^{(n)} \leq OPT$. The first step of the approximation algorithm is to compute $OPT^{(n)}$ for a given $Obj_q^{(\Pi)}$, without constraining the target dimension.

**Theorem 8.** *Let $(X, d_X)$ be an $n$-point metric space and $\Pi$ be any distribution. Then for any $q \geq 2$ and for $Obj_q^{(\Pi)} \neq Stress^{*(\Pi)}_q$ there exists a polynomial time algorithm that computes an embedding $f : X \to \ell_2^n$ such that $Obj_q^{(\Pi)}(f)$ approximates $OPT^{(n)}$ to within any level of precision. For $Obj_q^{(\Pi)} = Stress^{*(\Pi)}_q$ there exists a polynomial time algorithm that computes an embedding $f : X \to \ell_2^n$ with $Stress^{*(\Pi)}_q(f) = O\left(OPT^{(n)}\right)$.*

The proof is based on formulating the appropriate *convex* optimization program, which can be solved in polynomial time by interior-point methods. The exception is $Stress_q^*$ which is inherently non-convex. We show that $Stress_q^*$ can be reduced to the case of $Stress_q$, with an additional constant factor loss, and that optimizing for $\Phi_{\sigma,q,2}$ can be reduced to the case of $Energy_q$. The second step in the algorithm is applying the JL to reduce the dimension to the desired number of dimensions $k$.

**Theorem 9.** *For any finite metric $(X, d_X)$, any distribution $\Pi$ over $\binom{X}{2}$, for any $k \geq 3$ and $2 \leq q \leq k-1$, there is a randomized polynomial time algorithm that finds an embedding $F : X \to \ell_2^k$, such that with high probability: $\ell_q\text{-}dist^{(\Pi)}(F) = (1 + O(\frac{1}{\sqrt{k}} + \frac{q}{k-q}))OPT$; and $Obj_q^{(\Pi)}(F) = O(OPT) + O(\sqrt{q/k})$, for $Obj_q^{(\Pi)} \in \{REM_q^{(\Pi)}, Energy_q^{(\Pi)}, \Phi_{\sigma,q,2}^{(\Pi)}, Stress_q^{(\Pi)}, Stress^{*(\Pi)}_q\}$.*

# 6 Empirical experiments

In this section we provide experiments to demonstrate that the theoretical results are exhibited in practical settings. We also compare in the experiments the bounds of the theoretical algorithms (JL and the approximation algorithm based on it) to some of the most common heuristics. In all the experiments, we use Normal distribution (with random variance) for sampling Euclidean input spaces.[5] Tests were made for a large range of parameters, averaging over at least 10 independent tests. The results are consistent for all settings and measures.

We first recall the main theoretical results to be verified. In Theorem 1 and Theorem 2 we showed that for $q < k$ the $\ell_q$-distortion is bounded by $1 + O(1/\sqrt{k}) + O(q/k)$, and all the rest measures are bounded by $O(\sqrt{q/k})$. Particularly, the bounds are independent of the size $n$ and dimension $d$ of the input data set. In addition, our lower bounds in Section 4.1 show that for $\ell_q$-distortion and $REM_q$ measures a phase transition must occur at $q \sim k$ for any dimensionality reduction method, where the bounds dramatically increase from being bounded by a constant to grow with $n$ as $\mathrm{poly}(n)$ for $q < k$. Finally, in Section 5 we exhibited an approximation algorithm for all distortion measures.

The graphs in Fig.1 and Fig.2a describe the following setting: A random Euclidean space $X$ of a fixed size $n$ and dimension $d = n = 800$ was embedded into $k \in [4, 30]$ dimensions with $q = 5$,

by the JL/PCA/Isomap methods. We stress that we run many more experiments for a wide range of parameter values of $n \in [100, 3000]$, $k \in [2, 100]$, $q \in [1, 10]$, and obtained essentially identical qualitative behavior. In Fig. 1a, the $\ell_q$-distortion as a function of $k$ of the JL embedding is shown for $q = 8, 10, 12$. The phase transitions are seen at around $k \sim q$ as predicted. In Fig. 1b the bounds and the phase transitions of the PCA and Isomap methods are presented for the same setting, as predicted. In Fig. 1c, $\ell_q$-distortion bounds are shown for increasing values of $k > q$. Note that the $\ell_q$-distortion of the JL is a *small constant close to* 1, as predicted, compared to values significantly $> 2$ for the compared heuristics. Overall, Fig. 1 clearly shows the superiority of JL to the other methods for all the range of values of $k$. The same conclusions as above hold for $\sigma$-distortion as well,

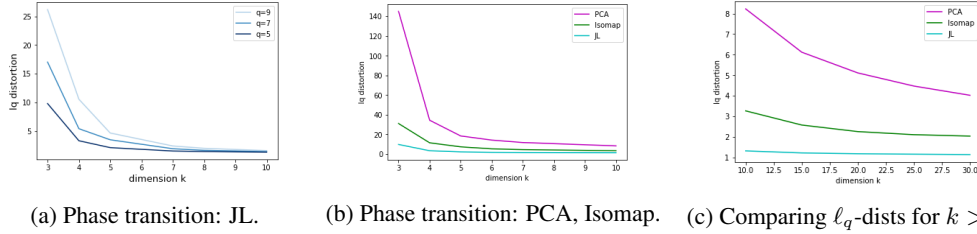

(a) Phase transition: JL.  (b) Phase transition: PCA, Isomap.  (c) Comparing $\ell_q$-dists for $k > q$.

Figure 1: Validating $\ell_q$-distortion behavior.

as shown in Fig. 2a. In the experiment shown in Fig. 2b, we tested the behavior of the $\sigma$-distortion as a function of $d$-the dimension of the input data set, similarly to that of (13)(Fig. 2), and tests are shown for embedding dimension $k = 20$ and $q = 2$. According to our theorems, the $\sigma$-distortion of the JL transform is bounded above by a constant independent of $d$, for $q < k$. Our experiment shows that the $\sigma$-distortion is growing as $d$ increases for both PCA/Isomap, whereas it is a *constant* for JL. Moreover, JL obtains significantly smaller value of $\sigma$-distortion.

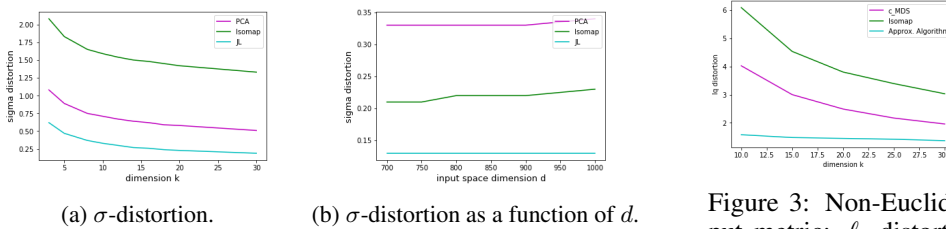

(a) $\sigma$-distortion.  (b) $\sigma$-distortion as a function of $d$.

Figure 2: Validating $\sigma$-dist. behavior.

Figure 3: Non-Euclidean input metric: $\ell_q$-distortion behavior.

In the last experiment, Fig.3, we tested the quality of our approximation algorithm on non-Euclidean input spaces versus the classical MDS and Isomap methods (adapted for non-Euclidean input spaces). The construction of the space is as follows: first, a sampled Euclidean space $X$, of size and dimension $n = d = 100$, is generated as above; second, the interpoint distances of $X$ are distorted with a noise factor $1 + \epsilon$, with $\epsilon \sim N(0, \delta)$, for $\delta < 1$. We ensure that the resulting space is a valid non-Euclidean metric. We then embed the final space into $k \in [10, 30]$ dimensions with $q = 5$. Since the non-Euclidean space is $1 + \epsilon$ far from being Euclidean, we expect a similar behavior to that shown in Fig. 1c. The result clearly demonstrates the superiority of the JL-based approximation algorithm.

## 7 On relevance of distortion measures for ML

In (13) the authors developed a set of properties a distortion measure has to satisfy in order to be useful for machine learning. Here we show that these properties can be generalized and that appropriate modifications of all the measurement criteria discussed in this paper satisfy all of them.

For an embedding $f : X \to Y$, let $\rho_f(u, v)$ be an error function of a pair $u \neq v \in X$, which is a function of the embedded distance and original distance between $u$ and $v$. Let $\rho(f) = (\rho_f(u, v))_{u \neq v \in X}$ denote the vector of $\rho_f(u, v)$ for all pairs $u \neq v \in X$. Let $M_q^{(\Pi)} : \rho(f) \to \mathbb{R}^+$ be a measure function, for any distribution $\Pi$ over $\binom{X}{2}$. For instance, for $\ell_q$-distortion measure and $REM_q$, $\rho_f(u, v) := dist_f(u, v)$ and $\rho_f(u, v) := dist_f(u, v) - 1$, respectively; for $Energy_q$, and $Stress_q$ measures,

$\rho_f(u,v) := |expans_f(u,v) - 1|$; for $\Phi_{\sigma,q,r}$, $\rho_f(u,v) := |expans_f(u,v)/\ell_r\text{-}expans(f) - 1|$. All the measures are then defined by $M_q^{(\Pi)}(\rho(f)) := (E_\Pi[\|\rho(f)\|_q^q])^{1/q}$. In what follows we will omit $\Pi$ from the notation. We propose the generalizations of the ML motivated properties defined in (13):

*Scalability.* Although a measurement criterion may not necessarily be scalable, it can be naturally modified to a scalable version as follows. For every $M_q$, define $\hat{M}_q(\rho(f)) = \min_{\alpha>0} M_q(\rho(\alpha \cdot f))$. Note that the upper and lower bounds that hold for $M_q$ also hold for its scalable version $\hat{M}_q$.

*Monotonicity.* We generalize this property as follows. Let $f, g : X \rightarrow Y$ be any embeddings. For a given measure $M_q$, let $\hat{f}$ and $\hat{g}$ be embeddings minimizing $M_q(\rho(\alpha \cdot f))$ and $M_q(\rho(\alpha \cdot g))$, respectively (over all scaling factors $\alpha > 0$). If $\hat{f}$ and $\hat{g}$ are such that for every pair $u \neq v \in X$ it holds that $\rho_{\hat{f}}(u,v) \geq \rho_{\hat{g}}(u,v)$, then the measure $\hat{M}_q$ is monotone iff $M_q(\rho(\hat{f})) \geq M_q(\rho(\hat{g}))$.

*Robustness to outliers in data/in distances.* The measure $\hat{M}_q$ is said to be robust to outliers if for any embedding $f_n$ of an $n$-point space, any modification $\tilde{f}_n$ where a constant number of changes occurs in either points or distances, it holds that $\lim_{n\to\infty} M_q(\rho(f_n)) = \lim_{n\to\infty} M_q(\rho(\tilde{f}_n))$.

*Incorporation of the probability distribution.* Let $h : X \rightarrow Y$ be an embedding and let $u \neq v \in X$ and $x \neq y \in X$, such that $\Pi(u,v) > \Pi(x,y)$ and $\rho_h(u,v) = \rho_h(x,y)$. Assume that $f : X \rightarrow Y$ is identical to $h$, except over $(u,v)$, and assume that $g$ is identical to $h$, except over $(x,y)$, and assume that $\rho_f(u,v) = \rho_g(x,y)$. Now let $\hat{f}$ and $\hat{h}$ be defined as above and assume $\rho_{\hat{f}}(u,v) \geq \rho_{\hat{h}}(u,v)$. Then, the measure $\hat{M}_q^{(\Pi)}$ is said to incorporate the probability distribution $\Pi$ if $M_q^{(\Pi)}(\rho(\hat{f})) > M_q^{(\Pi)}(\rho(\hat{g}))$.

*Robustness to noise* was not formally defined in (13). Assuming the model of noise that affects the error $\rho$ by at most a factor of $1 + \epsilon$ (alternatively an additive error of $\epsilon$) for each pair, the requirement is that the measure $\hat{M}_q$ will be changed by at most factor of $1 + O(\epsilon)$ (or additive $O(\epsilon)$).

It is easy to see that all distortion criteria (adapted to be scalable as in the first property) discussed in this paper obey all the above properties, implying their relevance to the ML applications.

# 8 Discussion

This work provides a new framework for theoretical analysis of embeddings in terms of performance measures that are of practical relevance, initiating a theoretical study of a wide range of *average case* quality measurement criteria, and providing the first rigorous analysis of these criteria.

We use this framework to analyze the new distortion measure developed in (13) designed to posses machine learning desired properties and show that all considered distortion measures can be adapted to posses similar qualities.

We show nearly tight bounds on the absolute values of all distortion criteria, essentially showing that the JL transform is near optimal for dimensionality reduction for most parameter regimes. When considering other methods, the JL bound can serve as guidance and it would make sense to treat a method useful only when it beats the JL bound. A phase transition exhibited in our bounds provides a direction on how to choose the target dimension $k$, i.e. $k$ should be greater than $q$ by a factor $> 1$. This means that the amount of outlier pairs is diminishing as $k$ grows.

A major contribution of our paper is providing the *first* approximation algorithms for embedding any finite metric (possibly non-Euclidean) into $k$-dimensional Euclidean space with provable approximation guarantees. Since these approximation algorithms achieve near optimal distortion bounds they are expected to beat most common heuristics in terms of the relevant distortion measures. Evidence exists that there is correlation between lower distortion measures and quality of machine learning algorithms applied on the resulting space, such as in (13), where such correlation is experimentally shown between $\sigma$-distortion and error bounds in classification. This evidence suggests that the improvement in distortion bounds should be reflected in better bounds for machine learning applications.

Our experiments show that the conclusions above hold in practical settings as well.

## Acknowledgments

This work is supported by ISF grant #1817/17 and BSF grant #2015813.

## Footnotes

[2] We note that if one desires scale invariability it may always be achieved by defining the scale-invariant measure to be the minimization of the measure over all possible scaling of the embedding. For simplicity we focus on the non-scalable version

[3] It is easy to verify that the general version satisfies all the properties considered in (13).

[4]We note that PCA is proven to minimize $\sum_{u \neq v \in X}(d_{uv}^2 - \hat{d}_{uv}^2)$ over all *projections* into $k$ dimensions (38), but not over *embeddings* (not even linear maps).

[5]We note that (13) used similar settings with Normal/Gamma distributions. Most of our experimental results hold also for the Gamma distribution.

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
