[Supplementary Material]

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

In particular, we show that there are metric space for which the optimal embedding minimizing the $\ell_q$-distortion may be arbitrarily better than any embedding minimizing for this measure which is restricted to be just non-expansive or non-contractive.

Observe that this definition is not scale invariant[2].

In many practical cases, where we may expect a near isometry for most pairs, the moments of distortion may not be sensitive enough and more delicate measures of quality, which examine directly the pairwise additive error, may be desired. The *relative error measure* (REM), commonly used in network applications ((65; 64; 29)) is a most natural choice. It turns out that this measure can be viewed as the moment of distortion about 1. This gives rise to the following generalization of Definition 1:

**Definition 2** ($\ell_q$**-distortion about** $c$**, REM**). *For $c \geq 0$, the $\ell_q$-distortion of $f$ about $c$ is given by:*

$$\ell_q\text{-}dist_{(c)}^{(\Pi)}(f) = (E_\Pi \left[|dist_f(u,v) - c|^q\right])^{\frac{1}{q}}, \quad REM_q^{(\Pi)}(f) = \ell_q\text{-}dist_{(1)}^{(\Pi)}(f).$$

**Additive distortion measures: Stress and Energy.** *Multi Dimensional Scaling* (see (30; 18)) is a well-established methodology aiming at embedding a metric representing the relations between objects into (usually Euclidean) low-dimensional space, to allow feature extraction often used for indexing, clustering, nearest neighbor searching and visualization in many application areas, including machine learning ((62)). The MDS methodology was created in the 1950-60's within the *Psychometrika* community in a series of influential papers by Torgerson (71), Gower (35), Shepard (66), Kruskal (48) and others. Several average additive error criteria for the embedding's quality have been suggested in the context of MDS over the years. Perhaps the most popular is the *stress* measure going back to (48). For $d_{uv} = d_X(u,v)$ and $\hat{d}_{uv} = d_Y(f(u), f(v))$, for normalized nonnegative weights $\Pi(u,v)$ (or distribution) we define the following natural generalizations, which include the classic Kruskal stress $Stress^*{}_2(f)$ and normalized stress $Stress_2(f)$ measures, as well as other common variants in the literature (e.g. (39; 67; 36; 19; 73; 23)): $Stress_q^{(\Pi)}(f) = \left(\frac{E_\Pi\left[|\hat{d}_{uv} - d_{uv}|^q\right]}{E_\Pi[(d_{uv})^q]}\right)^{1/q}$,

and $Stress^*{}_q^{(\Pi)}(f) = \left(\frac{E_\Pi[|\hat{d}_{uv} - d_{uv}|^q]}{E_\Pi[(\hat{d}_{uv})^q]}\right)^{1/q}$. Another popular and used in many fields additive error measure is *energy* and its special case, *Sammon* cost (see e.g. (63; 16; 28; 54; 55; 25)). We define the following generalizations, which include some common variants (e.g. (60; 65; 64; 51)):

$$Energy_q^{(\Pi)}(f) = \left(E_\Pi\left[\left(\frac{|\hat{d}_{uv} - d_{uv}|}{d_{uv}}\right)^q\right]\right)^{1/q} \text{ and } REM_q^{(\Pi)}(f) = \left(E_\Pi\left[\left(\frac{|\hat{d}_{uv} - d_{uv}|^q}{\min\{\hat{d}_{uv}, d_{uv}\}}\right)^q\right]\right)^{1/q}.$$

It immediately follows from the definitions that: $Energy_q^{(\Pi)}(f) \leq REM_q^{(\Pi)}(f) \leq \ell_q\text{-}dist^{(\Pi)}(f)$. Also it's not hard to observe that $Stress_q^{(\Pi)}$ and $Energy_q^{(\Pi')}(f)$ are equivalent via a simple transformation of weights.

**New ML motivated measure: $\sigma$-distortion.** Recently, the paper by (27)[NeurIPS' 18] studies various existing and commonly used quality criteria in terms of their relevance in machine learning. Particularly, they suggest a new measure, $\sigma$- *distortion*, which is claimed to possess all the necessary properties for machine learning applications. We present a generalized version of $\sigma$-distortion[3]. Let $\ell_r\text{-}expans(f) = \left(\binom{n}{2}^{-1} \sum_{u \neq v} (expans_f(u,v))^r\right)^{1/r}$. For a distribution $\Pi$ over $\binom{X}{2}$, let $\Phi_{\sigma,q,r}^{(\Pi)}(f) = \left(E_\Pi\left[\left|\frac{expans_f(u,v)}{\ell_r\text{-}expans(f)} - 1\right|^q\right]\right)^{1/q}$ (for $q = 2$, $r = 1$ this is the square root of the measure defined by (27)). We show that the tools we develop can be applied to $\sigma$-distortion to obtain theoretical bounds on its value.

We further show, generalizing their work, that all other average distortion measures considered here can be easily adapted to satisfy similar ML motivated properties to those defined by (27).

A basic contribution of our paper is showing deeper tight relations between these different objective functions (Section C), and further developing properties and tools for analyzing embeddings for

these measures (Section D). While these measures have been extensively studied from a practical point of view, and many heuristics are known in the literature, almost nothing is known in terms of rigorous analysis and absolute bounds. Moreover, many real-world misconceptions exist about what dimension may be necessary for good embeddings. In this paper we present the *first theoretical analysis* of all these measures providing absolute bounds that shed light on these questions. We exhibit approximation algorithms for optimizing these measures, and further applications.

In this paper we focus only on analyzing objective measures that attempt to preserve metric structure. As a result, some popular objective measures used in applied settings are beyond the scope of this paper, this includes the widely used t-SNE heuristic (which aims at reflecting the cluster structure of the data, and generally does not preserve metric structure), and various heuristics with local structure objectives. When validating our theoretical findings experimentally (Section 6), we chose to compare our results with the most common in practice heuristics PCA/classical-MDS and Isomap amongst the various methods that appear in the literature.

**Moment analysis of dimensionality reduction.** A major bottleneck in processing large data sets is the immense quantity of associated dimensions, making dimensionality reduction a most desirable goal. Our paper proposes the following general question as fundamental basis for theoretical study:

**Problem 1** ((**k, q**)-**Dimension Reduction**). *Given a dimension bound $k$ and $1 \leq q \leq \infty$, what is the least $\alpha(k, q)$ such that every finite subset of Euclidean space embeds into $k$ dimensions with* $\mathrm{Measure}_q \leq \alpha(k, q)$ ?

This question can be phrased for each $\mathrm{Measure}_q$ of practical importance. A stronger demand would be to require a single embedding to *simultaneously* achieve best possible bounds for all values of $q$.

We answer Problem 1 by providing upper and lower bounds on $\alpha(k, q)$. In particular we show that the Johnson-Lindenstrauss dimensionality reduction achieves bounds in terms of $q$ and $k$ that dramatically outperforms a widely used in practice PCA dimensionality reduction. Moreover, our experiments show that the same holds for the Isomap and classical MDS methods.

The bounds we obtain provide some interesting conclusions regarding the expected behavior of dimensionality reduction methods. As expected the bound for the JL method is improving as $k$ grows, confirming the intuition expressed in (27). Yet, countering their intuition, the bound *does not* increase as a function of the original dimension $d$. When considering other embedding methods, the JL bound can serve as guidance and it would make sense to treat the method as useful only when it beats the JL bound. A phase transition, exhibited in our bounds, provides guidance on how to choose the target dimension $k$.

Another consequence arises by combining our result with the embedding of (3) (by composing it with JL): we obtain an embedding of general spaces into *constant* dimensional Euclidean space with *constant* distortion (for all discussed measures). Here, the dimension is constant even if the original space is not doubling, improving on the result obtained in (27).

**Approximation algorithms.** The bounds achieved for the Euclidean $(k, q)$-dimension reduction are then applied to provide the *first* approximation algorithms for embedding general metric spaces into low dimensional Euclidean space, for all the various distortion criteria. This is based on composing convex programming with the JL-transform. It should be stressed that such a composition may not necessarily work in general, however, we are able to show that this yields efficient approximation algorithms for all the criteria considered in this paper.

Since these approximation algorithms achieve near optimal distortion bounds they are expected to beat most common heuristics in terms of the relevant distortion measures. Evidence exists that there is correlation between lower distortion measures and quality of machine learning algorithms applied on the resulting space, such as in (27) where such correlation is shown between $\sigma$-distortion and error bounds in classification. This evidence suggests that the improvement in distortion bounds should be reflected in better bounds for machine learning applications.

**Empirical Experiments.** We validate our theoretical findings experimentally on various randomly generated Euclidean and non-Euclidean metric spaces, in Section 6. In particular, as predicted by our lower bounds, the phase transition is clearly seen in the JL, PCA and Isomap embeddings for all

the measurement criteria. Moreover, in our simulations the JL based approximation algorithm (as well as the JL itself, when applied on Euclidean metrics) has shown dramatically better performance than the PCA and Isomap heuristics for all distortion measures, indicating that the JL-based approximation algorithm is a preferable choice when the preservation of metric properties is desirable.

**Additional Applications.** Our results have further implications to improved embedding of general metrics and *distance oracles* (appear in Appendix K), and to *metric hyper sketching* (appears in Appendix L).

**Related work.** Analyzing refined notions of the distortion in metric embedding was suggested by (47) in the context of network embedding (65). The authors of (47) noted that heuristic embeddings into low dimensional Euclidean space had surprisingly good performance, and opted for a theoretical explanation. They introduced the notion of partial and scaling embeddings (originally termed slack and gracefully-degrading), and gave some preliminary results. Following, (1), (3) provided asymptotically optimal partial and scaling embedding to $\ell_p$ spaces, and bounds on the moments of distortion. These notions were further studied in various contexts, e.g., spanners (24), spanning trees (5; 15), volume respecting (4).

Sketching and distance labeling have been intensively studied over the last two decades. Some of the notable results are the sketching of (43) for $\ell_p$, for $0 < p \leq 2$, and that for any $k \geq 1$ there is a labeling scheme for arbitrary metrics with stretch $k$ and size $\tilde{O}(n^{1/k})$ per point (45; 61; 70; 43). There is a host of other results on distance sketching for various metric spaces, we refer the reader to (9) and the references therein for some more background.

For Euclidean embedding, (53) showed that using SDP one can obtain arbitrarily good approximation of the distortion. However, such a result is impossible when restricting the target dimension to $k$, as (58) showed that unless P=NP, the approximation factor must be at least $n^{\Omega(1/k)}$.

For completeness we mention a few works on the positive side of this line of research: in (20) authors provide $O\left(n^{1/3}\right)$-approximation algorithm for embedding unweighted tress into $\mathbb{R}^1$, and $O\left(n^{1/2}\right)$-approximation factor for unweighted graphs. In (21) authors develop an $O\left(n^{1/3}\right)$ approximation algorithm for embedding ultrametrics in $\ell_2^2$.

Of all the measures studied in this paper, the only measure that was previously theoretically studied is $Stress_q$. Cayton and Dasgupta (22) show that computing an embedding into $\mathbb{R}^1$ with optimal $Stress_q$ is NP-hard, for any given $q$. To the best of our knowledge, the only approximation algorithms known for this problem are the following: (38) gave a 2-approximation algorithm to $Stress_\infty$ for embedding finite metrics into $\mathbb{R}^1$; (34) presented $O(\log^{1/q} n)$-approximation algorithm to $Stress_q$ for embedding any $n$-point metric into $\mathbb{R}^1$; (11) constructed $O(1)$-approximation algorithm to $Stress_\infty$ for embedding finite metric spaces into $\ell_1^2$.

Compared to these results, the approximation algorithms presented in this paper, are far more general in that they apply to several distortion measures, and their weighted versions, and to general values of $q$ and dimension $k$. Our result for the additive distortion measures, and in particular for $Stress_q$, are within $O(1)$ factor of the optimal cost, though with an additional additive term that diminishes as function of $q/k$. Our result for $\ell_q$-$dist$ provides a multiplicative approximation (with no additive term).

## 2   On the limitations of classical MDS

We begin with examining the widely used and quoted heuristic, known as classical MDS or PCA(71; 35), which is often practically applied, even on non-Euclidean metric spaces. In the literature there are many misconceptions regarding this heuristic. However, we show that it may perform extremely poor in terms of all basic additive measures, and moreover this holds even if the input metric is effectively low-dimensional.

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

The proof for a fixed value of $q$ is in AppendixG:Theorem 45 and Theorem 51. The proof for the simultaneous version is in AppendixG.2: Theorem 55.

Note that for large $q$ our theorem shows that a phase transition emerges around $q = k$. The necessity of this phenomenon is implied by nearly tight lower bounds given in Section 4.1.

### 3.2 Additive distortion and $\sigma$-distortion measures analysis

The following theorem provides tight upper bounds for all the additive distortion measures and for $\sigma$-distortion, for $1 \leq q \leq k - 1$. This follows from analyzing the $REM$ (via similar approach to the raw moments analysis) and tight relations between the measures (Appendix G.5: Theorem 59).

**Theorem 2.** *Given a finite set $X \subset \ell_2^d$ and an integer $k \geq 2$, let $f : X \to \ell_2^k$ be the JL transform of dimension $k$. For any distribution $\Pi$ over $\binom{X}{2}$, with constant probability, for all $1 \leq r \leq q \leq k - 1$:*
$$REM_q^{(\Pi)}(f), Energy_q^{(\Pi)}(f), \Phi_{\sigma,q,r}{}^{(\Pi)}(f), Stress_q^{(\Pi)}(f), Stress^*{}_q^{(\Pi)}(f) = O\left(\sqrt{q/k}\right).$$

The more challenging part of the analysis is figuring out how good are the JL performance bounds. Therefore our main goal is the task of establishing lower bounds for Problem 1.

## 4 Partially tight lower bounds: $q < k$

As the stated upper bounds behave differently in different ranges of values of $q$, the same holds for the lower bounds. Therefore we must provide lower bounds for each range. We show that JL is essentially optimal when *simultaneous* guarantees are required (Appendix I). If that requirement is removed, it is still the case for most of the ranges of $q$. Providing lower bound for each range requires a different technique. One of the most interesting cases, is the proof of the lower bound of $1 + \Omega(q/(k - q))$ for the range $1 \leq q \leq k - 1$. For $q \leq \sqrt{k}$, this turns out to be a consequence of the tightness results for the additive distortion measures and $\sigma$-distortion, which are shown to be tight for $q \geq 2$. The proof is based on a delicate application of the technique of (8).

**Additive distortion measures.** We show that the analysis of the JL transform for the additive measures and $\sigma$-distortion, provides tight bounds for all values of $2 \leq q \leq k$. Due to tight relations between the additive measures, the lower bounds for all measures follow from Energy measure (Appendix H.1:Claim 64). Let $E_n$ denote the $n$-point equilateral metric space.

**Claim 3.** *For all $k \geq 2$, $k \geq q \geq 2$, and $n \geq 4 \left(9 \cdot \frac{k}{q}\right)^{q/2}$, for all $f : E_n \to \ell_2^k$ it holds that $Energy_q(f) = \Omega(\sqrt{\frac{q}{k}})$.*

From the proof of the above claim we derive the lower bound for $1 \leq q < 2$, in Appendix H.1:Claim 65:

**Claim 4.** *For all $k \geq 1$, $1 \leq q < 2$, and $n \geq 18k$, for all $f : E_n \to \ell_2^k$, $Energy_q(f) = \Omega\left(\frac{1}{k^{1/q}}\right)$.*

**Moments of distortions.** A more involved argument shows that Claim 3 implies a lower bound on the $\ell_q$-distortion as well, proved in Appendix H.2: Corollary 7.

**Corollary 1.** *For any $k \geq 1$, let $E_n$ be an equilateral metric space, for $n \geq 18k$. Then, for any embedding $f : E_n \to \ell_2^k$ it holds that $\ell_q\text{-}dist(f) = 1 + \Omega\left(\frac{q}{k}\right)$, for all $1 \leq q \leq \sqrt{k}$.*

The following is proven in Appendix H.2: Theorem 66, based on (49):

**Theorem 5.** *For all $k \geq 16$, for all $N$ large enough, there is a metric space $Z \subseteq \ell_2$ on $N$ points, such that for any $F : Z \to \ell_2^k$ it holds that $\ell_q\text{-}dist(F) \geq 1 + \Omega\left(\frac{q}{k-q}\right)$, for all $q = \Omega\left(\sqrt{k \log k}\right)$.*

## 4.1 Phase transition: moment analysis lower bounds for $q \geq k$

An important consequence of our analysis is that the $q$-moments of the distortion (including $REM_q$), exhibit am impressive *phase transition* phenomenon occurring around $q = k$. This follows from lower bounds for $q \geq k$. The case $q = k$ (and $\approx k$) is of special interest where we obtain a tight bound of $\Theta((\sqrt{\log n})^{1/k})$ in Appendix H.3: Theorem 67:

**Theorem 6.** *Any embedding $f : E_n \rightarrow \ell_2^k$ has $\ell_k\text{-}dist(f) = \Omega((\sqrt{\log n})^{1/k}/k^{1/4})$, for any $k \geq 1$.*

Hence, for any $q$, the theorem tells that only $k \geq 1.01q$ may be suitable for dimensionality reduction. This new consequence may serve an important guide for practical considerations, that seems to be missing prior to our work.

In Appendix H.2: Claims 73, 74 we prove the following lower bounds:

**Claim 7.** *For any embedding $f : E_n \rightarrow \ell_2^k$, for all $k \geq 1$, for all $q > k$, $\ell_q\text{-}dist(f) = \Omega(\max\{n^{(\frac{1}{2\lceil k/2 \rceil} - \frac{2}{q})}, n^{\frac{1}{2k} - \frac{1}{2q}}\})$.*

# 5 Approximate optimal embedding of general metrics

Perhaps the most basic goal in dimensionality reduction theory and essentially, the main problem of MDS, is: Given an *arbitrary* metric space compute an embedding into $k$ dimensional Euclidean space which approximates the best possible embedding, in terms of minimizing a particular distortion measure objective. [6]. Except for some very special cases no such approximation algorithms were known prior to this work. Applying our moment analysis bounds for JL we are able to obtain the *first* general approximation guarantees to all the discussed measures.

The bounds are obtained via convex programming combined with the JL-transform. While the basic idea is quite simple, it is not obvious that it can actually go through. The main obstacle is that all $q$-moment measures are *not* associative. In fact, this is not generally the case that combining two embeddings results in a good final embedding. However, as we show, this is indeed true specifically for JL-type embeddings.

Let $OBJ_q^{(\Pi)} = \{\ell_q\text{-}dist^{(\Pi)}, REM_q^{(\Pi)}, Energy_q^{(\Pi)}, \Phi_{\sigma,q,2}^{(\Pi)}, Stress_q^{(\Pi)}, Stress^{*(\Pi)}_q\}$ denote the set of the objective measures. Given any $Obj_q^{(\Pi)} \in OBJ_q^{(\Pi)}$, denote $OPT^{(n)} = \inf_{f:X \rightarrow \ell_2^n} \left\{ Obj_q^{(\Pi)}(f) \right\}$, and $OPT = \inf_{h:X \rightarrow \ell_2^k} \left\{ Obj_q^{(\Pi)}(h) \right\}$. Note that $OPT^{(n)} \leq OPT$.

The first step of the approximation algorithm is to compute $OPT^{(n)}$ for a given $Obj_q^{(\Pi)}$, without constraining the target dimension.

**Theorem 8.** *Let $(X, d_X)$ be an $n$-point metric space and $\Pi$ be any distribution. Then for any $q \geq 2$ and for $Obj_q^{(\Pi)} \neq Stress^{*(\Pi)}_q$ there exists a polynomial time algorithm that computes an embedding $f : X \rightarrow \ell_2^n$ such that $Obj_q^{(\Pi)}(f)$ approximates $OPT^{(n)}$ to within any level of precision. For $Obj_q^{(\Pi)} = Stress^{*(\Pi)}_q$ there exists a polynomial time algorithm that computes an embedding $f : X \rightarrow \ell_2^n$ with $Stress^{*(\Pi)}_q(f) = O\left(OPT^{(n)}\right)$.*

The proof is based on formulating the appropriate *convex* optimization program, which can be solved in polynomial time by interior-point methods. The formulation of the optimization problem follows the approach of (53) who used an SDP optimization to compute an optimal worst case distortion embedding. The exception is $Stress^*_q$ which is inherently non-convex. We show in Appendix J: Claim 78 that $Stress^*_q$ can be reduced to the case of $Stress_q$, with an additional constant factor loss. In addition, in Appendix J: Claim 81 we show that the optimizing for $\Phi_{\sigma,q,2}$ can be reduced to the case of $Energy_q$. The details of the proof are in Appendix J:Theorem 80.

The second step in the approximation algorithm is applying the JL to reduce the dimension to the desired number of dimensions $k$. Next theorem states the approximation result (proved in Appendix J:Theorem 82.) The proof is based on the composition results for the additive measures, presented in Appendix D.2.

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

[5]This means that no bound for $q$ can be asymptotically improved without losing in the bounds for other values of $q$.

[6](22) prove that finding the optimal embedding into one dimension minimizing $Stress_q$ for any fixed $q$ is NP-hard.

[7]We note that (27) used similar settings with Normal/Gamma distributions. Most of our experimental results hold also for the Gamma distribution.

[8] The formula for $\rho_f(u, v)$ for $f$ being $Stress^*$ is more involved and omitted from this version.

[9]Unlike other measures we consider, this notion generally is not valuable, when the embedding is arbitrary and the weight function is known in advance, since in this case it can be made to equal 1. However, it is valuable when the embedding has special properties, such as being non-expansive or non-contractive (as in (3)), or being efficiently computable and oblivious, such as the case of the JL-transform.

[10]We observe that by normalizing the embeddings of ABN and the JL to be non-expansive, we obtain that any $n$-point metric space $(X, d_X)$ can be embedded with non-expansive embedding $f$ into $\ell_2^k$ with $\ell_q\text{-}dist(f) = O(\log n \sqrt{\log n})$, for any $2 \leq k$ and $q < k$.

[11]By Markov's inequality, we can obtain that with probability at least $1 - \delta$: $\ell_q\text{-}dist^{(\Pi)}(f) \leq 1 + \frac{1}{\delta} \cdot \left(\frac{\max\{2,q\}}{k-q} + \frac{1}{\sqrt{\pi}\sqrt{k}}\right)$.

[13]We note that $n$ can be substituted by $\sqrt{support(\Pi)}$.

[14] Where $k^- = \left(1 - \frac{1}{\log n + 1}\right) \cdot k$.

[15]We note that, essentially, our technique also provides approximation guarantees for values of $q > k-1$, yet they are not interesting for $Energy_q$, $Stress_q$ and $Stress^*$, since these measures are less than 1 (as the approximation we provide is additive).

[16]The constant $\tilde{c}$ is defined in Remark 46.

[17]It should be noted that sketching scheme which answers pairwise distances does not necessarily extend to hyper-sketching, since answering correctly all the $\binom{|S|}{2}$ distances in a set may fail with probability $\approx \delta\binom{|S|}{2}$.

[18]In fact, the worst case distortion may be at most $O(\log n)$.

[19]In fact, the worst case distortion may be at most $O(\log n)$.

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

## A  Preliminaries and basic definitions

For metric spaces $(X, d_X)$ and $(Y, d_Y)$, an injective map $f : X \to Y$ is an embedding. The *distortion* of $f$ is defined by $dist(f) = \max_{u \neq v \in X}\{expansion_f(u,v)\} \cdot \max_{u \neq v \in X}\{contraction_f(u,v)\}$.

**Scalable metric space and family of scalable metric spaces.** Most of our results are general in a sense that they hold not only for a hosting space $Y$ being a normed space, but rather for any 'scalable' metric space.

**Definition 3.** *Let $(Y, d_Y)$ be an infinite metric space. We say that $Y$ is a scalable metric space if*

$$\forall Y' \subseteq Y, \ \forall \alpha > 0, \ \exists Y'', \ \exists \text{ a bijection } F : Y' \to Y''$$
$$s.t. \ \forall u' \neq v' \in Y', \ d_Y(F(u'), F(v')) = \alpha \cdot d_Y(u', v')$$

*We say that $Y''$ is an $\alpha$-scaled version of $Y'$, and we refer it by $\alpha \cdot Y'$.*

To capture the notion of scalable finite metric space we extend the above definition as follows.

**Definition 4.** *Let $\mathcal{Y}$ be a family of an $n$-point metric spaces. We say that $\mathcal{Y}$ is a scalable family of metric spaces if*

$$\forall (Y', d_{Y'}) \in \mathcal{Y}, \ \forall \alpha > 0, \ \exists (Y'', d_{Y''} \in \mathcal{Y}, \ \exists \text{ a bijection } F : Y' \to Y''$$
$$s.t. \ \forall u' \neq v' \in Y', \ d_{Y''}(F(u'), F(v')) = \alpha \cdot d_{Y'}(u', v')$$

*We say that $Y''$ is an $\alpha$-scaled version of $Y'$, and we refer it by $\alpha \cdot Y'$.*

For results that hold for a scalable metric space $Y$, we will use the following informal notation.

**Notation 1.** *Let $(X, d_X)$ be an $n$-point metric space, and $(Y, d_Y)$ be a scalable metric space (or a memeber of a scalable family of metric spaces). Let $f : X \to Y$ be an embedding, and $\alpha > 0$ a scaling parameter. Let $Y''$ be an $\alpha$ -scaled version of $f(X)$, and let $F$ be the bijection between $Y''$ and $f(X)$. We denote by $\alpha \cdot f : X \to Y$ the embedding defined by*

$$\forall u \in X, \ \alpha \cdot f(u) = F(f(u))$$

Another useful notation is bellow:

**Notation 2.** *Let $(X, d_X)$, $(Y, d_Y)$ be any metric spaces, and let $X' \subseteq X$ be any subset. Given any embedding $f : X \to Y$ we denote by $f|X'$ the embedding $f$ induced on the set $X'$, i.e. for all $x \in X', f|X'(x) = f(x)$, and for all $x \in X \setminus X'$, $f|X'(x)$ is undefined.*

The following basic probability lemma is useful in providing an upper bound on the probability that a weighted sum of nonnegative (dependent) random variables is small.

**Lemma 10.** *Let $X_1, X_2, \ldots X_n$ be nonnegative random variables. Let $w_1, w_2, \ldots, w_n$ be non-negative weights normalized such that $\sum_{1 \leq i \leq n} w_i = 1$, and let $Y = \sum_{1 \leq i \leq n} w_i X_i$. For all $i$ let $p_i = \Pr[X_i \leq \alpha]$, for some $\alpha \geq 0$, and let $z = \Pr[Y \leq \beta\alpha]$, for some $0 < \beta < 1$. Then*

$$z \leq \frac{\sum_{1 \leq i \leq n} w_i p_i}{1 - \beta}.$$

*In particular, if $p_i \leq p$ for all $i$ then $z \leq \frac{p}{1-\beta}$.*

*Proof.* Define $S(\alpha) = \{1 \leq i \leq n | X_i \leq \alpha\}$. For every $S \subseteq [n]$ let $z_S$ denote the probability that $S(\alpha) = S$. Observe that if $\sum_{i \notin S(\alpha)} w_i \geq \beta$ then $Y = \sum_{1 \leq i \leq n} w_i X_i > \beta\alpha$. Conversely, if $Y \leq \beta\alpha$ then $\sum_{i \in S(\alpha)} w_i \geq 1 - \beta$. Let $\mathcal{B} = \{S \subseteq [n] | \sum_{i \in S} w_i \geq 1 - \beta\}$. Then

$$z = \Pr[Y \leq \beta\alpha] \leq \Pr[\sum_{i \in S(\alpha)} w_i \geq 1 - \beta] = \sum_{S \in \mathcal{B}} z_S.$$

As for every $i$ we have

$$p_i = \Pr[X_i \leq \alpha] = \sum_{S \ni i} z_S \geq \sum_{S \in \mathcal{B}; S \ni i} z_S$$

We get that

$$\sum_{1 \leq i \leq n} w_i p_i \geq \sum_{1 \leq i \leq n} w_i \sum_{S \in \mathcal{B}; S \ni i} z_S = \sum_{S \in \mathcal{B}} z_S \sum_{i \in S} w_i \geq (1 - \beta)z.$$

$\square$

**Additive distortions.** To put our results in context we make the following remark. Given any finite metric space $(X, d_X)$, and any scalable metric space $(Y, d_Y)$, there exists an embedding $f : X \to Y$ such that $Energy_q^{(\Pi)}(f)$, $Stress_q^{(\Pi)}(f)$ are less then 1 (or such that $Stress^{*(\Pi)}_q(f)$ less than 1). This is because we can embed $X$ into some $n$ point set of $Y$, and either scale it by factor $\delta > 0$ small enough for $Energy$ and $Stress$ to be less than 1, or by factor $\delta > 0$ large enough for $Stress^*$ be less than 1.

# B  Gamma function: inequalities and estimations

The following was proved in Wendel (75): For all $x > 0$, for all $0 \le r \le 1$,

$$\left(\frac{x}{x+r}\right)^{1-r} \le \frac{\Gamma(x+r)}{x^r \Gamma(x)} \le 1. \tag{1}$$

From this, we derive: For all $x > 0$, for all $0 \le r \le 1$, such that $r < x$,

$$\frac{x^r \Gamma(x-r)}{\Gamma(x)} \le \frac{x}{x-r}. \tag{2}$$

In (17), the following inequalities were proved:

$$\forall\, x \ge 0, \qquad \frac{\sqrt{2}\left(x+\frac{1}{2}\right)^{x+\frac{1}{2}}}{e^x} \le \Gamma(x+1) < \frac{\sqrt{2\pi}\left(x+\frac{1}{2}\right)^{x+\frac{1}{2}}}{e^{x+\frac{1}{2}}}. \tag{3}$$

$$\forall x \ge 1, \quad \sqrt{2\pi}x^{x-1}e^{-x}\sqrt{x+1/6} \le \Gamma(x) \le \sqrt{2\pi}x^{x-1}e^{-x}\sqrt{x+(e^2/2\pi-1)}. \tag{4}$$

We prove the following lemma:

**Lemma 11.** *For all $x > 0$, $\sqrt{2\pi}x^{x-1}e^{-x}\sqrt{x} \le \Gamma(x) \le \sqrt{2\pi}x^{x-1}e^{-x}\sqrt{x+1/2}$.*

*Proof.* For $x \ge 1$ the inequality is just a weaker form of the inequality 4. Thus, we focus on $0 < x < 1$. To prove the right-hand side, we use the right-hand side of the Eq.3. For any $x > 0$:

$$\Gamma(x+1) = x \cdot \Gamma(x) \le \frac{\sqrt{2\pi}\left(x+\frac{1}{2}\right)^{x+\frac{1}{2}}}{e^{x+\frac{1}{2}}}.$$

Thus, it is enough to prove that for any $x > 0$:

$$\frac{(x+1/2)^{x+1/2}}{\sqrt{e}} \le x^x\sqrt{x+1/2},$$

which is true since $(1+1/2x)^x \le \sqrt{e}$. For the the left-hand side, by the same consideration, it is enough to show for all $0 < x < 1$:

$$\sqrt{\pi}x^x\sqrt{x} \le (x+1/2)^{x+1/2} \Leftrightarrow \sqrt{\pi} \le (1+1/2x)^{x+1/2},$$

which is true since the function $f(x) = (1+1/2x)^{x+1/2}$ is decreasing, and therefore for $0 < x < 1$, $f(x) \ge (1.5)^{1.5} \ge \sqrt{\pi}$. $\qquad\square$

As a corollary, we obtain an estimation of the flowing term, which we frequently use in the paper:

**Lemma 12.** *For any $k \ge 1$, $\dfrac{\left(\frac{k}{2}\right)^{\frac{k}{2}-1}}{\Gamma\left(\frac{k}{2}\right)} \le \dfrac{e^{k/2}}{\sqrt{\pi k}}$.*

Another useful inequality, which involves incomplete Gamma function, is proven in (13).

**Lemma 13.** *For all $x > 0$, for all $0 \le r \le 1$, for all $y \ge 0$ it holds*

$$\Gamma(x+r, y) \le x^r \Gamma(x, y) + r \cdot \frac{y^x x^{r-1}}{e^y}.$$

The proof is a generalization of inequality in (75).

## C  Moment analysis notions: basic properties and relations

In this section we define a basic notions of the moment analysis of metric embeddings and state their several useful properties.

**Definition 5** ($\ell_q$-**expans,** $\ell_q$-**contr**). *Let $f : X \to Y$ be an embedding. Given any distribution $\Pi$ over $\binom{X}{2}$, any $q \geq 1$, we define:*

$$\ell_q\text{-}expans^{(\Pi)}(f) = \left(E_\Pi\left[(expans_f(u,v))^q\right]\right)^{1/q}, \quad \ell_q\text{-}contr^{(\Pi)}(f) = \left(E_\Pi\left[(contr_f(u,v))^q\right]\right)^{1/q}.$$

We refer to $\ell_q\text{-}expans^{(\mathcal{U})}(f)$ by $\ell_q\text{-}expans(f)$, and to $\ell_q\text{-}contr^{(\mathcal{U})}(f)$ by $\ell_q\text{-}contr(f)$. Note that for any embedding $f$ it holds that $\max\left\{\ell_q\text{-}contr^{(\Pi)}(f), \ell_q\text{-}expans^{(\Pi)}(f)\right\} \leq \ell_q\text{-}dist^{(\Pi)}(f)$.

**Lemma 14.** *Let $(X, d_X)$ be a finite metric, and let $(Y, d_Y)$ be a metric space. For any $f : X \to Y$:*

1. *$dist(f) \leq (\ell_\infty\text{-}dist(f))^2$.*

2. *For any distribution $\Pi$ over $\binom{X}{2}$, $\forall 1 \leq s \leq t \leq \infty$, $\ell_s\text{-}dist^{(\Pi)}(f) \leq \ell_t\text{-}dist^{(\Pi)}(f) \leq \Upsilon(\Pi)^{\frac{1}{s}-\frac{1}{t}} \ell_s\text{-}dist^{(\Pi)}(f)$, where $\Upsilon(\Pi) = \max_{\Pi(u,v) \neq 0}\left\{(\Pi(u,v))^{-1}\right\}$.*

*Proof.* The first item immediately follows from the definitions of the $dist(f)$ and of the $\ell_\infty\text{-}dist(f)$. The L.H.S. of the second item immediately follows from the Jensen's inequality. To prove the R.H.S. is equivalent to prove the following inequity

$$\left(\sum_{u \neq v \in X} \Pi(u,v)\Upsilon(\Pi)(dist_f(u,v))^t\right)^{1/t} \leq \left(\sum_{u \neq v \in X} \Pi(u,v)\Upsilon(\Pi)(dist_f(u,v))^s\right)^{1/s},$$

which is equivalent to showing that

$$\sum_{u \neq v \in X} \Pi(u,v)\Upsilon(\Pi)(dist_f(u,v))^t \leq \left(\sum_{u \neq v \in X} \Pi(u,v)\Upsilon(\Pi)(dist_f(u,v))^s\right)^{t/s}.$$

Note that real function $p(y) = y^{t/s}$ is a convex function (since $t \geq s$), and also $p(0)=0$, therefore, $p(y)$ is superadditive function. Therefore, we have

$$\left(\sum_{u \neq v \in X} \Pi(u,v)\Upsilon(\Pi)(dist_f(u,v))^s\right)^{t/s} \geq \sum_{u \neq v \in X} (\Pi(u,v)\Upsilon(\Pi))^{t/s}(dist_f(u,v))^t,$$

what finishes the proof, since $\Pi(u,v)\Upsilon(\Pi) \geq 1$, for all $u \neq v \in X$. $\qquad\square$

**Distortion of $\ell_q$-norm.** When considering specific embeddings[9] the following notion of *distnorm* is a natural measure, which we show to be closely related to the notion of the $\ell_q$-distortion.

**Definition 6.** *Let $f : X \to Y$ be an embedding. Given any distribution $\Pi$ over $\binom{X}{2}$, any $q \geq 1$, the distortion of $\ell_q$-norm of $f$ with respect to $\Pi$ is defined by*

$$distnorm_q^{(\Pi)}(f) = \max\left\{\frac{E_\Pi[(d_Y(f(u),f(v)))^q]^{1/q}}{E_\Pi[(d_X(u,v))^q]^{1/q}}, \frac{E_\Pi[(d_X(u,v))^q]^{1/q}}{E_\Pi[(d_Y(f(u),f(v)))^q]^{1/q}}\right\}.$$

For $\Pi = \mathcal{U}$ we use the notation $distnorm_q^{(\mathcal{U})}(f) = distnorm_q(f)$. The following claim establishes the connections to the $\ell_q$-distortion notion.

**Claim 15.** *For any $f : X \to Y$, any distribution $\Pi$ over $\binom{X}{2}$, any $q \geq 1$ it holds that:*

$$distnorm_q^{(\Pi)}(f) = \max\left\{\ell_q\text{-}expans^{(\Pi')}(f), \frac{1}{\ell_q\text{-}expans^{(\Pi')}(f)}\right\} \leq \max\left\{\ell_q\text{-}expans^{(\Pi')}(f), \ell_q\text{-}contr^{(\Pi')}(f)\right\}$$

$$\leq \ell_q\text{-}dist^{(\Pi')}(f),$$

*where the distribution $\Pi'$ over the pairs $\binom{X}{2}$ is defined by $\Pi'(u,v) = \frac{\Pi(u,v)\cdot(d_X(u,v))^q}{E_\Pi[(d_X(u,v))^q]}$.*

*Proof.* The equality follows by substituting the formula of $\Pi'$ into the definition of $\ell_q\text{-}expans^{(\Pi)}(f)$. Jensen's inequality implies $\frac{1}{\ell_q\text{-}expans^{(\Pi')}(f)} \leq \ell_q\text{-}contr^{(\Pi')}(f)$. $\qquad\square$

**Claim 16.** *For any $f : X \to Y$, any distribution $\Pi$ over $\binom{X}{2}$, and any $q \geq 1$ it holds that:*

$$distnorm_q^{(\Pi)}(f) \leq \ell_q\text{-}dist^{(\Pi')}(f),$$

*where the distribution $\Pi'$ over the pairs $\binom{X}{2}$ is defined by: for all $(u,v) \in \binom{X}{2}$, $\Pi'(u,v) = \frac{\Pi(u,v)\cdot(d_X(u,v))^q}{E_\Pi[(d_X(u,v))^q]}$.*

*Proof.* Note that $\Pi'$ is indeed a distribution. We have

$$\left(\ell_q\text{-}dist^{(\Pi')}(f)\right)^q = E_{\Pi'}\left[(dist_f(u,v))^q\right] \geq E_{\Pi'}\left[\left(\frac{d_Y(f(u),f(v))}{d_X(u,v)}\right)^q\right] =$$

$$\frac{1}{E_\Pi\left[(d_X(u,v))^q\right]} \cdot \sum_{u \neq v \in X} \Pi(u,v)(d_Y(f(u),f(v)))^q = \frac{E_\Pi[(d_Y(f(u),f(v)))^q]}{E_\Pi[(d_X(u,v))^q]}.$$

For the second direction we have by Jensen's inequality

$$\frac{1}{\left(\ell_q\text{-}dist^{(\Pi')}(f)\right)^q} = \frac{1}{E_{\Pi'}\left[(dist_f(u,v))^q\right]} \leq E_{\Pi'}\left[\frac{1}{(dist_f(u,v))^q}\right] \leq$$

$$E_{\Pi'}\left[\left(\frac{d_Y(f(u),f(v))}{d_X(u,v)}\right)^q\right] = \frac{E_\Pi[(d_Y(f(u),f(v)))^q]}{E_\Pi[(d_X(u,v))]^q}.$$

This finishes the proof of the claim. $\qquad\square$

We conclude this section with the following basic fact.

**Fact C.1.** *Let $(X, d_X)$, $(Y, d_Y)$ be $n$-point metric spaces. For any distribution $\Pi$ over $\binom{n}{2}$, $\forall q \geq 1$ it holds that $\ell_q\text{-}dist^{(\Pi)}(f) = \ell_q\text{-}dist^{(\Pi)}(f^{-1})$. Particularly, for $\hat{f} = \underset{f:X\to Y}{\operatorname{argmin}}\left\{\ell_q\text{-}dist^\Pi(f)\right\}$, and $\hat{g} = \underset{g:Y\to X}{\operatorname{argmin}}\left\{\ell_q\text{-}dist^{(\Pi)}(g)\right\}$ it holds that $\hat{g} \equiv f^{-1}$.*

**Additive distortion measures.** Bellow we list the relations between the additive distortion measures.

**Claim 17.** *For any $f : X \to Y$, for any distribution $\Pi$ over $\binom{X}{2}$, for any $q \geq 1$, there is a distribution $\Pi'$ over $\binom{X}{2}$ such that $Stress_q^{(\Pi)}(f) = Energy_q^{(\Pi')}(f)$.*

*Proof.* Let $\Pi'(ij) = \frac{\Pi(ij)(d_{ij})^q}{\sum_{s\neq t}\Pi(st)(d_{st})^q}$, then $Energy_q^{(\Pi')}(f) = Stress_q^{(\Pi)}(f)$. $\qquad\square$

**Claim 18.** *For all $q \geq 1$ it holds that $\left(Energy_q^{(\Pi)}(f)\right)^q \leq \left(\ell_q\text{-}expans^{(\Pi)}(f)\right)^q + 1$.*

*Proof.*

$$\left(Energy_q^{(\Pi)}(f)\right)^q = \sum_{1 \le i < j \le n} \Pi(i,j) \left(\left|\frac{\hat{d}_{ij} - d_{ij}}{d_{ij}}\right|\right)^q = \sum_{1 \le i < j \le n} \Pi(i,j) \cdot \max\left\{\left(\frac{\hat{d}_{ij}}{d_{ij}} - 1\right)^q, \left(1 - \frac{\hat{d}_{ij}}{d_{ij}}\right)^q\right\}.$$

Let $A = \{(i,j) | expans_f(x_i, x_j) \ge 1\}$, and $B = \{(i,j) | expans_f(x_i, x_j) < 1\}$. Then, we have

$$\left(Energy_q^{(\Pi)}(f)\right)^q = \sum_{(i,j) \in A} \Pi(i,j)(expans_f(x_i, x_j) - 1)^q + \sum_{(i,j) \in B} \Pi(i,j)(1 - expans_f(x_i, x_j))^q \le$$

$$\sum_{(i,j) \in A} \Pi(i,j)(expans_f(x_i, x_j))^q + \sum_{(i,j) \in B} \Pi(i,j) \le (\ell_q\text{-}expans(f))^q + 1,$$

as required. $\qquad\square$

**Claim 19.** *Let $(X, d_X)$ be any finite metric space, and $(Y, d_Y)$ be any metric space. For any distribution $\Pi$ over $\binom{X}{2}$, for any embedding $f : X \to Y$, for all $q \ge 1$ it holds that*

$$\left(\ell_q\text{-}dist^{(\Pi)}(f)/2\right)^q - 1 \le \left(REM_q^{(\Pi)}(f)\right)^q \le \left(\ell_q\text{-}dist^{(\Pi)}(f)\right)^q - 1.$$

*Proof.*

$$\left(REM_q^{(\Pi)}(f)\right)^q + 1 = \left(\ell_q\text{-}dist_{(1)}^{(\Pi)}(f)\right)^q + 1 = E_\Pi\left[(|dist_f(u,v) - 1|)^q + 1\right].$$

Recalling that $\ell_q\text{-}dist^{(\Pi)}(f) = E_\Pi[(dist_f(u,v))^q]$, and noting that $dist_f(u,v) \ge 1$ for all $u \ne v \in X$, the claim follows from the inequality: $x^q + 1 \le (x+1)^q \le 2^q(x^q + 1)$ for $x \ge 0$, applied for $x = dist_f(u,v) - 1$. $\qquad\square$

The following two claims follow simply by definitions.

**Claim 20.** *For all $q \ge 1$ it holds that $Energy_q^{(\Pi)}(f) \le REM_q^{(\Pi)}(f)$.*

**Claim 21.** *For all $q \ge 1$: $Stress^{*(\Pi)}_q(f) = Stress_q^{(\Pi)}(f)/\ell_q\text{-}expans_q^{(\Pi')}(f) = Energy_q^{(\Pi')}(f)/\ell_q\text{-}expans_q^{(\Pi')}(f)$, where $\Pi'$ defined in Claim 15.*

In the next claim we show that under some conditions, the value of $\sigma$-distortion is bounded by the value of $Energy$:

**Claim 22.** *Let $f : X \to Y$ be any embedding, and let $q, r \ge 1$. If $\ell_r\text{-}contr(f) \le 1 + \alpha$, and $\ell_r\text{-}expans(f) \le 1 + \alpha$, for some $\alpha \ge 0$, then, for any distribution $\Pi$ over $\binom{X}{2}$:*

$$\Phi_{\sigma,q,r}^{(\Pi)}(f) \le 2(1+\alpha) \cdot Energy_q^{(\Pi)}(f) + 2\alpha.$$

*Proof.* By Jensen's inequality:

$$\frac{1}{\ell_r\text{-}expans(f)} \le \ell_r\text{-}contr(f) \le 1 + \alpha.$$

Therefore, $\frac{1}{1+\alpha} \le \ell_r\text{-}expans(f) \le 1 + \alpha$, Thus, by the definition of $\sigma$-distortion:

$$(\Phi_{\sigma,q,r}^{(\Pi)}(f))^q = \sum_{u \ne v \in X} \Pi(u,v) \left|\frac{expans_f(u,v)}{\ell_r\text{-}expans(f)} - 1\right|^q.$$

We consider two cases. If for a pair $u \ne v$, $\frac{expans_f(u,v)}{\ell_r\text{-}expans(f)} - 1 \ge 0$, then

$$\frac{expans_f(u,v)}{\ell_r\text{-}expans(f)} - 1 \le expans_f(u,v)(1+\alpha) - 1 \le |expans_f(u,v) - 1| + \alpha|expans_f(u,v)|$$

$$\leq (1+\alpha)|expans_f(u,v) - 1| + \alpha.$$

If for a pair $u \neq v$, $\frac{expans_f(u,v)}{\ell_r\text{-}expans(f)} - 1 < 0$, then

$$1 - \frac{expans_f(u,v)}{\ell_r\text{-}expans(f)} \leq 1 - \frac{expans_f(u,v)}{1+\alpha} \leq 1 - (1-\alpha)expans_f(u,v)$$

$$\leq |expans_f(u,v) - 1| + \alpha|expans_f(u,v)| \leq (1+\alpha)|expans_f(u,v) - 1| + \alpha,$$

implying that

$$(\overset{(\Pi)}{\Phi_{\sigma,q,r}}(f))^q \leq 2^q(1+\alpha)^q \sum_{u \neq v \in X} \Pi(u,v) \, |expans_f(u,v) - 1|^q + (2\alpha)^q,$$

which completes the proof. $\qquad\square$

# D  Advanced properties of distortion measures

We investigate key properties of the moments of distortion.

## D.1  Lower/upper bound relations

We show the quantitative connections between the $\ell_q$-distortion of a non-contractive (or non-expansive) embedding and the $\ell_q$-distortion of a general type embedding. Particularly, we investigate the upper and lower bounds on $\ell_q$-distortion of a general type embedding that can be derived from corresponding bounds on a non-contractive or non-expansive embedding.

**Claim 23.** *Let $(X, d_X)$ and $(Y, d_Y)$ be finite metric spaces, and $f : X \to Y$ be an embedding. Then, for any distribution $\Pi$ over $\binom{X}{2}$ the following statements hold:*

1. *$\forall q \geq 1 : \quad \ell_q\text{-}dist^{(\Pi)}(f) \leq \left((\ell_q\text{-}expans^{(\Pi)}(f))^q + (\ell_q\text{-}contr^{(\Pi)}(f))^q\right)^{1/q} \leq \ell_q\text{-}expans^{(\Pi)}(f) + \ell_q\text{-}contr^{(\Pi)}(f)$.*

2. *If $Y$ is scalable, then $\forall q \geq 1$ there is $\hat{f} : X \to Y$ with $\ell_q\text{-}dist^{(\Pi)}(\hat{f}) \leq 2^{1/q}\sqrt{\ell_q\text{-}expans^{(\Pi)}(f)} \cdot \sqrt{\ell_q\text{-}contr^{(\Pi)}(f)}$.*

*Proof.* The first statement of the claim follows directly from the definition of the $\ell_q$-distortion. For the second statement, let $\alpha > 0$ be a normalization parameter (will be chosen later). Define an embedding $\hat{f}$ by $\hat{f} = \alpha \cdot f$. Denote

$$A = \{(x,y) \mid x \neq y \in X, \ \alpha \cdot d_Y(f(x), f(y)) \leq d_X(x,y)\},$$

and

$$B = \{(x,y) \mid x \neq y \in X, \ \alpha \cdot d_Y(f(x), f(y)) \geq d_X(x,y)\}.$$

We have

$$(\ell_q\text{-}dist^{(\Pi)}(\hat{f}))^q = \sum_{(x,y) \in A} \Pi(x,y) \left(\frac{1}{\alpha} \cdot \frac{d_X(x,y)}{d_Y(f(x), f(y))}\right)^q + \sum_{(x,y) \in B} \Pi(x,y) \left(\alpha \cdot \frac{d_Y(f(x), f(y))}{d_X(x,y)}\right)^q$$

$$\leq \frac{1}{\alpha^q} \cdot (\ell_q\text{-}contr^{(\Pi)}(f))^q + \alpha^q \cdot (\ell_q\text{-}expans^{(\Pi)}(f))^q.$$

Choosing the normalization factor $\alpha = \sqrt{\dfrac{\ell_q\text{-}contr^{(\Pi)}(f)}{\ell_q\text{-}expans^{(\Pi)}(f)}}$ implies the claim. $\qquad\square$

**Corollary 2.** *Let $(X, d_X)$ be any finite metric space, $(Y, d_Y)$ be any scalable metric space. Let $\Pi$ be any distribution over $\binom{X}{2}$, and $q \geq 1$. If there exists a non-contractive (non-expansive embedding) $f : X \to Y$ with $\ell_q\text{-}dist^{(\Pi)}(f) \leq D(q)$, then there exists an embedding $\hat{f} : X \to Y$ with $\ell_q\text{-}dist^{(\Pi)}(\hat{f}) \leq 2^{1/q}\sqrt{D(q)}$.*

For the lower bounds we have the following result. For a metric space $X$, denote the aspect ratio of $X$ by $\Phi_X = d_{max}/d_{min}$, where $d_{max} = \max_{x \neq y \in X} d_X(x, y)$, and $d_{min} = \min_{x \neq y \in X} d_X(x, y)$.

**Claim 24.** *Let $(X, d_X)$ be an $n$-point metric space and let $(Y, d_Y)$ be a scalable metric space, and let $q \geq 1$. For all $1 \leq s \leq n$, let $D_q(s)$ be such that for every subset $S \subseteq X$, of size $s$, for any non-expansive embedding $f : S \to Y$ it holds that $\ell_q\text{-}dist_S(f) \geq D_q(s)$. Then for any embedding $g : X \to Y$ it holds that $\ell_q\text{-}dist(g) = \Omega\left(\max\left\{\sqrt{\frac{D_q(n/3)}{\Phi_X}}, \frac{D_q(n/3)}{\Phi_X \cdot \ell_1\text{-}dist(g)}\right\}\right)$.*

*Proof.* We start with the first item. Let $(X, d_X)$ be an $n$-point metric space as in the claim. Let $g : X \to Y$ be any embedding, and let $q \geq 1$. Consider the set of images of $X$ under $g$, i.e. $G = \{g(x)|x \in X\} \subseteq Y$, and consider a radius $R \geq 1$ (will be chosen later). We have the following two cases.

1.  If there exists a point $y \in G$ s.t. the ball $B_G(y, R)$ contains at least $n/3$ points, then consider the pre-image set of these points, $X' \subseteq X$ - a metric space on at least $n/3$ points. The embedding $g$ on $X'$ has expansion $E \leq \frac{2R}{d_{min}}$. Note that if $E \leq 1$, then we get the required lower bound on $\ell_q\text{-}dist(g)$, since $\ell_q\text{-}dist_X(g) = \Omega\left(\ell_q\text{-}dist_{X'}(g)\right) = \Omega(D_q(n/3))$ (we get even better lower bound in this case).

    Otherwise, assume that $E \geq 1$, and consider the embedding $\hat{g} : X' \to Y$ defined by $\hat{g}(x) = \frac{g(x)}{E}$. This embedding is non-expansive, therefore $\ell_q\text{-}dist(\hat{g}) \geq D_q(n/3)$. On the other hand we have

    $$\ell_q\text{-}dist_X(g) = \Omega\left(\ell_q\text{-}dist_{X'}(g)\right) = \Omega\left(\frac{\ell_q\text{-}dist(\hat{g})}{E}\right) = \Omega\left(\frac{D_q(n/3)}{E}\right) = \Omega\left(\frac{D_q(n/3) \cdot d_{min}}{R}\right). \tag{5}$$

2.  Otherwise, for all point $z \in G$ the ball $B_G(z, R)$ contains less that $n/3$ points, meaning there are at least $\frac{n^2}{3}$ pairs $(x, y)$ of $X$ such that

    $$dist_g(x, y) \geq \frac{d_Y(g(x), g(y))}{d_X(x, y)} \geq \frac{R}{d_{max}}.$$

    Namely, $\ell_q\text{-}dist(g) = \Omega(\frac{R}{d_{max}})$. Choosing $R = \sqrt{D_q(n/3)d_{max}d_{min}}$ implies the claim.

Next, we prove the second item.

Denote $\ell_1\text{-}dist(g) = \alpha$. First, note that there exists a point $z \in X$ such that there are at least $n/3$ pairs $(z, x) \in \binom{X}{2}$ with $dist_g(z, x) \leq 3\alpha$. Since otherwise, we will get $\ell_1\text{-}dist(g) > \alpha$. Denote the set of points of these pairs by $S \subseteq X$, $|S| \geq n/3$. For any $x \neq y \in S$ it holds that

$$expans_g(x, y) = \frac{d_Y(g(x), g(y))}{d_X(x, y)} \leq \frac{d_Y(g(x), g(z))}{d_X(x, y)} + \frac{d_Y(g(z), g(y))}{d_X(x, y)} \leq 3\alpha\left(\frac{d_X(x, z)}{d_X(x, y)} + \frac{d_X(z, y)}{d_X(x, y)}\right) \leq 6\alpha\Phi_X.$$

Thus, the embedding $g$ has expansion at most $6\alpha\Phi_X$ on the set $S$. Therefore, defining an embedding $\hat{g} : S \to Y$ by $\hat{g} = \frac{g}{\alpha\Phi_X}$, we obtain a non-expansive embedding on $S$. And thus, applying the same considerations as in Equation 5, we obtain the stated bound. $\square$

### D.1.1 Lower bound due to the worst case barrier

We show that the worst case distortion poses a barrier to the moments of distortions.

**Theorem 25.** *Let $(X, d_X)$ be a finite metric space, and $(Y, d_Y)$ be an arbitrary metric space. Let $\alpha > 1$ such that for any embedding $f : X \to Y$ it holds that $dist(f) \geq \alpha$. Let $n_X = |X|$, then there exists $\alpha' > 1$, such that for all $N \geq n_X$ there exists a metric space $Z$, $|Z| = N$ such that the following holds*

$$\forall F : Z \to Y, \ \ell_q\text{-}dist(F) \geq \alpha',$$

*where $\alpha' = \left(1 + \frac{2\left(\alpha^{q/2}-1\right)}{n_X{}^2}\right)^{1/q}$, for a given $q \geq 1$.*

Let us first introduce the notion of composition of metric spaces, that was defined in (14) (we present here a simplification of the original definition).

**Definition 7.** *Let $(S, d_S)$, $(T, d_T)$ be finite metric spaces. The composition of $S$ with $T$, denoted by $Z = S[T]$, is a metric space of size $|Z| = |S| \cdot |T|$ constructed as follows. Each point $u \in S$ is substituted with a copy of the metric space $T$, denote this copy $T^{(u)}$. The distances are defined as follows. Let $u, v \in S$, and $z_i \neq z_j \in Z$, such that $z_i \in T^{(u)}$, and $z_j \in T^{(v)}$, then*

$$d_Z(z_i, z_j) = \begin{cases} \frac{1}{\gamma} \cdot d_T(z_i, z_j) & u = v \\ d_S(u, v) & u \neq v, \end{cases}$$

*where $\gamma = \frac{\max_{t \neq t' \in T}\{d_T(t, t')\}}{\min_{s \neq s' \in S}\{d_S(s, s')\}}$.*

It is easily checked that the choice of the factor $1/\gamma$ guarantees that $d_Z$ is indeed a metric.

*Proof of Theorem 25.* Given any $N \geq n_X$ let $m = N/n_X \geq 1$, and let $T$ be any $m$-point metric space. Define the metric space $Z$ to be the metric composition of $X$ with $T$. (We note that the choice of $T$ is arbitrary, which allows us to apply the theorem for specific families of spaces that are closed under metric composition).

Let $F : Z \to Y$ be any embedding, and let $B \subseteq \binom{Z}{2}$, $B = \{(z_i, z_j)|z_i \in T^{(u)}, z_j \in T^{(v)}, \forall u \neq v \in X\}$. Then, $|B| = m^2 \cdot \binom{n_X}{2}$. We have to lower bound the $\ell_q\text{-}dist(F)$, for any given $q$. Let $q \geq 1$, and note that by definition for any $z_i \neq z_j \in Z$ it holds that $dist_F(z_i, z_j) \geq 1$. Then it holds that

$$(\ell_q\text{-}dist(F))^q \geq \frac{1}{\binom{N}{2}} \sum_{z_i \neq z_j \in B} (dist_F(z_i, z_j))^q + \frac{\binom{N}{2} - \binom{n_X}{2}m^2}{\binom{N}{2}} = \tag{6}$$

$$1 + \frac{\binom{n_X}{2}m^2}{\binom{N}{2}} \left( \frac{1}{\binom{n_X}{2}m^2} \cdot \sum_{z_i \neq z_j \in B} (dist_F(z_i, z_j))^q - 1 \right)$$

Next we estimate the $\ell_q$-distortion of the embedding $F$ induced on the set of pairs $B \subseteq \binom{Z}{2}$. Denote by $\mathcal{P}$ the family of all possible $n_X$-point subsets $P^{(t)} \subset Z$, $P^{(t)} = \{p_u^{(t)}|p_u^{(t)} \in T^{(u)}\}_{u \in X}$. Then $|\mathcal{P}| = m^{n_X}$, and it holds that

$$\frac{1}{m^{n_X}} \sum_{P^{(t)} \in \mathcal{P}} (\ell_q\text{-}dist_{P^{(t)}}(F))^q = \frac{1}{m^{n_X}} \sum_{P^{(t)} \in \mathcal{P}} \frac{1}{\binom{n_X}{2}} \sum_{p_u^{(t)}, p_v^{(t)} \in P^{(t)}} (dist_F(p_u^{(t)}, p_v^{(t)}))^q =$$

$$\frac{1}{m^{n_X}} \frac{1}{\binom{n_X}{2}} \sum_{z_i \neq z_j \in B} m^{n_X - 2} \cdot (dist_F(z_i, z_j))^q = \frac{1}{\binom{n_X}{2}m^2} \cdot \sum_{z_i \neq z_j \in B} (dist_F(z_i, z_j))^q.$$

Now, for all $t$ the metric space $(P^{(t)}, d_Z)$ is an isometry of $X$. Recall that by assumption, any embedding of $X$ into $Y$ has distortion at least $\alpha$, implying $\ell_\infty$-distortion at least $\sqrt{\alpha}$, therefore there exists a pair of points in $X$ with distortion at least $\sqrt{\alpha}$. Therefore,

$$(\ell_q\text{-}dist_{P^{(t)}}(F))^q \geq 1 + \frac{\alpha^{q/2} - 1}{\binom{n_X}{2}}.$$

Therefore,

$$\frac{1}{\binom{n_X}{2}m^2} \cdot \sum_{z_i \neq z_j \in B} (dist_F(z_i, z_j))^q \geq 1 + \frac{\alpha^{q/2} - 1}{\binom{n_X}{2}}.$$

Putting it in (6) implies

$$(\ell_q\text{-}dist(F))^q \geq 1 + \frac{m^2}{\binom{N}{2}} \left(\alpha^{q/2} - 1\right) \geq 1 + \frac{2\left(\alpha^{q/2} - 1\right)}{n_X{}^2}.$$

$\square$

## D.2  Analysis if distortion measures under embedding composition

We study the behavior of the measures we have defined under composition of embeddings.

**Moments of distortion measure.** Applying Hölder's inequality we obtain basic results on $\ell_q$-expansion and $\ell_q$-contraction of the composition of two embeddings.

**Claim 26.** *Let $(X, d_X)$, $(Y, d_Y)$, and $(Z, d_Z)$ be $n$-point metric spaces. Let $f : X \to Y$, and $g : Y \to Z$ be embeddings, and let $\Pi$ be a distribution over $\binom{n}{2}$. Let $q \geq 1$, $s, t \geq 1$ such that $1/s + 1/t = 1$. Then*

$$\ell_q\text{-}expans^{(\Pi)}(g \circ f) \leq \ell_{q \cdot s}\text{-}expans^{(\Pi)}(f) \cdot \ell_{q \cdot t}\text{-}expans^{(\Pi)}(g),$$

*and*

$$\ell_q\text{-}contr^{(\Pi)}(g \circ f) \leq \ell_{q \cdot s}\text{-}contr^{(\Pi)}(f) \cdot \ell_{q \cdot t}\text{-}contr^{(\Pi)}(g).$$

*Proof.* For all $x_i \neq x_j \in X$, let $\alpha_{ij} = d_X(x_i, x_j)$, $\beta_{i,j} = d_Y(f(x_i), f(x_j))$, and $\gamma_{i,j} = d_Z(g(f(x_i)), g(f(x_j)))$. Then we have

$$(\ell_q\text{-}expans^{(\Pi)}(g \circ f))^q = \sum_{i \neq j} \Pi(i, j)\left(\frac{\gamma_{ij}}{\alpha_{ij}}\right)^q = \sum_{i \neq j} (\Pi(i, j))^{1/s}\left(\frac{\gamma_{ij}}{\beta_{ij}}\right)^q \cdot (\Pi(i, j))^{1/t}\left(\frac{\beta_{ij}}{\alpha_{ij}}\right)^q.$$
(7)

Applying Holder's inequality we get that (7) is at most

$$\left(\sum_{i \neq j} \Pi(i, j)\left(\frac{\gamma_{ij}}{\beta_{ij}}\right)^{sq}\right)^{1/s} \cdot \left(\sum_{i \neq j} \Pi(i, j)\left(\frac{\beta_{ij}}{\alpha_{ij}}\right)^{tq}\right)^{1/t},$$

implying the first part of the claim. For $\ell_q\text{-}contr^{(\Pi)}(g \circ f)$, exactly the same considerations imply the second part of the claim. $\square$

The following claim follows by applying Claim 23 to the bounds in Claim 26.

**Claim 27.** *Let $(X, d_X)$, $(Y, d_Y)$, and $(Z, d_Z)$ be $n$-point metric spaces. Let $f : X \to Y$, and $g : Y \to Z$ be embeddings, and let $\Pi$ be a distribution over $\binom{n}{2}$. Let $s_1, t_1, s_2, t_2 \geq 1$, such that $1/s_1 + 1/t_1 = 1$, and $1/s_2 + 1/t_2 = 1$. Then the following statements hold.*

1. *For all $q \geq 1$:*

$$\ell_q\text{-}dist^{(\Pi)}(g \circ f) \leq ((\ell_{(q \cdot s_1)}\text{-}expans^{(\Pi)}(f) \cdot \ell_{(q \cdot t_1)}\text{-}expans^{(\Pi)}(g))^q + (\ell_{(q \cdot s_2)}\text{-}contr^{(\Pi)}(f) \cdot \ell_{(q \cdot t_2)}\text{-}contr^{(\Pi)}(g))^q)^{1/q}$$

$$\leq \ell_{(q \cdot s_1)}\text{-}expans^{(\Pi)}(f) \cdot \ell_{(q \cdot t_1)}\text{-}expans^{(\Pi)}(g) + \ell_{(q \cdot s_2)}\text{-}contr^{(\Pi)}(f) \cdot \ell_{(q \cdot t_2)}\text{-}contr^{(\Pi)}(g).$$

2. *Assume $Z$ is scalable. For any $q \geq 1$, there exists an embedding $h : X \to Z$ such that:*

$$\ell_q\text{-}dist^{(\Pi)}(h) \leq 2^{1/q} \cdot \sqrt{\ell_{(q \cdot s_1)}\text{-}expans^{(\Pi)}(f) \cdot \ell_{(q \cdot t_1)}\text{-}expans^{(\Pi)}(g)} \cdot \sqrt{\ell_{(q \cdot s_2)}\text{-}contr^{(\Pi)}(f) \cdot \ell_{(q \cdot t_2)}\text{-}contr^{(\Pi)}(g)}.$$

We conclude the following corollary.

**Corollary 3.** *Let $(X, d_X)$, $(Y, d_Y)$, $(Z, d_Z)$ be $n$-point metric spaces, and let $f : X \to Y$ be a non-contractive (or even if $\ell_q\text{-}contr^{(\Pi)}(f) \leq 1$) embedding, and $g : Y \to Z$ be a non-expansive (or even if $\ell_q\text{-}expans^{(\Pi)}(g) \leq 1$) embedding. Then, for any distribution $\Pi$ over $\binom{n}{2}$, the following statements hold:*

1. *For all $q \geq 1$: $\ell_q\text{-}dist^{(\Pi)}(g \circ f) \leq \left( \left( \ell_q\text{-}dist^{(\Pi)}(f) \right)^q + \left( \ell_q\text{-}dist^{(\Pi)}(g) \right)^q \right)^{1/q} \leq \ell_q\text{-}dist^{(\Pi)}(f) + \ell_q\text{-}dist^{(\Pi)}(g).$*

2. *If $Z$ is scalable, then $\forall q \geq 1$ there exists $h : X \to Z$ such that: $\ell_q\text{-}dist^{(\Pi)}(h) \leq 2^{1/q} \cdot \sqrt{\ell_q\text{-}dist^{(\Pi)}(f)} \cdot \sqrt{\ell_q\text{-}dist^{(\Pi)}(g)}.$*

*The same holds also if we exchange the assumptions between $f$ and $g$.*

*Proof.* In Claim 27, for $f$ having $\ell_q\text{-}contr^{(\Pi)}(f) \leq 1$ (in particular if it's non-contractive) and $g$ having $\ell_q\text{-}expans^{(\Pi)}(g) \leq 1$ (in particular if it's non-expansive) take $s_1 = 1$, $t_1 = \infty$, and $s_2 = \infty$, $t_2 = 1$; if $f$ and $g$ obey the reverse conditions take $s_1 = \infty$, $t_1 = 1$, and $s_2 = 1$, $t_2 = \infty$. $\square$

**Claim 28.** *Let $(X, d_X)$, $(Y, d_Y)$, and $(Z, d_Z)$ be $n$-point metric spaces. Let $f : X \to Y$, $g : Y \to Z$, and let $\Pi$ be a distribution over $\binom{n}{2}$. Let $q \geq 1$, then for $s, t \geq 1$ such that $1/s + 1/t = 1$ it holds that $\ell_q\text{-}dist^{(\Pi)}(g \circ f) \leq \ell_{q \cdot s}\text{-}dist^{(\Pi)}(f) \cdot \ell_{q \cdot t}\text{-}dist^{(\Pi)}(g).$*

*Proof.* For all $x_i \neq x_j \in X$, let $\alpha_{ij} = d_X(x_i, x_j)$, $\beta_{i,j} = d_Y(f(x_i), f(x_j))$, and $\gamma_{i,j} = d_Z(g(f(x_i)), g(f(x_j)))$. Then we have

$$\left( \ell_q\text{-}dist^{(\Pi)}(g \circ f) \right)^q = \sum_{1 \leq i < j \leq n} \Pi(i, j) \max \left\{ \left( \frac{\gamma_{ij}}{\beta_{ij}} \cdot \frac{\beta_{ij}}{\alpha_{ij}} \right)^q, \left( \frac{\alpha_{ij}}{\beta_{ij}} \cdot \frac{\beta_{ij}}{\gamma_{ij}} \right)^q \right\}$$

$$\leq \sum_{1 \leq i < j \leq n} \Pi(ij)(dist_f(i, j))^q \cdot (dist_g(i, j))^q.$$

The claim follows by Holder's inequality. $\square$

We also show that in a special case of embedding $g : Y \to Z$ being a randomized embedding with a particular property, the upper bound can be improved. We will later apply this claim on composition of a deterministic embedding $f$ with the randomized JL transform (which will be shown to satisfy this property).

**Claim 29.** *Let $(X, d_X)$ be an $n$-point metric space, $(Y, d_Y)$ and $(Z, d_Z)$ any metric spaces, $k \geq 1$ be any integer, $q \geq 1$ and $\Pi$ be any distribution over $\binom{n}{2}$. Let $f : X \to Y$ be any deterministic embedding, and let $g : Y \to Z$ be a randomized embedding such that for any pair $(y_i, y_j) \in Y$ it holds that $E\left[ (dist_g(y_i, y_j))^q \right] = D$.*

$$E\left[ (\ell_q\text{-}dist^{(\Pi)}(g \circ f)) \right] \leq \ell_q\text{-}dist^{(\Pi)}(f) \cdot \left( E\left[ \left( \ell_q\text{-}dist^{(\Pi)}(g) \right)^q \right] \right)^{\frac{1}{q}}.$$

*The same inequality holds for $\ell_q\text{-}expans^{(\Pi)}$ and $\ell_q\text{-}contr^{(\Pi)}$ measures.*

*Proof.* For all $1 \leq i < j \leq n$ denote $\alpha_{ij} = d(x_i, x_j)$, $\beta_{i,j} = d_Y(f(x_i), f(x_j))$, and $\gamma_{ij} = d_Z(g(f(x_i)), g(f(x_j)))$. Then we have

$$E\left[\left(\ell_q\text{-}dist^{(\Pi)}(g \circ f)\right)^q\right] = E\left[\sum_{1 \leq i < j \leq n} \Pi(i,j) \max\left\{\left(\frac{\gamma_{ij}}{\alpha_{ij}}\right)^q, \left(\frac{\alpha_{ij}}{\gamma_{ij}}\right)^q\right\}\right]$$

$$\leq E\left[\sum_{1 \leq i < j \leq n} \Pi(i,j)(dist_f(i,j))^q \cdot (dist_g(i,j))^q\right]$$

$$= \sum_{1 \leq i < j \leq n} \Pi(i,j)(dist_f(i,j))^q E\left[(dist_g(i,j))^q\right] = \left(\ell_q\text{-}dist^{(\Pi)}(f)\right)^q E\left[\left(\ell_q\text{-}dist^{(\Pi)}(g)\right)^q\right],$$

where the lest inequality follows by the linearity of expectation. Therefore, by Jensen's inequality we obtain the required. $\square$

**Fact D.1.** *Let* $(X, d_X)$, $(Y, d_Y)$, *and* $(Z, d_Z)$ *be any n-point metric spaces. Let* $f : X \rightarrow Y$, *and* $g : Y \rightarrow Z$ *be any embeddings. Then for any distribution* $\Pi$ *over* $\binom{n}{2}$, *and for all* $q \geq 1$ *it holds that* $distnorm_q^{(\Pi)}(g \circ f) \leq distnorm_q^{(\Pi)}(f) \cdot distnorm_q^{(\Pi)}(g)$.

**Additive distortion measures.** We study the behavior of the additive distortion measures under composition of embeddings. Particularly, we investigate a special case of composition of any deterministic embedding with a randomized JL transform.

**Claim 30.** *Let* $(X, d_X)$ *be an n-point metric space,* $(Y, d_Y)$ *and* $(Z, d_Z)$ *any metric spaces,* $k \geq 1$ *an integer,* $q \geq 1$ *and* $\Pi$ *be any distribution over* $\binom{n}{2}$. *Let* $f : X \rightarrow Y$ *be any deterministic embedding, and let* $g : Y \rightarrow Z$ *be a randomized embedding such that for every pair* $(y_i, y_j)$ *it holds that* $E\left[|expans_g(y_i, y_j) - 1|^q\right] = A$, *and* $E\left[|contr_g(y_i, y_j) - 1|^q\right] = B$. *Then*

$$E\left[\left(REM_q^{(\Pi)}(g \circ f)\right)\right] \leq 4\left(1 + \left(E\left[\left(REM_q^{(\Pi)}(g)\right)^q\right]\right)^{\frac{1}{q}}\right)\left(REM_q^{(\Pi)}(f)\right) + 4\left(E\left[\left(REM_q^{(\Pi)}(g)\right)^q\right]\right)^{\frac{1}{q}}.$$

*Proof.* For all $1 \leq i < j \leq n$ denote $\alpha_{ij} = d(x_i, x_j)$, $\beta_{i,j} = d_Y(f(x_i), f(x_j))$, and $\gamma_{ij} = d_Z(g(f(x_i)), g(f(x_j)))$. Then, using the inequality $|x + y|^q \leq 2^{q-1}(|x|^q + |y|^q)$, we get

$$\left(REM_q^{(\Pi)}(g \circ f)\right)^q = \sum_{1 \leq i < j \leq n} \Pi(i,j) \cdot \max\left\{\left|\frac{\gamma_{ij} - \alpha_{ij}}{\alpha_{ij}}\right|^q, \left|\frac{\gamma_{ij} - \alpha_{ij}}{\gamma_{ij}}\right|^q\right\}$$

$$\leq 2^{q-1} \sum_{1 \leq i < j \leq n} \Pi(i,j) \cdot \max\left\{\left|\frac{\gamma_{ij} - \beta_{ij}}{\alpha_{ij}}\right|^q + \left|\frac{\beta_{ij} - \alpha_{ij}}{\alpha_{ij}}\right|^q, \left|\frac{\gamma_{ij} - \beta_{ij}}{\gamma_{ij}}\right|^q + \left|\frac{\beta_{ij} - \alpha_{ij}}{\gamma_{ij}}\right|^q\right\}$$

$$= 2^{q-1} \sum_{1 \leq i < j \leq n} \Pi(i,j) \cdot \max\left\{\left|\frac{\gamma_{ij} - \beta_{ij}}{\beta_{ij}}\right|^q \cdot \left(\frac{\beta_{ij}}{\alpha_{ij}}\right)^q + \left|\frac{\beta_{ij} - \alpha_{ij}}{\alpha_{ij}}\right|^q, \left|\frac{\beta_{ij} - \gamma_{ij}}{\gamma_{ij}}\right|^q + \left|\frac{\alpha_{ij} - \beta_{ij}}{\beta_{ij}}\right|^q \cdot \left(\frac{\beta_{ij}}{\gamma_{ij}}\right)^q\right\}.$$

Using the inequality $\max\{a, b\} \leq a + b$, for all $a, b \geq 0$, and the assumption $E\left[|expans_g(y_i, y_j) - 1|^q\right] = A = E\left[\left(\ell_q\text{-}expans_{(1)}^{(\Pi)}(g)\right)^q\right]$, and $E\left[|contr_g(y_i, y_j) - 1|^q\right] = B = E\left[\left(\ell_q\text{-}contr_{(1)}^{(\Pi)}(g)\right)^q\right]$, together with linearity of expectation, we obtain that

$$E\left[\left(REM^{(\Pi)}(g \circ f)\right)^q\right] \leq 2^{q-1} E\left[\left(\ell_q\text{-}expans_{(1)}^{(\Pi)}(g)\right)^q\right] \cdot \sum_{1 \leq i < j \leq n} \Pi(i,j)\left(\frac{\beta_{ij}}{\alpha_{ij}}\right)^q + 2^{q-1}\left(\ell_q\text{-}expans_{(1)}^{(\Pi)}(f)\right)^q$$

$$+ 2^{q-1} E\left[\left(\ell_q\text{-}contr_{(1)}^{(\Pi)}(g)\right)^q\right] + 2^{q-1}\left(\ell_q\text{-}contr_{(1)}^{(\Pi)}(f)\right)^q \cdot \sum_{1 \leq i < j \leq n} \Pi(i,j) E\left[\left(\frac{\beta_{ij}}{\gamma_{ij}}\right)^q\right].$$

Presenting $\frac{\beta_{ij}}{\alpha_{ij}} = 1 + \left(\frac{\beta_{ij}}{\alpha_{ij}} - 1\right)$, and $\frac{\beta_{ij}}{\gamma_{ij}} = 1 + \left(\frac{\beta_{ij}}{\gamma_{ij}} - 1\right)$, and using $|x + y|^q \leq 2^{q-1}(|x|^q + |y|^q)$ again we get

$$\sum_{1 \leq i < j \leq n} \Pi(i,j)\left(\frac{\beta_{ij}}{\alpha_{ij}}\right)^q \leq 2^{q-1}\left(1 + \left(\ell_q\text{-}expans_{(1)}^{(\Pi)}(f)\right)^q\right),$$

and

$$\sum_{1 \leq i < j \leq n} \Pi(i,j) E\left[\left(\frac{\beta_{ij}}{\gamma_{ij}}\right)^q\right] \leq 2^{q-1}\left(1 + E\left[(\ell_q\text{-}contr_{(1)}^{(\Pi)}(g))^q\right]\right),$$

putting it back into the inequalities, and observing that for any $h$: $\ell_q\text{-}contr_{(1)}^{(\Pi)}(h), \ell_q\text{-}expans_{(1)}^{(\Pi)}(h) \leq REM_q^{(\Pi)}(h)$, immediately results the stated bound. $\qquad\square$

**Claim 31.** *Let $(X, d_X)$ be an $n$-point metric space, $(Y, d_Y)$ and $(Z, d_Z)$ any metric spaces, $k \geq 1$ an integer, $q \geq 1$ and $\Pi$ be any distribution over $\binom{n}{2}$. Let $f : X \to Y$ be any deterministic embedding, and let $g : Y \to Z$ be a randomized embedding such that for every pair $(y_i, y_j)$ it holds that $E\left[|expans_g(y_i, y_j) - 1|^q\right] = A$, and $E\left[|contr_g(y_i, y_j) - 1|^q\right] = B$. Then*

$$E\left[Energy_q^{(\Pi)}(g \circ f)\right] \leq 4Energy_q^{(\Pi)}(f) \cdot E\left[\left(Energy_q^{(\Pi)}(g)\right)^q\right]^{\frac{1}{q}} + 4E\left[\left(Energy_q^{(\Pi)}(g)\right)^q\right]^{\frac{1}{q}}.$$

For $Stress_q$ and $Stress_q^*$ measures we obtain similar bounds on the composition of two general embeddings.

**Claim 32.** *Let $(X, d_X)$, $(Y, d_Y)$, and $(Z, d_Z)$ be any $n$-point metric spaces. Let $f : X \to Y$, and $g : Y \to Z$ be any embeddings. Then for any distribution $\Pi$ over $\binom{n}{2}$, and for all $q \geq 1$ it holds that*

$$Stress_q^{(\Pi)}(g \circ f) \leq 2Stress_q^{(\Pi)}(f) + 4Stress_q^{(\Pi)}(g) \cdot Stress_q^{(\Pi)}(f) + 4Stress_q^{(\Pi)}(g).$$

**Claim 33.** *Let $(X, d_X)$, $(Y, d_Y)$, and $(Z, d_Z)$ be any $n$-point metric spaces. Let $f : X \to Y$, and $g : Y \to Z$ be any embeddings. Then for any distribution $\Pi$ over $\binom{n}{2}$, and for all $q \geq 1$ it holds that*

$$Stress_q^{*(\Pi)}(g \circ f) \leq 4Stress_q^{*(\Pi)}(f) + 4Stress_q^{*(\Pi)}(f) \cdot Stress_q^{*(\Pi)}(g) + 2Stress_q^{*(\Pi)}(g).$$

## E  On the advantage of being both contractive and expansive

In this section we show that in order to achieve the best $\ell_q$-distortion we must allow embeddings that are both contractive and expansive (for different pairs of points). The first example is a family of constant degree expander graphs.

**Theorem 34.** *Let $(G, d_G)$ be the metric of a constant degree expander graph $G$ (of constant expansion) on $n$ vertices. Let $k \geq 2$ and let $f : G \to \ell_2^k$ be an embedding. If $f$ is non-contractive, then $\ell_1\text{-}dist(f) = \Omega(n^{1/k}/\log n)$, and if $f$ is non-expansive, then $\ell_1\text{-}dist(f) = \Omega(\log n)$. Yet, there exists $f$ with $\ell_1\text{-}dist(f) = O(1)$.*

The proof immediately follows from the following lemmas and claims.

**Lemma 35.** *Let $E_n$ be an $n$-point equilateral metric space. For all $k \geq 1$ and $q \geq 1$, any non-contractive embedding $f : E_n \to \ell_2^k$ has $\ell_q\text{-}dist(f) = \Omega\left(n^{1/k}\right)$.*

*Proof.* Since $\ell_q$-distortion is increasing function of $q$, it is enough to show the lower bound on $\ell_1$-distortion. The proof is based on the well known lower bound, that states that there exists a universal constant $C \geq 1$ (independent of $n$ and $k$), such that any embedding $f : E_n \to \ell_2^k$ has distortion $dist(f) \geq Cn^{1/k}$.

Let $f : E_n \to \ell_2^k$ be a non-contractive embedding. Assume by contradiction that $\ell_1\text{-}dist(f) < \frac{C}{132} \cdot \left(\frac{3}{4}\right)^{1/k} n^{1/k}$. Denote $B \subset X$ the set of points with 'many bad' neighbors:

$$B = \left\{x \in E_n \,\middle|\, \exists\, Y \subset E_n, |Y| > \frac{n}{4}, s.t. \ \forall y \in Y, dist_f(x, y) > \frac{C}{4} \cdot \left(\frac{3}{4}\right)^{1/k} n^{1/k}\right\}.$$

Then, $|B| \leq \frac{n}{4}$, since otherwise we have $\ell_1\text{-}dist(f) > \frac{C}{128} \cdot \left(\frac{3}{4}\right)^{1/k} n^{1/k}$, a contradiction.

Consider the equilateral metric space $E' = E_n \setminus B$, $|E'| \geq \frac{3}{4}n$. For all $x \in E'$ there are at least $n/2$ points $y \in E'$ such that $dist_f(x,y) \leq \frac{C}{4} \cdot \left(\frac{3}{4}\right)^{1/k} n^{1/k}$, implying that for any pair $x \neq y \in E'$ there exists $z \in E'$, $z \neq x, z \neq y$, such that $dist_f(x,z) \leq \frac{C}{4} \cdot \left(\frac{3}{4}\right)^{1/k} n^{1/k}$ and $dist_f(y,z) \leq \frac{C}{4} \cdot \left(\frac{3}{4}\right)^{1/k} n^{1/k}$. Therefore, since $f$ is non-contractive, it holds that for any two points $x \neq y \in E'$, $dist_f(x,y) \leq \frac{C}{2} \cdot \left(\frac{3}{4}\right)^{1/k} n^{1/k}$. Therefore, we have an embedding $f : E' \to \ell_2^k$ with $dist(f) \leq \frac{C}{2} \cdot \left(\frac{3}{4}\right)^{1/k} n^{1/k}$. Since $|E'| \geq \frac{3}{4}n$ it holds that any embedding of $E'$ into $\ell_2^k$ should have distortion at least $C \cdot \left(\frac{3}{4}\right)^{1/k} n^{1/k}$, contradiction. $\square$

**Remark 36.** *The above claim is true for embedding into any $k$-dimensional normed space, since the lower bound we used is true for any $k$-dimensional normed space.*

Let us conclude that embedding an expander into $\ell_2^k$ via non-contractive embedding results in a bad average distortion.

**Claim 37.** *Let $(G, d_G)$ be a constant degree expander graph $G$ (of constant expansion) on $n$ vertices, for $n$ big enough, where $d_G$ is a shortest path metric. Then any non-contractive embedding $f : G \to \ell_2^k$ has $\ell_1\text{-}dist(f) = \Omega(n^{1/k}/\log n)$, for any given $k \geq 1$.*

*Proof.* First, note that $diam(G) = O(\log n)$. Assume by contradiction that there is a non-contractive $f : G \to \ell_2^k$ with $\ell_1\text{-}dist(f) = o(n^{1/k}/\log n)$. Then, we can build a non-contractive embedding of an $n$-point equilateral space $g : E_n \to \ell_2^k$ in the following way: First, embed $E_n$ with a non-contractive embedding into $G$ with (worst case) distortion of $O(\log n)$; Next, embed $G$ with $f$ into $\ell_2^k$. The $\ell_1$-distortion of such embedding is $o\left(n^{1/k}\right)$, which is a contradiction to Lemma 35. $\square$

**Remark 38.** *The above claim is true for embedding any metric space with bounded aspect ratio into any $k$-dimensional normed space (replacing $\log n$ by the aspect ratio).*

Next, we prove that embedding an expander into $\ell_2$ with a non-expansive embedding also results in a bad average distortion. Actually, we present a more general result of non-embedability into $\ell_p$, for any $p \geq 1$.

**Claim 39.** *Let $(G, d_G)$ be a constant degree expander graph $G$ (of constant expansion) on $n$ vertices, for $n$ big enough, where $d_G$ is a shortest path metric. Then for all non-expansive embedding $f : G \to \ell_p$ it holds that $\ell_1\text{-}dist(f) = \Omega(\log n)$, for any given $p \geq 1$.*

*Proof.* We use the Poincare inequity for expander graphs that was proved by Matousek (57): for all constant degree expander graph $G = (V, E)$ on $n$ nodes it holds that for any embedding $f : G \to \ell_p$

$$\frac{\sum\limits_{u \neq v \in V} \|f(u) - f(v)\|_p^p}{\sum\limits_{(u,v) \in E} \|f(u) - f(v)\|_p^p} \leq c_p \cdot n$$

for a constant $c_p$ (dependent on $p$).

In addition, for all constant degree graph $G = (V, E)$ on $n$ nodes it holds that $|E| = \Theta(n)$, and there are at least $cn^2$ pairs $u \neq v \in V$ with $d_G(u,v) \geq c' \log n$, for a constants $c, c' > 0$ - denote this set of pairs $X \subseteq \binom{V}{2}$. Therefore,

$$\frac{\sum\limits_{u \neq v \in V} (d_G(u,v))^p}{\sum\limits_{(u,v) \in E} (d_G(u,v))^p} \geq \Omega(n \log^p n)$$

Let $f : G \to \ell_p$ be any non-expansive embedding of a constant degree expander graph on $n$ nodes. For all $u \neq v \in V$ we have

$$\frac{1}{(contr_f(u,v))^p} \cdot (d_G(u,v))^p = \|f(u) - f(v)\|_p^p \leq (d_G(u,v))^p$$

Therefore,

$$\frac{\sum\limits_{u \neq v \in V} \|f(u) - f(v)\|_p^p}{\sum\limits_{(u,v) \in E} \|f(u) - f(v)\|_p^p} \geq \frac{\sum\limits_{u \neq v \in V} \frac{1}{(contr_f(u,v))^p} \cdot (d_G(u,v))^p}{\sum\limits_{(u,v) \in E} (d_G(u,v))^p} \geq \Omega \left( \frac{\log^p n \sum\limits_{u \neq v \in X} \frac{1}{contr_f(u,v)^p}}{\sum\limits_{(u,v) \in E} (d_G(u,v))^p} \right) \geq$$

$$\geq \Omega \left( \frac{n^2 \log^p n}{\sum\limits_{(u,v) \in E} (d_G(u,v))^p} \cdot \frac{\sum\limits_{u \neq v \in X} \frac{1}{contr_f(u,v)^p}}{|X|} \right) \geq \Omega \left( n \log^p n \cdot \frac{\sum\limits_{u \neq v \in X} \frac{1}{contr_f(u,v)^p}}{|X|} \right)$$

Therefore, we have

$$\frac{\sum\limits_{u \neq v \in X} \frac{1}{contr_f(u,v)^p}}{|X|} \leq O \left( \frac{1}{\log^p n} \right)$$

Namely, there are at least $\Theta(n^2)$ pairs $(u, v) \in X$ such that $\frac{1}{contr_f(u,v)^p} \leq O \left( \frac{1}{\log^p n} \right)$, what finishes the proof. $\qquad \square$

Finally, we contrast these lower bounds by noting that for general embeddings, that are not restricted to be either contractive or expansive, we can get constant $\ell_1$-distortion into $\ell_2^2$, as follows from Corollary 9. This complete the proof of Theorem 34.

The above result rises the following question: are there metric spaces $(X, d_X)$ and $(Y, d_Y)$ such that any non-expansive or non-contractive embedding $f : X \to Y$ has $\ell_q\text{-}dist(f) = n^{\Omega(\alpha(q))}$, while there exists a general type embedding $g : X \to Y$ with $\ell_q\text{-}dist(g) = O(1)$?[10]

**Theorem 40.** *Given any $q \geq 1$, there exist an $n$-point metric space $(P, d_P)$ and a scalable family of an $n$-point metric spaces $\mathcal{Q}$ such that the following holds: any non-expansive embedding $f : P \to \mathcal{Q}$ has $\ell_q\text{-}dist(f) = \Omega \left( n^{2/q} \right)$; any non-contractive embedding $f : P \to \mathcal{Q}$ has $\ell_q\text{-}dist(f) = \Omega \left( n^{2/q} \right)$; there exists an embedding $f : P \to \mathcal{Q}$ with $\ell_q\text{-}dist(f) = O(1)$.*

Following is the definition of composition of metric spaces we will use in this section.

**Definition 8.** *Let $(X, d_X)$ and $(Y, d_Y)$ be any metric spaces. The composition of $X$ with $Y$ is a metric space $(Z, d_Z) = comp^*(X, Y)$, defined by substituting an arbitrary point $x_0 \in X$ with a copy of $Y$, i.e. $Z = X \setminus \{x_0\} \cup Y$. The metric on $Z$ is defined as follows: for all $x \neq y \neq x_0 \in X$, $d_Z(x, y) = d_X(x, y)$; for all $x \neq y \in Y$, $d_Z(x, y) = d_Y(x, y)$; for all $x \neq x_0 \in X, y \in Y$, $d_Z(x, y) = d_X(x, x_0)$.*

*Proof of Theorem 40.* Let $(X, d_X)$ be an $(n/2)$-point equilateral metric space, and let $(Y, d_Y)$ be an $(n/2)$-point almost equilateral metric space: all the pairs except one have distance $1$, and the remaining pair has distance $\frac{1}{n^{2/q}}$. We call the pair of $Y$ with distance $\frac{1}{n^{2/q}}$ a *special pair of $Y$*.

Let $\epsilon = 1/n^4$. Define $(P, d_P) = comp^*(X, \ \epsilon \cdot Y)$, and define the family $\mathcal{Q} = \{\alpha \cdot Q | \alpha > 0, \alpha \in \mathbb{R}\}$, where $Q = comp^*(Y, \ \epsilon \cdot X)$.

Let us first prove item (3) of the claim. Choose $\alpha = 1$, and let $F : P \to Q$ be defined as an arbitrary bijection between the points of $X \subseteq P$ to $Y \subseteq Q$, and between the points of $\epsilon \cdot Y \subseteq P$ to $\epsilon \cdot X \subseteq Q$. It can be seen that only the special pair suffers distortion $n^{2/q}$ in this embedding, while other pairs have distortion $1$, thus

$$\ell_q\text{-}dist(F) \leq \left( \binom{n}{2} \cdot 1^q + 2 \left( n^{2/q} \right)^q \right)^{1/q}.$$

To prove item (1), let $f : P \to Q$ be any non-expansive embedding. Denote $(u, v) \in P$ the special pair of the subspace $\epsilon \cdot Y$, i.e. the only pair of $\epsilon \cdot Y$ of distance $\frac{\epsilon}{n^{2/q}}$. We consider the following cases:

1. Both $f(u), f(v) \in \epsilon \cdot X \subset \alpha \cdot Q$, for some $\alpha > 0$. Then, since $f$ is non-expansive, it should be that $\alpha \cdot \epsilon \leq \frac{\epsilon}{n^{2/q}}$, meaning $\alpha \leq \frac{1}{n^{2/q}}$. In such case, again since $f$ is non-expansive, all points of $\epsilon \cdot Y \subset P$ embed to $\epsilon \cdot X \subset \alpha \cdot Q$, implying $\ell_q\text{-}dist(f) \geq \left(\frac{\epsilon}{\alpha \cdot \epsilon}\right) \geq \Omega(n^{2/q})$.

2. The other case is when the distance between $f(u), f(v)$ is a distance of $\alpha \cdot Y$. . Then, since $f$ is non-expansive, it should be that either $\frac{\alpha}{n^{2/q}} \leq \frac{\epsilon}{n^{2/q}}$, in case $(u, v)$ embeds to the special pair of the subspace $Y \subset \alpha \cdot Q$. Otherwise, it must be $\alpha \leq \frac{\epsilon}{n^{2/q}}$. In both cases $\alpha \leq \epsilon$. Therefore, any pair $x, y \in X \subset P$ will suffer distortion $dist_f(x, y) = \frac{1}{\alpha} \geq \frac{1}{\epsilon} = n^4$. The distortion can only be larger if $x, y$ embed to $\alpha \cdot \epsilon \cdot X$ In both cases, it holds that $\ell_q\text{-}dist(f) \geq \Omega(n^2) = \Omega(n^{2/q})$.

The proof of item (2) follows by symmetric considerations.

$\square$

# F   On the limitations of classical MDS

**PCA Algorithm.** The input set of $n$ points in $\mathbb{R}^d$ is given explicitly as rows of an $n \times d$ matrix $X$ (it is also possible to define given distance matrix). Center the rows of $X$, i.e. subtract from each row the row's mean, to get matrix $\tilde{X}$. Consider $G = \tilde{X}\tilde{X}^t$ and let $G = Q\Lambda Q^t$ be its eigenvalue decomposition: $\Lambda$ is a diagonal $n \times n$ matrix with eigenvalues of $G$ ordered in decreasing order; $Q$ is the matrix of the corresponding eigenvectors at its columns (chosen to form an orthonormal basis). The embedding to $k$ dimensions is given by $Q_k \sqrt{\Lambda_k}$, where $Q_k$ is an $n \times k$ matrix of the first $k$ columns of $Q$ and $\sqrt{\Lambda_k}$ is a $k \times k$ diagonal matrix with the square roots of the largest eigenvalues of $G$.

**Lemma 41.** *There is an embedding $f : X \to \mathbb{R}$ with $Stress_q/Energy_q(f) = O(\alpha/d^{1/q})$, for any $q \geq 1$.*

*Proof.* Consider the following embedding of $X$: For all $i \in [1, d]$, all the points of $X_i^+$ are embedded into the point $\alpha^i$, and all the points of $X_i^-$ are embedded into the point $(-\alpha^i)$. We estimate $Stress_q$ value of this embedding. We note that the bound on $Energy_q$ is obtained for the embedding applied on the metric space defined by taking each $s_i = 1$, by a similar calculations. For all $u \neq v \in X$, let $d_{uv} = \|u - v\|_2$, and let $\hat{d}_{uv} = \|f(u) - f(v)\|_2$. Then, $\sum_{u \neq v}(d_{uv})^q \geq 4 \sum_{1 \leq i \leq j \leq d} s_i s_j (\sqrt{\alpha^{2i} + \alpha^{2j}})^q \geq 4 \sum_{1 \leq i \leq j \leq d} s_j s_i (\alpha^i)^q \geq 4d \cdot s_d$. The sum of additive errors is bounded by $\sum_{1 \leq i < j \leq d} 4 s_i s_j \left| \sqrt{\alpha^{2i} + \alpha^{2j}} - (\alpha^i - \alpha^j) \right|^q \leq 4 \sum_{1 \leq i < j \leq d} 2^q s_i s_j (\alpha^j)^q \leq 16 \cdot 2^q s_{d-1}$ (the details are deferred to the full version). Therefore, $Stress_q(f) = O(\alpha/d^{1/q})$.

$\square$

**Lemma 42.** *Let $F : X \to \mathbb{R}^k$ be an embedding produced by applying PCA algorithm on $X$, for $k \leq \beta \cdot d$, for a constant $\beta < 1$. Then, $Stress_q/Energy_q(F) = \Omega(1)$, and $\ell_q\text{-}dist/REM_q(F) = \infty$.*

*Proof.* We show that PCA will actually project the points of $X$ into the first $k$ dimensions. Note that the associated $n \times d$ matrix $X$ is already row centered, where $n = |X|$. Thus, we compute $G = X \cdot X^t$. The matrix $G$ is a block-diagonal matrix, having block matrices $\{B_i\}_{i \in [1,d]}$ as its diagonal. Each $B_i$ is an $2s_i \times 2s_i$ matrix of the form $B_i = \alpha^{2i} \cdot ww^t$, for a $2s_i$ dimensional column vector $w = (1, 1, \ldots, 1, -1, -1, \ldots -1)$. The eigenvalues of each $B_i$ are $\{0, 2\alpha_i s_i\}$. The eigenvector of the eigenvalue $2\alpha_i s_i$ is the vector $w$. Normalizing these vectors by $1/\sqrt{2s_i}$ we obtain a matrix $Q_k$, for any $k < d$. Eventually, we multiply $Q_k \cdot \sqrt{\Lambda}$ and obtain the resulting embedding, which is just the orthogonal projection to the first $k$ dimensions.

Since $|X| > 2d$, for $k < d$ there are points that c-MDS maps to 0, resulting in $\ell_q\text{-}dist/REM_q = \infty$. As for the $Stress_q$, we estimate $\sum_{u \neq v}(d_{uv})^q \leq 8d \sum_{j=1}^{d} s_j \leq 16d \cdot s_d$, and $\sum_{u \neq v \in X} |d_{uv} -$

$\hat{d}_{uv}|^q \geq \sum_{k < i \leq j \leq d} s_i s_j \left| \sqrt{\alpha^{2i} + \alpha^{2j}} \right|^q \geq \sum_{k < i \leq j \leq d} s_j \geq s_d \cdot (d - k)$, implying $Stress_q = \Omega(1)$,
for $k \leq \beta d$. $\qquad\qquad\qquad\qquad\qquad\qquad\qquad\qquad\qquad\qquad\qquad\qquad\qquad\qquad\qquad\square$

# G    Euclidean dimension reduction: moment analysis of the JL transform

## G.1    Moments of distortion analysis

In this appendix we complete the proofs of the statements that we left unproven in Section 3. In addition, we provide more precise (in terms of constants) analysis of the JL transform.

### G.1.1    Moment analysis for small $q$

We start with the full versions of lemmas that analyze the $\ell_q$-contraction ad $\ell_q$-expansion of the JL transform.

**Lemma 43.** *Given a finite set $X \subset \ell_2^d$ and integer $k \geq 2$, let $f : X \to \ell_2^k$ be the JL transform of dimension $k$. For any distribution $\Pi$ and for any $1 \leq q < k$:*

1. *For $1 \leq q < 2$, $E\left[ \left( \ell_q\text{-}contr^{(\Pi)}(f) \right)^q \right] \leq 1 + \frac{q}{k-q}$.*

2. *For $q \geq 2$, $E\left[ \left( \ell_q\text{-}contr^{(\Pi)}(f) \right)^q \right] \leq \left( 1 + \frac{q}{k-q} \right)^{\lfloor \frac{q}{2} \rfloor} \cdot \left( 1 + \frac{2\left( \frac{q}{2} - \lfloor \frac{q}{2} \rfloor \right)}{k - 2\left( \frac{q}{2} - \lfloor \frac{q}{2} \rfloor \right)} \right)$.*

3. *For $q \geq 2$, $E\left[ \left( \ell_q\text{-}contr^{(\Pi)}(f) \right)^q \right] \leq \left( 1 + \frac{q}{k-q} \right) \cdot e^{q/2}$.*

*Proof.* By the definition we have

$$E\left[ \left( \ell_q\text{-}contr^{(\Pi)}(f) \right)^q \right] = \int_0^\infty \left( \frac{k}{x} \right)^{\frac{q}{2}} \frac{x^{\frac{k}{2} - 1}}{2^{\frac{k}{2}} \Gamma\left( \frac{k}{2} \right) e^{\frac{x}{2}}} \, dx.$$

Denote $q/2 = l + r$, where $l = \lfloor q/2 \rfloor$, and $0 \leq r < 1$. Note that using this notation the term $\left( \frac{q}{2} - \lfloor \frac{q}{2} \rfloor \right)$ just expresses $r$. Next, we estimate the integral for various values of $q$.

$$\int_0^\infty \left( \frac{k}{x} \right)^{\frac{q}{2}} \frac{x^{\frac{k}{2} - 1}}{2^{\frac{k}{2}} \Gamma\left( \frac{k}{2} \right) e^{\frac{x}{2}}} \, dx = \frac{\left( \frac{k}{2} \right)^{\frac{q}{2}} \Gamma\left( \frac{k}{2} - \frac{q}{2} \right)}{\Gamma\left( \frac{k}{2} \right)} \qquad (8)$$

$$= \frac{\left( \frac{k}{2} \right)^l \left( \frac{k}{2} \right)^r \Gamma\left( \left( \frac{k}{2} - r \right) - l \right)}{\Gamma\left( \frac{k}{2} \right)}.$$

For $1 \leq q < 2$, $l = 0$ and $r = q/2$. Therefore, by Eq. 2, we obtain

$$\frac{\left( \frac{k}{2} \right)^r \Gamma\left( \frac{k}{2} - r \right)}{\Gamma\left( \frac{k}{2} \right)} \leq \frac{k/2}{k/2 - r} = 1 + \frac{q}{k - q}.$$

For $q \geq 2$, applying the rule $\Gamma(x) = (x - 1)\Gamma(x - 1)$, $l$ times we get

$$\frac{\left( \frac{k}{2} \right)^l \left( \frac{k}{2} \right)^r \Gamma\left( \left( \frac{k}{2} - r \right) - l \right)}{\Gamma\left( \frac{k}{2} \right)} = \frac{\left( \frac{k}{2} \right)^l}{\prod_{i=0}^{l-1} \left( \frac{k}{2} - \frac{q}{2} + i \right)} \cdot \frac{\left( \frac{k}{2} \right)^r \Gamma\left( \frac{k}{2} - r \right)}{\Gamma\left( \frac{k}{2} \right)} \leq \frac{\left( \frac{k}{2} \right)^l}{\prod_{i=0}^{l-1} \left( \frac{k}{2} - \frac{q}{2} + i \right)} \cdot \left( 1 + \frac{2r}{k - 2r} \right).$$

Denote $\Psi(q, k) := \frac{\left( \frac{k}{2} \right)^l}{\prod_{i=0}^{l-1} \left( \frac{k}{2} - \frac{q}{2} + i \right)} \cdot \left( 1 + \frac{2r}{k - 2r} \right)$. Since for all $0 \leq i \leq l - 1$ it holds that $\left( \frac{k}{2} - \frac{q}{2} + i \right) \geq \left( \frac{k}{2} - \frac{q}{2} \right)$, we have

$$\Psi(k, q) \leq \left( 1 + \frac{q}{k - q} \right)^{\lfloor \frac{q}{2} \rfloor} \cdot \left( 1 + \frac{2r}{k - 2r} \right),$$

which completes the proof of the second item of the lemma. To prove the third item of the lemma, we write

$$\frac{\left(\frac{k}{2}\right)^{\frac{q}{2}}\Gamma\left(\frac{k}{2}-\frac{q}{2}\right)}{\Gamma\left(\frac{k}{2}\right)} = \frac{k/2}{k/2-q/2} \cdot \frac{\left(\frac{k}{2}\right)^{q/2-1}\Gamma\left(\frac{k}{2}-\frac{q}{2}+1\right)}{\Gamma\left(\frac{k}{2}\right)}.$$

Using the estimations of Lemma 11 we get that the above is upper bounded by

$$\left(1+\frac{q}{k-q}\right) \cdot \frac{\left(\frac{k}{2}\right)^{\frac{q}{2}-1} \cdot \left(\frac{k}{2}-\frac{q}{2}\right)^{\frac{k}{2}-\frac{q}{2}} \cdot \sqrt{\frac{k}{2}-\frac{q}{2}+\frac{1}{2}} \cdot e^{\frac{k}{2}}}{e^{\frac{k}{2}-\frac{q}{2}} \cdot \sqrt{\frac{k}{2}} \cdot \left(\frac{k}{2}\right)^{\frac{k}{2}-1}} \leq \left(1+\frac{q}{k-q}\right) \cdot e^{\frac{q}{2}},$$

which completes the proof of the third item of the lemma. □

**Lemma 44.** *Given a finite set $X \subset \ell_2^d$ and integer $k \geq 1$, let $f : X \to \ell_2^k$ the JL transform of dimension $k$. For any distribution $\Pi$ and for any $q \geq 1$:*

$$E\left[\left(\ell_q\text{-}expans^{(\Pi)}(f)\right)^q\right] \leq \left(1+\frac{q}{k}\right)^{\lfloor\frac{q}{2}\rfloor}.$$

*Proof.* By the definition we have

$$E[(\ell_q\text{-}expans^{(\Pi)}(f))^q] = \int_0^\infty \left(\frac{x}{k}\right)^{\frac{q}{2}} \frac{x^{\frac{k}{2}-1}}{2^{\frac{k}{2}}\Gamma\left(\frac{k}{2}\right)e^{\frac{x}{2}}} \, dx = \frac{\Gamma\left(\frac{k}{2}+\frac{q}{2}\right)}{\left(\frac{k}{2}\right)^{\frac{q}{2}}\Gamma\left(\frac{k}{2}\right)}.$$

Denote $\frac{q}{2} = l + r$, where $l = \lfloor\frac{q}{2}\rfloor \geq 0$, and $0 \leq r < 1$. Therefore

$$\frac{\Gamma\left(\frac{k}{2}+\frac{q}{2}\right)}{\left(\frac{k}{2}\right)^{\frac{q}{2}}\Gamma\left(\frac{k}{2}\right)} = \frac{\Gamma\left(\left(\frac{k}{2}+r\right)+l\right)}{\left(\frac{k}{2}\right)^l\left(\frac{k}{2}\right)^r\Gamma\left(\frac{k}{2}\right)}.$$

If $l = 0$ we have

$$\frac{\Gamma\left(\left(\frac{k}{2}+r\right)+l\right)}{\left(\frac{k}{2}\right)^l\left(\frac{k}{2}\right)^r\Gamma\left(\frac{k}{2}\right)} = \frac{\Gamma\left(\frac{k}{2}+r\right)}{\left(\frac{k}{2}\right)^r\Gamma\left(\frac{k}{2}\right)} \leq 1,$$

as required. For all $q \geq 2$, we have

$$\frac{\Gamma\left(\left(\frac{k}{2}+r\right)+l\right)}{\left(\frac{k}{2}\right)^l\left(\frac{k}{2}\right)^r\Gamma(\frac{k}{2})} = \frac{\prod\limits_{0 \leq i \leq l-1}\left(\frac{k}{2}+r+i\right)}{\left(\frac{k}{2}\right)^l} \cdot \frac{\Gamma\left(\frac{k}{2}+r\right)}{\left(\frac{k}{2}\right)^r\Gamma\left(\frac{k}{2}\right)} \leq$$

$$\prod_{0 \leq i \leq l-1}\left(1+\frac{r+i}{\frac{k}{2}}\right) \leq \left(1+\frac{q}{k}\right)^{\lfloor\frac{q}{2}\rfloor}.$$

□

By Jensen's inequality the following corollary is (in almost all cases) immediate:

**Corollary 4.** *Given a finite set $X \subset \ell_2^d$ and integer $k \geq 1$, for the JL projection of dimension $k$, $f : X \to \ell_2^k$ the following assertions hold:*

1. *For all $k \geq 2$ and all $1 \leq q < k$:*

$$E\left[\ell_q\text{-}contr^{(\Pi)}(f)\right] \leq \left(E\left[\left(\ell_q\text{-}contr^{(\Pi)}(f)\right)^q\right]\right)^{\frac{1}{q}} \leq 1 + \frac{\max\{1, q/2\}}{(k-q)}.$$

2. *For all $k \geq 2$ and all $1 \leq q < k$:*

$$E\left[\ell_q\text{-}contr^{(\Pi)}(f)\right] \leq \left(E\left[\left(\ell_q\text{-}contr^{(\Pi)}(f)\right)^q\right]\right)^{\frac{1}{q}} \leq \sqrt{e} \cdot \left(1 + \frac{q}{k-q}\right)^{1/q}.$$

3. For all $k, q \geq 1$ it holds that

$$E\left[\ell_q\text{-}expans^{(\Pi)}(f)\right] \leq \left(E\left[\left(\ell_q\text{-}expans^{(\Pi)}(f)\right)^q\right]\right)^{\frac{1}{q}} \leq 1 + \frac{q}{2k}.$$

*Proof.* The last two items are trivial. Thus we focus on proving the first item, for which some algebra should be done. For $1 \leq q < 2$, from the first item of Lemma 43 we have

$$E\left[\ell_q\text{-}contr^{(\Pi)}(f)\right] \leq \left(E\left[\left(\ell_q\text{-}contr^{(\Pi)}(f)\right)^q\right]\right)^{\frac{1}{q}} \leq \left(1 + \frac{q}{k-q}\right)^{1/q} \leq 1 + \frac{1}{k-q},$$

as required. For $q \geq 2$, denote $q/2 = l + r$, where $l = \lfloor \frac{q}{2} \rfloor$. Then, from the second item of Lemma 43 we have

$$E\left[\ell_q\text{-}contr^{(\Pi)}(f)\right] \leq \left(E\left[\left(\ell_q\text{-}contr^{(\Pi)}(f)\right)^q\right]\right)^{\frac{1}{q}} \leq \left(1 + \frac{2r}{k-2r}\right)^{\frac{1}{q}} \cdot \left(1 + \frac{q}{k-q}\right)^{\lfloor \frac{q}{2} \rfloor \cdot \frac{1}{q}}$$

$$\leq \left(1 + \frac{2r}{q(k-2r)}\right) \cdot \left(1 + \frac{\lfloor \frac{q}{2} \rfloor}{k-q}\right) = \left(\frac{qk - 2qr + 2r}{q(k-2r)}\right) \cdot \left(\frac{k - \frac{q}{2} - r}{k-q}\right),$$

we have to show that the above is upper bounded by $\frac{k - \frac{q}{2}}{k-q}$, which is equivalent to showing that

$$-qkr + 2qr^2 + 2kr - qr - 2r^2 \leq 0 \Leftrightarrow 2(q-1)r - q \leq k(q-2) \Leftrightarrow 2(q-1) - q \leq k(q-2) \Leftrightarrow k \geq 1,$$

which completes the proof. $\qquad\square$

**Theorem 45.** *For a finite set $X \subset \ell_2^d$ and an integer $k \geq 2$, let $f : X \to \ell_2^k$ be the JL transform of dimension $k$. For any $1 \leq q < k$, and any distribution $\Pi$ over $\binom{X}{2}$, with positive probability,*[11]
$$\ell_q\text{-}dist^{(\Pi)}(f) \leq \left(1 + \frac{\max\{1, \frac{q}{2}\}}{(k-q)}\right)\left(1 + \min\left\{\frac{1}{\sqrt{\pi k}}, \frac{1}{q}\right\}\right).$$

*Proof.* By Claim 23, $E[(\ell_q\text{-}dist^{(\Pi)}(f))^q] \leq E[(\ell_q\text{-}contr^{(\Pi)}(f))^q] + E[(\ell_q\text{-}expans^{(\Pi)}(f))^q]$. By the second item of Lemma 43,

$$E[(\ell_q\text{-}contr^{(\Pi)}(f))^q] \leq \left(1 + \frac{q}{k-q}\right)^{\lfloor \frac{q}{2} \rfloor} \cdot \left(1 + \frac{2\left(\frac{q}{2} - \lfloor \frac{q}{2} \rfloor\right)}{k - 2\left(\frac{q}{2} - \lfloor \frac{q}{2} \rfloor\right)}\right).$$

In addition, by Lemma 44, it holds that

$$E[(\ell_q\text{-}expans^{(\Pi)}(f))^q] \leq \left(1 + \frac{q}{k}\right)^{\lfloor \frac{q}{2} \rfloor}.$$

Note that the bound on $E[(\ell_q\text{-}expans^{(\Pi)}(f))^q$ is bounded above by the bound of $E[(\ell_q\text{-}contr^{(\Pi)}(f))^q$. Thus, by Jensen's inequality:

$$E[\ell_q\text{-}dist^{(\Pi)}(f)] \leq 2^{1/q} \cdot \left(1 + \frac{q}{k-q}\right)^{\lfloor \frac{q}{2} \rfloor \cdot \frac{1}{q}} \cdot \left(1 + \frac{2\left(\frac{q}{2} - \lfloor \frac{q}{2} \rfloor\right)}{k - 2\left(\frac{q}{2} - \lfloor \frac{q}{2} \rfloor\right)}\right)^{\frac{1}{q}} \leq \left(1 + \frac{1}{q}\right)\left(1 + \frac{\max\{1, \frac{q}{2}\}}{(k-q)}\right),$$

where the last inequality follows from technical estimations appearing in Appendix G: Corollary 4. Note that for $q \geq \sqrt{\pi}\sqrt{k}$ this already proves the theorem. Also note that a slightly weaker bound (in terms of constants) follows by monotonicity for all $1 \leq q < \sqrt{\pi}\sqrt{k}$. To get the precise bound as stated in the theorem, we perform a more technically involved analysis. The full details of the proof appear in Theorem 50. $\qquad\square$

For values of $q$ that are very close to $k$ we can improve the estimations of the $\ell_q\text{-}dist(f)$.

**Remark 46.** *We note that the estimations we present next are correct for all $1 \leq q < k$, and the value of $q$ for which the bounds are better is obtained by comparing to the bounds of Theorem 45. Particularly, for $q > \tilde{c}k$, where $\tilde{c} = (e-1)/e$, the following estimations are better that the ones presented in Theorem 45.*

**Theorem 47.** *For a finite set $X \subset \ell_2^d$ and integer $k \geq 2$, let $f : X \to \ell_2^k$ be the JL transform of dimension $k$. For any distribution $\Pi$ over $\binom{X}{2}$ and any $1 \leq q < k$, with positive probability:*

$$\ell_q\text{-}dist^{(\Pi)}(f) \leq 2\sqrt{e}\left(\frac{k}{k-q}\right)^{\frac{1}{q}}.$$

*Proof.* The proof follows the analysis presented in Theorem 45, and uses the third item of Lemma 43:

$$E\left[\ell_q\text{-}dist^{(\Pi)}(f)\right] \leq \left(E\left[\left(\ell_q\text{-}dist^{(\Pi)}(f)\right)^q\right]\right)^{\frac{1}{q}} \leq 2^{1/q}\sqrt{e}\left(1+\frac{q}{k-q}\right)^{1/q},$$

which completes the proof. $\qquad\square$

**Theorem 48.** *Given a finite $X \subset \ell_2^d$ and an integer $k \geq 1$, let $f :\to \ell_2^k$ be the JL transform into $k$ dimensions. For any distribution $\Pi$ over $\binom{X}{2}$ and any $1 \leq q < k$, there exists $\alpha > 0$ such that with positive probability:*

$$\ell_q\text{-}dist^{(\Pi)}(\alpha \cdot f) \leq 4\left(\frac{k}{k-q}\right)^{\frac{1}{2q}}.$$

*Proof.* By Corollary 4(items 2 and 3), with positive probability $f$ is such that for a given $1 \leq q < k$ both $\ell_q\text{-}contr^{(\Pi)}(f) \leq \sqrt{e}\left(\frac{k}{k-q}\right)^{1/q}$ and $\ell_q\text{-}expans^{(\Pi)}(f) \leq \left(1+\frac{q}{k}\right)^{1/2}$. Therefore, by the second item of Claim 23, there exists a normalization factor $\alpha$, such that for the embedding $\alpha \cdot f$:

$$\ell_q\text{-}dist^{(\Pi)}(\alpha \cdot f) \leq 2^{1/q}e^{1/4} \cdot \left(1+\frac{q}{k-q}\right)^{\frac{1}{2q}} \cdot \left(1+\frac{q}{k}\right)^{\frac{1}{4}} \leq 4\left(\frac{k}{k-q}\right)^{\frac{1}{2q}},$$

which completes the proof. $\qquad\square$

**Corollary 5.** *Given a finite set $X \subset \ell_2^d$, $1 \leq q < k$ and $0 < \delta < 1$, an embedding $f : X \to \ell_2^k$ can be computed in time $O(dk\log(1/\delta))$, so that for any distribution $\Pi$ over $\binom{X}{2}$, with probability at least $1-\delta$: if $q \leq \tilde{c}k$ then $\ell_q\text{-}dist^{(\Pi)}(f) \leq 1 + \frac{1}{\delta} \cdot \left(\frac{\max\{2,q\}}{k-q} + \frac{1}{\sqrt{\pi}\sqrt{k}}\right)$; if $q > \tilde{c}k$ then $\ell_q\text{-}dist^{(\Pi)}(f) \leq \frac{1}{\delta} \cdot 8\left(\frac{1}{k-q}\right)^{\frac{1}{2q}}.$*

*Proof.* In the proof of Theorem 45, we estimated the expected value of the random variable $\ell_q\text{-}dist^{(\Pi)}(f)$. Namely, the bounds stated in the theorem are upper bounds on the $E\left[\ell_q\text{-}dist^{(\Pi)}(f)\right]$.

For $q \leq \tilde{c}k$, we have the following estimation

$$E\left[\ell_q\text{-}dist^{(\Pi)}(f) - 1\right] \leq \frac{\max\{2,q\}}{k-q} + \frac{1}{\sqrt{\pi}\sqrt{k}}.$$

Therefore, the first part of the claim follows from Markov's inequality. For $q > \tilde{c}k$, we have the estimation

$$E\left[\ell_q\text{-}dist^{(\Pi)}(f)\right] \leq 8/(k-q)^{\frac{1}{2q}},$$

the bound follows from Markov's inequality. $\qquad\square$

**Remark 49.** *We note that all the bounds on the $\ell_q$-distortion we have presented grow to infinity when $q$ tends to $k$ what is, obviously, an overestimation. For $\Pi$ being a uniform distribution[12], let $k^- = \left(1 - \frac{1}{\log n + 1}\right) \cdot k$ (for any $0 < \tilde{c} < 1$ there exists $n$ big enough such that that $k^- > \tilde{c}k$). Then, $\ell_{k^-}\text{-}dist^{(\Pi)}(f) = O\left(\left(\sqrt{\log n}\right)^{\frac{1}{k}}\right)$, and in addition by Fact 14, for all $q \geq k^-$, $\ell_q\text{-}dist^{(\Pi)}(f) = O\left(n^{\frac{2}{k^-} - \frac{2}{q}} \cdot (\sqrt{\log n})^{\frac{1}{k}}\right)$, i.e. for all $k^- \leq q < k$, $\ell_q\text{-}dist^{(\Pi)}(f) = O\left(\left(\sqrt{\log n}\right)^{1/k}\right)$.*

### G.1.2 Precise analysis of moments of distortions of the JL transform

We prove the following theorem:

**Theorem 50** (Precise analysis of the JL for small $q$). *Given a finite set $X \subset \ell_2^d$ and integer $k \geq 2$, let $f : X \to \ell_2^k$ be the JL transform of dimension $k$. For any $1 \leq q < k$, and any distribution $\Pi$ over $\binom{X}{2}$, with positive probability*

$$\ell_q\text{-}dist^{(\Pi)}(f) \leq \left(1 + \frac{\max\{1, \frac{q}{2}\}}{(k-q)}\right) \cdot \left(1 + \frac{1}{\sqrt{\pi}\sqrt{k}}\right).$$

*Proof.* By the definition we have

$$E\left[\left(\ell_q\text{-}dist^{(\Pi)}(f)\right)^q\right] = \int_0^k \left(\frac{k}{x}\right)^{\frac{q}{2}} \frac{x^{\frac{k}{2}-1}}{2^{\frac{k}{2}}\Gamma\left(\frac{k}{2}\right) e^{\frac{x}{2}}}\, dx + \int_k^\infty \left(\frac{x}{k}\right)^{\frac{q}{2}} \frac{x^{\frac{k}{2}-1}}{2^{\frac{k}{2}}\Gamma\left(\frac{k}{2}\right) e^{\frac{x}{2}}}\, dx =$$

$$\int_0^\infty \left(\frac{k}{x}\right)^{\frac{q}{2}} \frac{x^{\frac{k}{2}-1}}{2^{\frac{k}{2}}\Gamma\left(\frac{k}{2}\right) e^{\frac{x}{2}}}\, dx + \tag{9}$$

$$\int_k^\infty \left(\frac{x}{k}\right)^{\frac{q}{2}} \frac{x^{\frac{k}{2}-1}}{2^{\frac{k}{2}}\Gamma\left(\frac{k}{2}\right) e^{\frac{x}{2}}}\, dx - \int_k^\infty \left(\frac{k}{x}\right)^{\frac{q}{2}} \frac{x^{\frac{k}{2}-1}}{2^{\frac{k}{2}}\Gamma\left(\frac{k}{2}\right) e^{\frac{x}{2}}}\, dx. \tag{10}$$

Denote $\frac{q}{2} = l + r$ where $l = \lfloor \frac{q}{2} \rfloor$, and $0 \leq r < 1$. Note that $l \geq 0$. The integral in Eq. 9 is $E\left[\left(\ell_q\text{-}contr^{(\Pi)}(f)\right)^q\right] \leq \Psi(q, k)$, as shown in Lemma 43.

Next we estimate the integrals in equation 10.

$$\int_k^\infty \left(\frac{x}{k}\right)^{\frac{q}{2}} \frac{x^{\frac{k}{2}-1}}{2^{\frac{k}{2}}\Gamma\left(\frac{k}{2}\right) e^{\frac{x}{2}}}\, dx - \int_k^\infty \left(\frac{k}{x}\right)^{\frac{q}{2}} \frac{x^{\frac{k}{2}-1}}{2^{\frac{k}{2}}\Gamma\left(\frac{k}{2}\right) e^{\frac{x}{2}}}\, dx = \frac{\Gamma\left(\frac{k}{2} + \frac{q}{2}, \frac{k}{2}\right)}{\left(\frac{k}{2}\right)^{\frac{q}{2}}\Gamma\left(\frac{k}{2}\right)} - \frac{\left(\frac{k}{2}\right)^{\frac{q}{2}}\Gamma\left(\frac{k}{2} - \frac{q}{2}, \frac{k}{2}\right)}{\Gamma\left(\frac{k}{2}\right)} =$$

$$\frac{\Gamma\left(\left(\frac{k}{2} + r\right) + l, \frac{k}{2}\right)}{\left(\frac{k}{2}\right)^l \left(\frac{k}{2}\right)^r \Gamma\left(\frac{k}{2}\right)} - \frac{\Gamma\left(\left(\frac{k}{2} - r\right) - l, \frac{k}{2}\right) \left(\frac{k}{2}\right)^l \left(\frac{k}{2}\right)^r}{\Gamma\left(\frac{k}{2}\right)}.$$

Each summand is treated separately in a sequel. Applying the rule $\Gamma(s, x) = (s - 1)\Gamma(s - 1, x) + x^{s-1}e^{-x}$, $l$ times we obtain

$$\frac{\Gamma\left(\left(\frac{k}{2} + r\right) + l, \frac{k}{2}\right)}{\left(\frac{k}{2}\right)^l \left(\frac{k}{2}\right)^r \Gamma\left(\frac{k}{2}\right)} = \frac{\prod_{i=1}^l \left(\frac{k}{2} + \frac{q}{2} - i\right)}{\left(\frac{k}{2}\right)^l} \cdot \frac{\Gamma\left(\frac{k}{2} + r, \frac{k}{2}\right)}{\left(\frac{k}{2}\right)^r \Gamma\left(\frac{k}{2}\right)} + S_1(q, k),$$

where

$$S_1(q, k) = \frac{\sum_{i=0}^{l-2}\left(\prod_{j=i+1}^{l-1} \frac{k}{2} - r + j\right)\left(\frac{k}{2}\right)^{\frac{k}{2}+i} + \left(\frac{k}{2}\right)^{\frac{k}{2}+(l-1)}}{\left(\frac{k}{2}\right)^l e^{\frac{k}{2}}\Gamma\left(\frac{k}{2}\right)},$$

and

---

$$\frac{\Gamma\left(\left(\frac{k}{2}-r\right)-l,\frac{k}{2}\right)\left(\frac{k}{2}\right)^l\left(\frac{k}{2}\right)^r}{\Gamma\left(\frac{k}{2}\right)} = \frac{\left(\frac{k}{2}\right)^l}{\prod_{i=0}^{l-1}\left(\frac{k}{2}-\frac{q}{2}+i\right)}\cdot\frac{\Gamma\left(\frac{k}{2}-r,\frac{k}{2}\right)\left(\frac{k}{2}\right)^r}{\Gamma\left(\frac{k}{2}\right)} - S_2(q,k),$$

where

$$S_2(q,k) = \sum_{i=1}^{l}\frac{\left(\frac{k}{2}\right)^{\frac{k}{2}-i}\left(\frac{k}{2}\right)^l}{e^{\frac{k}{2}}\Gamma\left(\frac{k}{2}\right)\prod_{j=i}^{l}\left(\left(\frac{k}{2}+r\right)-j\right)}.$$

Let $T_1 := \frac{\prod_{i=1}^{l}\left(\frac{k}{2}+\frac{q}{2}-i\right)}{\left(\frac{k}{2}\right)^l}$, and $T_2 := \frac{\left(\frac{k}{2}\right)^l}{\prod_{i=0}^{l-1}\left(\frac{k}{2}-\frac{q}{2}+i\right)}$. Note that $T_1 \leq T_2 \leq \Psi(q,k)$, and $T_2 = \Psi(q,k)\cdot\frac{k/2}{k/2-r}$. Therefore, Equation 10 is upper bounded by

$$T_2\cdot\left(\frac{\Gamma\left(\frac{k}{2}+r,\frac{k}{2}\right)}{\left(\frac{k}{2}\right)^r\Gamma\left(\frac{k}{2}\right)} - \frac{\Gamma\left(\frac{k}{2}-r,\frac{k}{2}\right)\left(\frac{k}{2}\right)^r}{\Gamma\left(\frac{k}{2}\right)}\right) + S_1(q,k) + S_2(q,k).$$

Next we estimate the difference of the first two terms. Using the estimation of Lemma 13 we get:

$$\frac{\Gamma\left(\frac{k}{2}+r,\frac{k}{2}\right)}{\left(\frac{k}{2}\right)^r\Gamma\left(\frac{k}{2}\right)} \leq \frac{\Gamma\left(\frac{k}{2},\frac{k}{2}\right)}{\Gamma\left(\frac{k}{2}\right)} + r\cdot\frac{\left(\frac{k}{2}\right)^{\frac{k}{2}-1}}{e^{\frac{k}{2}}\Gamma\left(\frac{k}{2}\right)},$$

and

$$\frac{\Gamma\left(\frac{k}{2}-r,\frac{k}{2}\right)\left(\frac{k}{2}\right)^r}{\Gamma\left(\frac{k}{2}\right)} \geq \frac{\Gamma\left(\frac{k}{2},\frac{k}{2}\right)}{\Gamma\left(\frac{k}{2}\right)}\cdot\left(\frac{\frac{k}{2}}{\frac{k}{2}-r}\right)^r - r\cdot\left(\frac{\frac{k}{2}}{\frac{k}{2}-r}\right)\cdot\frac{\left(\frac{k}{2}\right)^{\frac{k}{2}-1}}{e^{\frac{k}{2}}\Gamma\left(\frac{k}{2}\right)}.$$

Therefore,

$$T_2\cdot\left(\frac{\Gamma\left(\frac{k}{2}+r,\frac{k}{2}\right)}{\left(\frac{k}{2}\right)^r\Gamma\left(\frac{k}{2}\right)} - \frac{\Gamma\left(\frac{k}{2}-r,\frac{k}{2}\right)\left(\frac{k}{2}\right)^r}{\Gamma\left(\frac{k}{2}\right)}\right) \leq T_2\cdot r\cdot\frac{\left(\frac{k}{2}\right)^{\frac{k}{2}-1}}{e^{\frac{k}{2}}\Gamma\left(\frac{k}{2}\right)} + T_2\cdot\left(\frac{\frac{k}{2}}{\frac{k}{2}-r}\right)\cdot r\cdot\frac{\left(\frac{k}{2}\right)^{\frac{k}{2}-1}}{e^{\frac{k}{2}}\Gamma\left(\frac{k}{2}\right)}$$

$$\leq 2r\cdot\Psi(q,k)\cdot\frac{\left(\frac{k}{2}\right)^{\frac{k}{2}-1}}{e^{\frac{k}{2}}\Gamma\left(\frac{k}{2}\right)}.$$

Next we estimate

$$S_1(q,k)+S_2(q,k) \leq \frac{\sum_{i=0}^{l-2}\left(\prod_{j=i+1}^{l-1}\frac{k}{2}-r+j\right)\left(\frac{k}{2}\right)^{\frac{k}{2}+i}+\left(\frac{k}{2}\right)^{\frac{k}{2}+(l-1)}}{\left(\frac{k}{2}\right)^l e^{\frac{k}{2}}\Gamma\left(\frac{k}{2}\right)} + \sum_{i=1}^{l}\frac{\left(\frac{k}{2}\right)^{\frac{k}{2}-i}\left(\frac{k}{2}\right)^l}{e^{\frac{k}{2}}\Gamma\left(\frac{k}{2}\right)\prod_{j=i}^{l}\left(\left(\frac{k}{2}+r\right)-j\right)}$$

$$\leq l\cdot\frac{\left(\left(\frac{k}{2}+\frac{q}{2}\right)-1\right)\left(\left(\frac{k}{2}+\frac{q}{2}\right)-2\right)\cdots\left(\left(\frac{k}{2}+\frac{q}{2}\right)-(l-1)\right)\left(\frac{k}{2}\right)^{\frac{k}{2}}}{\left(\frac{k}{2}\right)^l e^{\frac{k}{2}}\Gamma\left(\frac{k}{2}\right)}$$

$$+l\cdot\frac{\left(\frac{k}{2}\right)^{l-1}\left(\frac{k}{2}\right)^{\frac{k}{2}}}{\left(\frac{k}{2}-\frac{q}{2}\right)\left(\left(\frac{k}{2}-\frac{q}{2}\right)+1\right)\cdots\left(\left(\frac{k}{2}-\frac{q}{2}\right)+(l-1)\right)e^{\frac{k}{2}}\Gamma\left(\frac{k}{2}\right)}$$

$$= \frac{l\left(\frac{k}{2}\right)^{\frac{k}{2}-1}}{e^{\frac{k}{2}}\Gamma\left(\frac{k}{2}\right)}\cdot\left(\frac{\left(\left(\frac{k}{2}+\frac{q}{2}\right)-1\right)\cdots\left(\left(\frac{k}{2}+\frac{q}{2}\right)-(l-1)\right)\frac{k}{2}}{\left(\frac{k}{2}\right)^l} + \frac{\left(\frac{k}{2}\right)^l}{\left(\frac{k}{2}-\frac{q}{2}\right)\cdots\left(\left(\frac{k}{2}-\frac{q}{2}\right)+(l-1)\right)}\right) \leq$$

$$2l\cdot\Psi(q,k)\cdot\frac{\left(\frac{k}{2}\right)^{\frac{k}{2}-1}}{e^{\frac{k}{2}}\Gamma\left(\frac{k}{2}\right)}.$$

Summarizing, we conclude that $\forall\, 1 \leq q < k$ it holds

$$E\left[\left(\ell_q\text{-}dist^{(\Pi)}(f)\right)^q\right] \leq \Psi(q,k) + q \cdot \Psi(q,k) \cdot \frac{\left(\frac{k}{2}\right)^{\frac{k}{2}-1}}{e^{\frac{k}{2}}\Gamma\left(\frac{k}{2}\right)} = \Psi(q,k)\left(1 + \frac{q \cdot \left(\frac{k}{2}\right)^{\frac{k}{2}-1}}{e^{\frac{k}{2}}\Gamma\left(\frac{k}{2}\right)}\right) \leq \Psi(q,k)\cdot\left(1 + \frac{q}{\sqrt{\pi}\sqrt{k}}\right),$$

where the last inequality is due to Lemma 12. Now, by Jensen's inequality we have

$$E\left[\ell_q\text{-}dist^{(\Pi)}(f)\right] \leq \left(E\left[\left(\ell_q\text{-}dist^{(\Pi)}(f)\right)^q\right]\right)^{\frac{1}{q}} \leq (\Psi(q,k))^{\frac{1}{q}} \cdot \left(1 + \frac{q}{\sqrt{\pi}\sqrt{k}}\right)^{\frac{1}{q}}$$

$$\leq \left(1 + \frac{\max\{1,\frac{q}{2}\}}{(k-q)}\right) \cdot \left(1 + \frac{1}{\sqrt{\pi}\sqrt{k}}\right),$$

by the estimations of the first item of Corollary 4. This completes the proof of the theorem. $\qquad\square$

### G.1.3 Moment analysis for large $q$

Next we present moment analysis of the JL transform for large values of $q$.

**Theorem 51.** *For any n-point $X \subset \ell_2^d$ and any integer $k \geq 1$, let $f : X \to \ell_2^k$ be the JL transform of dimension $k$. For any $q \geq k$ and any distribution $\Pi$ over $\binom{X}{2}$, there is a constant $\alpha > 0$ such that with constant probability:*[13]

*(A) For $q = k$: $\ell_k\text{-}dist^{(\Pi)}(\alpha \cdot f) = O\left(\left(\sqrt{\log n}\right)^{\frac{1}{k}}\right)$.*

*(B) For $q > k$: $\ell_q\text{-}dist^{(\Pi)}(\alpha \cdot f) = O\left(\frac{n^{\frac{1}{k}-\frac{1}{q}}}{(q-k)^{1/(2q)}} \cdot \left(\frac{q}{k}\right)^{\frac{1}{4}}\right)$.*

We first prove the following lemma.

**Lemma 52.** *For any $C \geq 1$, let $A_C$ denote the event $A_C := \frac{\|f(x)-f(y)\|_2^2}{\|x-y\|_2^2} \geq \frac{1}{C}, \forall x \neq y \in X$. Then for $C \geq 2en^{\frac{4}{k}}$ it holds that $\Pr[A_C] \geq \frac{1}{2}$.*

*Proof.* Recall that $\forall\, x \in \mathbb{R}^d$, s.t $\|x\|_2 = 1$, the variable $\|f(x)\|_2^2 \sim \chi_k^2/k$. Therefore, for any such $x$ and for any $C \geq 1$ it holds that

$$Pr\left[\|f(x)\|_2^2 < \frac{1}{C}\right] = Pr\left[\chi_k^2 < \frac{k}{C}\right] = \int_0^{\frac{k}{C}} \frac{x^{\frac{k}{2}-1}}{2^{\frac{k}{2}}\Gamma\left(\frac{k}{2}\right)e^{\frac{x}{2}}}\, dx \leq \frac{\left(\frac{k}{2}\right)^{\frac{k}{2}-1}}{\Gamma\left(\frac{k}{2}\right)} \cdot \frac{1}{(C)^{\frac{k}{2}}}.$$

For $k \geq 1$, using the estimation of Lemma 12, we get $\frac{\left(\frac{k}{2}\right)^{\frac{k}{2}-1}}{\Gamma\left(\frac{k}{2}\right)} \leq e^{\frac{k}{2}}$. Therefore, choosing $C \geq 2en^{\frac{4}{k}}$ we obtain

$$Pr\left[\|f(x)\|_2^2 < \frac{1}{C}\right] \leq \frac{1}{n^2}.$$

Applying the union bound we conclude the lemma. $\qquad\square$

The proof of Theorem 51 essentially follows from the following lemma.

**Lemma 53** (Higher Moments Analysis.). *Given any integer $k \geq 1$, any $q > k$, and any distribution $\Pi$ over $\binom{X}{2}$, the JL transform $f : X \to \ell_2^k$ satisfies the following:*

*(A)* $E\left[\left(\ell_k\text{-}contr^{(\Pi)}(f)\right)^k \Big| A_C\right] = O(\log n)$, *and* $E\left[\left(\ell_k\text{-}expans^{(\Pi)}(f)\right)^k\right] \leq 2^{\frac{k}{2}}$.

*(B)* $E\left[\left(\ell_q\text{-}contr^{(\Pi)}(f)\right)^q \Big| A_C\right] = O\left(\frac{(2e)^{q/2}\cdot\sqrt{k}\cdot n^{2\left(\frac{q}{k}-1\right)}}{(q-k)}\right)$, *and* $E\left[\left(\ell_q\text{-}expans^{(\Pi)}(f)\right)^q\right] \leq \left(2 \cdot \left(\frac{q}{k}\right)\right)^{\frac{q}{2}}$.

*Proof.* First we note that for all $q \geq k \geq 1$, by Lemma 44 it holds that

$$E\left[\left(\ell_q\text{-}expans^{(\Pi)}(f)\right)^q\right] \leq \left(1 + \frac{q}{k}\right)^{\frac{q}{2}},$$

which proves the second estimations in both $A$ and $B$. Thus, it remains to prove the first estimations in $A$ and $B$. For $A$ we have

$$E\left[\left(\ell_k\text{-}contr^{(\Pi)}(f)\right)^k \Big| A_C\right] \leq 2\int_{\frac{k}{C}}^{\infty} \left(\frac{k}{x}\right)^{\frac{k}{2}} \frac{x^{\frac{k}{2}-1}}{2^{\frac{k}{2}}\Gamma\left(\frac{k}{2}\right)e^{\frac{x}{2}}} \, dx \leq$$

$$\frac{2\left(\frac{k}{2}\right)^{\frac{k}{2}}}{\Gamma\left(\frac{k}{2}\right)} \int_{\frac{k}{C}}^{k} \frac{1}{x} \, dx + \frac{2\left(\frac{k}{2}\right)^{\frac{k}{2}}}{\Gamma\left(\frac{k}{2}\right)} \int_{k}^{\infty} \frac{1}{e^{\frac{x}{2}}} \, dx,$$

where the first integral is bounded by ignoring $e^{-\frac{x}{2}}$, and the second integral is bounded by ignoring $\frac{1}{x}$. Computing the first integral and using the estimation of Lemma 12, we obtain

$$\frac{2\left(\frac{k}{2}\right)^{\frac{k}{2}}}{\Gamma\left(\frac{k}{2}\right)} \int_{\frac{k}{C}}^{k} \frac{1}{x} \, dx = O\left(e^{k/2}\log n\right).$$

The second integral is bounded by $O\left(\frac{1}{\sqrt{k}}\right)$. Therefore, we obtain the first part of $A$. For the part $B$ we have

$$E\left[\left(\ell_q\text{-}contr^{(\Pi)}(f)\right)^q \Big| A_C\right] \leq 2\int_{\frac{k}{C}}^{\infty} \left(\frac{k}{x}\right)^{\frac{q}{2}} \frac{x^{\frac{k}{2}-1}}{2^{\frac{k}{2}}\Gamma\left(\frac{k}{2}\right)e^{\frac{x}{2}}} \, dx \leq \frac{2k^{\frac{q}{2}}}{2^{\frac{k}{2}}\Gamma\left(\frac{k}{2}\right)} \int_{\frac{k}{C}}^{\infty} x^{\left(\frac{k-q}{2}-1\right)} \, dx.$$

Therefore, computing the value of the integral we obtain

$$\int_{\frac{k}{C}}^{\infty} x^{\frac{k-q}{2}-1} \, dx = \frac{x^{(k/2-q/2)}}{k/2-q/2}\Big|_{\frac{k}{C}}^{\infty} \leq \frac{(C/k)^{q/2-k/2}}{(q-k)/2}.$$

Substituting $C$ with its value, we conclude that $E\left[\left(\ell_q\text{-}contr^{(\Pi)}(f)\right)^q \Big| A_C\right] = O\left(\frac{(2e)^{q/2}\sqrt{k}}{q-k} \cdot n^{2\left(\frac{q}{k}-1\right)}\right)$, which concludes the first part of $B$. This completes the proof. $\quad\square$

Now we are ready to prove the theorem.

*Proof of Theorem 51.* The proof simply follows from Jensen's and Markov's inequalities applied to Lemma 53, and invoking second part of Claim 23.

$\square$

**Remark 54.** *Note that the estimation in item $(B)$ grows to infinity when $q$ gets closer to $k$. Using similar consideration to Remark 49, let $k^+ = \left(1 + \frac{1}{\log n + 1}\right) \cdot k$, then for all $k < q \leq k^+$, $\ell_q\text{-}dist^{(\Pi)}(f) \leq \ell_{k^+}\text{-}dist^{(\Pi)}(f) = O\left((\sqrt{\log n})^{1/k}\right).$*

## G.2  All $q$ Simultaneously

**Theorem 55.** *Given an $n$-point set $X \subset \ell_2^d$ and an integer $k \geq 1$, let $f : X \to \ell_2^k$ be the JL transform of dimension $k$. For any distribution $\Pi$ over $\binom{X}{2}$, with probability at least $1/2$, $\ell_q\text{-}dist^{(\Pi)}(f)$ is bounded above by*

| $1 \leq q \leq \sqrt{k}$ | $\sqrt{k} \leq q \leq \frac{k}{4}$ | $\frac{k}{4} \leq q \leq k^-$ | $k^- \leq q \leq k^+$ | $k^+ \leq q \leq \log n$ |
|---|---|---|---|---|
| $1 + O\left(\frac{1}{\sqrt{k}}\right)$ | $1 + O\left(\frac{q}{k-q}\right)$ | $O\left(\left(\frac{k}{k-q}\right)^{1/q}\left(\log\left(\frac{k}{k-q}\right)\right)^{2/q}\right)$ | $O\left((\log n)^{1/k}\right)$ | $O\left(n^{\frac{2}{k}-\frac{2}{q}} \cdot W\right)$, where $W = \min\left\{(\log n)^{1/k}, \left(\frac{\log n}{q-k}\right)^{1/q}\right\}$ |

*Proof.* For any $q \geq 1$ denote by $D(q)$ the estimation on $E\left[\ell_q\text{-}dist(f)^q\right]^{\frac{1}{q}}$, where $f$ is the JL transform of dimension $k$. We divide into cases according to the intervals of the theorem. In (almost) each interval we implement the same paradigm. We make use of the following fact:

**Fact G.1.** *For any $t_q \geq 1$ it holds that $Pr[\ell_q\text{-}dist^{(\Pi)}(f) > t_q(E[(\ell_q\text{-}dist^{(\Pi)}(f))^q]^{\frac{1}{q}})] \leq \frac{1}{(t_q)^q}$.*

Given an interval $I$, we define the finite set of discrete values $Q = \{q_j\} \subset I$, the real function $t : \mathbb{R} \to \mathbb{R}$ such that the following two properties hold:

Property 1: For all $q_j \in Q$, $\sum_j \frac{1}{(t(q_j))^{q_j}} \leq 1/10$.

Property 2: For all $q_{j-1} < q < q_j$, $\frac{t(q_j)D(q_j)}{t(q)D(q)} \leq O(1)$.

From the first property, Fact G.1 and the union bound it follows that with probability at least $9/10$, for all $q_j \in Q$ it holds that

$$\ell_{q_j}\text{-}dist(f) \leq t(q_j)D(q_j).$$

From the second property, and from monotonicity of the $\ell_q$-distortion, it follows that for all $q_j < q < q_{j+1}$,

$$\ell_q\text{-}dist(f) \leq O(1)t(q)D(q).$$

Putting all together, we conclude that with probability at least $9/10$ the embedding $f$ has bounded $\ell_q$-distortion with loss of factor $t(q)$ (with respect to the upper bound $D(q)$), for all $q \in I$ simultaneously. Since we have 5 ranges to treat, we conclude that with probability at least $1/2$ the embedding $f$ has bounded $\ell_q$-distortion for all $q$ simultaneously.

1. In this range it holds that $D(q) = 1 + \Theta\left(\frac{1}{\sqrt{k}}\right)$, for all $q$. We define $Q = \{\sqrt{k}\}$, and $t(q) = 10^{1/\sqrt{k}}$, for all $q$. Therefore, $t(\sqrt{k}) = 10^{1/\sqrt{k}} = 1 + \Theta\left(\frac{1}{\sqrt{k}}\right)$. Therefore, with probability at least $9/10$, for $q = \sqrt{k}$ it holds that $\ell_q\text{-}dist(f) \leq 1 + O\left(\frac{1}{\sqrt{k}}\right)$. Since $\ell_q$-distortion is monotonic, we have that for all $1 \leq q \leq \sqrt{k}$, $\ell_q\text{-}dist(f) \leq 1 + O\left(\frac{1}{\sqrt{k}}\right)$, with probability at least $9/10$.

2. In this range it holds that $D(q) = 1 + \Theta\left(\frac{q}{k-q}\right)$. For $1 \leq j \leq \frac{\log n}{2} - 1$, define $q_j = \sqrt{k}2^j$. For all $q \geq 1$, define $t(q) = 1 + \frac{q}{k-q}$. Then,

$$\sum_j \frac{1}{(t(q_j))^{q_j}} = \sum_j \left(1 - \frac{2^j}{\sqrt{k}}\right)^{\sqrt{k}2^j} \leq \sum_j e^{-4j} \leq 1/10.$$

Since this range treats small distortions, we would like to refine the requirement of Property 2. Given any $q_{j-1} < q < q_j$

$$\frac{t(q_j)D(q_j) - 1}{t(q)D(q) - 1} = O\left(\frac{q_j(k-q)}{q(k-q_j)}\right) = O(1).$$

This finishes the proof of this case.

3. In this range it holds that $D(q) = \Theta\left(\left(\frac{k}{k-q}\right)^{1/q}\right)$. We make the following choice. Let $q_0 = k/4$, and for $1 \leq j \leq \frac{\log\log n}{k}$ let $q_j = k\left(1 - \frac{1}{2^{jk}}\right)$. Let the function $t$ be such that on the points $q_j$ it satisfies $t(q_j) = (20j^2)^{1/q_j}$. Namely, for all $q \geq 1$, the function is defined by $t(q) = \left(\frac{20\log\left(\frac{k}{k-q}\right)}{k}\right)^{2/q}$. For the first property, we estimate

$$\sum_j \frac{1}{(t(q_j))^{q_j}} = \sum_j \frac{1}{20j^2} \leq 1/10,$$

as required.

For the second property we have for all $q_{j-1} < q < q_j$,

$$D(q_j) = \Theta\left(\left(\frac{k}{k-q_j}\right)^{1/q_j}\right) = \Theta\left((2^{jk})^{1/q_j}\right),$$

since $q_j \geq k/2$, for all $j \geq 1$, we obtain $D(q_j) = \Theta(4^j)$. Since $D(q)$ is monotonically increasing on $q$, we have that $D(q) \geq D(q_{j-1})$, therefore $\frac{D(q_j)}{D(q)} = O(1)$, as required. In addition, since for all $j$, $t(q_j) = \Theta\left((\log(D(q_j)^{q_j}))^{2/q_j}\right)$ it holds that $\frac{t(q_j)}{t(q)} = O(1)$, as required to satisfy the second property.

4. In this range it holds that $D(q) = \Theta\left((\log n)^{1/k}\right)$. In this case we do not follow the general paradigm, but rather use the basic $\ell_q$-distortion relations. Particularly, the JL transform $f$ is such that with probability at least $9/10$, $\ell_k\text{-}dist(f) = \Theta\left((\log n)^{1/k}\right)$. Therefore, for any $k^- \leq q \leq k^+$ it holds that $\ell_q\text{-}dist(f) = O(\ell_k\text{-}dist(f))$.

5. In this range it holds that $D(q) = \Theta\left(\frac{n^{\frac{2}{k}-\frac{2}{q}}}{(q-k)^{1/q}} + \sqrt{q/k}\right)$. It can easily be verified that $\sqrt{q/k}$ term (which is basically an $\ell_q\text{-}expans(f)$) is negligible compared to the $\ell_q\text{-}contr(f)$ (the first term). We divide this case into the two following sub-cases.

**Range $k^+ \leq q \leq k^{++}$.** Let $k^{++} = k\left(1 + \frac{1}{\log\log n}\right)$. In this range we do not sample on new points. We note that for $q$ in the range, by the standard norm relations, it holds that $\ell_q\text{-}dist(f) \leq n^{\frac{2}{k}-\frac{2}{q}}\ell_k\text{-}dist(f) = O\left(n^{\frac{2}{k}-\frac{2}{q}}(\log n)^{1/k}\right)$. Thus, for this range we lose a factor of $\left((\log n)^{1/k}(q-k)^{1/q}\right)$ in distortion (with respect to the upper bound for a given $q$, $D(q)$).

**Range $k + \frac{k^2}{\log\log n} \leq q \leq \log n$.** For this range we define the following set $Q$: $q_1 = k\left(1 + \frac{k}{\log\log n}\right)$, for all $j \geq 1$, $q_{(j+1)} = q_j\left(1 + \frac{q_j}{\log n}\right)$. Note that for this range we have that $j \leq 2\log n$. The function $t$ is defined by $t(q) = (20\log n)^{1/q}$, for all $q \geq 1$. For the first property we estimate

$$\sum_{1 \leq j \leq 2\log n} \frac{1}{(t(q_j))^{q_j}} = \sum_{1 \leq j \leq 2\log n} \frac{1}{20\log n} = 1/10,$$

as required. For the second property, we have for all $q_{j-1} \leq q \leq q_j$ that

$$\frac{2}{q} - \frac{2}{q_j} = \frac{2(q_j - q)}{qq_j} \leq \frac{q_j - q_{(j-1)}}{q_{(j-1)}^2} \leq \frac{2}{\log n}.$$

Therefore,

$$\frac{D(q_j)}{D(q)} = n^{\frac{2}{q}-\frac{2}{q_j}}\frac{(q-k)^{1/q}}{(q_j-k)^{1/q_j}} = \Theta\left(\frac{(q-k)^{1/q}}{(q_j-k)^{1/q_j}}\right) =$$

$$\Theta\left(\left(\frac{q-k}{q_j-k}\right)^{1/q_j}(q-k)^{\frac{1}{q}-\frac{1}{q_j}}\right) = \Theta(1),$$

as required.

$\square$

### G.3 Tightness of the analysis for small $q$

In this section we show that the computations we made in the analysis of the moments of the JL transform are tight (for $q < \tilde{c}k$). In fact, we show a stronger result: the normalization (by any positive factor) of the JL transform does not help to improve the upper bounds. Also, it is not hard to see that for the equilateral space, the $\ell_q$-distortion is highly concentrated around its expected value.

**Claim 56.** *Given any $n$-point $X \subset \ell_2^d$ and any integer $k \geq 2$, let $f : X \to \ell_2^k$ be the JL transform into dimension $k$. For any $\alpha > 0$: $E\left[(\ell_1\text{-}dist(\alpha \cdot f))\right] \geq 1 + \frac{c'}{\sqrt{k}}$, for a positive constant $c'$.*

*Proof.* We present a sketch of the proof. Following the lines of the moment analysis for $q < k$, we have

$$E\left[(\ell_1\text{-}dist(\alpha \cdot f))\right] = \left(1 + \frac{1}{k-1}\right)\frac{\Gamma\left(\frac{k}{2} + \frac{1}{2}\right)}{\left(\frac{k}{2}\right)\Gamma\left(\frac{k}{2}\right)^{\frac{1}{2}}} \cdot \frac{1}{\sqrt{\alpha}} + \sqrt{\alpha} \cdot \frac{\Gamma\left(\frac{k}{2} + \frac{1}{2}, \frac{k}{2\alpha}\right)}{\left(\frac{k}{2}\right)^{\frac{1}{2}}\Gamma\left(\frac{k}{2}\right)} - \frac{1}{\sqrt{\alpha}} \cdot \frac{\Gamma\left(\frac{k}{2} - \frac{1}{2}, \frac{k}{2\alpha}\right)\left(\frac{k}{2}\right)^{\frac{1}{2}}}{\Gamma\left(\frac{k}{2}\right)}.$$

Using the lower bound of (75), and applying the recursive rule of the incomplete gamma function, we obtain that the expected value is bounded from below by

$$\left(1 - \frac{1}{k}\right) \cdot \left(1 - \frac{1}{k+1}\right)^{\frac{1}{2}} \cdot \frac{1}{\sqrt{\alpha}} + \frac{\Gamma\left(\frac{k}{2} - \frac{1}{2}, \frac{k}{2\alpha}\right)}{\Gamma\left(\frac{k}{2}\right)} \cdot \left(\frac{k}{2} - \frac{1}{2}\right)^{\frac{1}{2}} \cdot \left(\left(\frac{k-1}{k}\alpha\right)^{\frac{1}{2}} - \left(\frac{k}{k-1}\frac{1}{\alpha}\right)^{\frac{1}{2}}\right) + \frac{\left(\frac{k}{2}\right)^{\frac{k}{2}-1}}{\alpha^{\frac{k}{2}-1}e^{\frac{k}{2\alpha}}\Gamma\left(\frac{k}{2}\right)}.$$

Applying the inequality 3, and the inequality $\Gamma(y) \leq \left(y - \frac{1}{2}\right)^{\frac{1}{2}}\Gamma\left(y - \frac{1}{2}\right)$, for all $y \geq 1$, which follows from estimation developed in (75), we obtain

$$E\left[(\ell_1\text{-}dist(\alpha \cdot f))\right] \geq \left(1 - \frac{1}{k}\right) \cdot \left(1 - \frac{1}{k+1}\right)^{\frac{1}{2}} \cdot \frac{1}{\sqrt{\alpha}} + \frac{\Gamma\left(\frac{k}{2} - \frac{1}{2}, \frac{k}{2\alpha}\right)}{\Gamma\left(\frac{k}{2}\right)} \cdot \left(\left(\frac{(k-1)\alpha}{k}\right)^{\frac{1}{2}} - \left(\frac{k}{(k-1)\alpha}\right)^{\frac{1}{2}}\right) + e^{\frac{k}{2}\left(1 - \frac{1}{\alpha} + \ln\left(\frac{1}{\alpha}\right)\right)}$$

Denote $\beta = \left(\frac{k-1}{k} \cdot \alpha\right)^{\frac{1}{2}}$. Then, the lower bound is

$$\left(1 - \frac{1}{k}\right) \cdot \frac{1}{\beta} + \frac{\Gamma\left(\frac{k-1}{2}, \frac{k-1}{2} \cdot \frac{1}{\beta^2}\right)}{\Gamma\left(\frac{k-1}{2}\right)} \cdot \left(\beta - \frac{1}{\beta}\right) + e^{\frac{k}{2}\left(1 - \left(\frac{k-1}{k}\right)\cdot\frac{1}{\beta^2} + \ln\left(\frac{k-1}{k}\cdot\frac{1}{\beta^2}\right)\right)} \cdot \frac{1}{\sqrt{k}}. \quad (11)$$

Denote $\frac{1}{\beta} = 1 + \frac{t}{\sqrt{k}}$, where values of $t$ are divided into two cases: $0 \leq t < \infty$, which is equivalent to $0 < \beta \leq 1$, and $-\sqrt{k} < t \leq 0$, which is equivalent to $1 \leq \beta < \infty$. In what follows we denote by $\beta(t)$ the value of $\beta$ for a chosen value of $t$ (which is indeed a function of $t$). Denote $G(k,t) = \left(1 - \frac{1}{k}\right) \cdot \frac{1}{\beta(t)} + \frac{\Gamma\left(\frac{k-1}{2}, \frac{k-1}{2} \cdot \frac{1}{(\beta(t))^2}\right)}{\Gamma\left(\frac{k-1}{2}\right)} \cdot \left(\beta(t) - \frac{1}{\beta(t)}\right)$, and $F(k,t) = e^{\frac{k}{2}\left(1 - \left(\frac{k-1}{k}\right)\cdot\frac{1}{(\beta(t))^2} + \ln\left(\frac{k-1}{k}\cdot\frac{1}{(\beta(t))^2}\right)\right)} \cdot \frac{1}{\sqrt{k}}$.

The idea is to show that there exists a value $-\sqrt{k} < t_0 < \infty$ such that the following two statements hold.

1. For all $-t_0 \leq t \leq t_0$ it holds that $F(k,t) \geq \frac{c'}{\sqrt{k}}$, for a positive $c'$, and it holds that $G(k,t) \geq 1 - \frac{c''}{k}$, for a positive $c''$.

2. For all $t \geq t_0$, and $t \leq -t_0$ it holds that $G(k,t) \geq 1 + \frac{\hat{c}}{\sqrt{k}}$, for a positive $\hat{c}$.

We start with the proving the first item. Substituting $1 + \frac{t}{\sqrt{k}}$ in place of $\frac{1}{\beta}$, and using the following standard inequality $\ln(1+x) \geq x - x^2/2$, for $x \in (-1, 1]$ (i.e. assuming $-\sqrt{k} < t \leq \sqrt{k}$) we arrive at

$$F(k,t) \geq e^{-3k\left(\frac{t}{\sqrt{k}}\right)^2} \cdot \frac{1}{\sqrt{k}} = e^{-3t^2}\frac{1}{\sqrt{k}}.$$

Therefore, choosing any constant $-\sqrt{k} \le t_0 \le \sqrt{k}$ implies $F(k,t) \ge e^{-3t_0^2} \frac{1}{\sqrt{k}}$, for all $-t_0 \le t \le t_0$.

It remains to show that there exists $t_0$ such that for all $-t_0 \le t \le t_0$ it holds that $G(k,t) \ge 1 - \frac{c''}{k}$, for a positive $c''$. For all $-t_0 \le t \le t_0$ it holds that

$$G(k,t) \ge \left(1 - \frac{1}{k}\right) \cdot \left(1 + \frac{t}{\sqrt{k}}\right) - \frac{2t}{\sqrt{k}} \cdot \frac{\Gamma\left(\frac{k-1}{2}, \frac{k-1}{2}\right)}{\Gamma\left(\frac{k-1}{2}\right)} = \left(1 - \frac{1}{k}\right) + \frac{t}{\sqrt{k}}\left(1 - \frac{1}{k} - 2\frac{\Gamma\left(\frac{k-1}{2}, \frac{k-1}{2}\right)}{\Gamma\left(\frac{k-1}{2}\right)}\right).$$

We divide our estimations into the following two cases. For $-t_0 \le t \le 0$, it is enough to show that $\frac{\Gamma\left(\frac{k-1}{2}, \frac{k-1}{2}\right)}{\Gamma\left(\frac{k-1}{2}\right)} \ge 1/2 - \frac{\tilde{c}}{\sqrt{k}}$, for some positive constant $\tilde{c}$, from what will follow that $G(k,t) \ge 1 - c^*/k$, for some positive $c^*$. For $0 \le t \le t_0$, it is enough to show that $\frac{\Gamma\left(\frac{k-1}{2}, \frac{k-1}{2}\right)}{\Gamma\left(\frac{k-1}{2}\right)} \le \frac{1}{2} + \frac{\hat{c}}{\sqrt{k}}$, from what will follow that there exists a constant $c''$ such that $G(k,t) \ge 1 - \frac{c''}{k}$. Thus, we have to show that $\frac{1}{2} - \frac{\tilde{c}}{\sqrt{k}} \le \frac{\Gamma\left(\frac{k-1}{2}, \frac{k-1}{2}\right)}{\Gamma\left(\frac{k-1}{2}\right)} \le \frac{1}{2} + \frac{\hat{c}}{\sqrt{k}}$. In (13) it is shown that for all $x \ge 1$ (recall that $k \ge 2$) it holds that

$$\frac{1}{2} - \frac{1}{2\sqrt{x}} \le \frac{\Gamma(x,x)}{\Gamma(x)} \le \frac{1}{2} + \frac{1}{\sqrt{x}}.$$

For the second item, we have for all $t \ge 0$, it holds that

$$G(k,t) \ge \left(1 - \frac{1}{k}\right) \cdot \left(1 + \frac{t}{\sqrt{k}}\right) - \frac{2t}{\sqrt{k}} \cdot \frac{\Gamma\left(\frac{k-1}{2}, \frac{k-1}{2} + 2t\sqrt{k-1}\right)}{\Gamma\left(\frac{k-1}{2}\right)}.$$

For all $t \le 0$, it holds that

$$G(k,t) \ge \left(1 - \frac{1}{k}\right) \cdot \left(1 - \frac{t}{\sqrt{k}}\right) + \frac{2t}{\sqrt{k}} \cdot \frac{\Gamma\left(\frac{k-1}{2}, \frac{k-1}{2} - \frac{t}{4}\sqrt{k-1}\right)}{\Gamma\left(\frac{k-1}{2}\right)}.$$

Therefore, to finish the proof we prove the following two estimations. The first states that

$$\forall t \ge 0, \quad \frac{\Gamma\left(\frac{k-1}{2}, \frac{k-1}{2} + 2t\sqrt{k-1}\right)}{\Gamma\left(\frac{k-1}{2}\right)} \le \frac{1}{e^{\frac{t^2}{16}}}. \tag{12}$$

Again, note that this formula describes $Pr\left[X \ge k - 1 + 4t\sqrt{k-1}\right]$, for $X \sim \chi^2_{k-1}$, therefore, using the bounds developed in (33), we obtain the stated inequality.

The second bound states that

$$\forall \; -\sqrt{k} \le t \le 0, \quad \frac{\Gamma\left(\frac{k-1}{2}, \frac{k-1}{2} - \frac{t}{4}\sqrt{k-1}\right)}{\Gamma\left(\frac{k-1}{2}\right)} \ge 0.65. \tag{13}$$

Note that this formula exactly describes $Pr\left[X \ge k - 1 - \frac{t}{2}\sqrt{k-1}\right]$, for $X \sim \chi^2_{k-1}$ therefore, it is enough to bound from above the $Pr\left[X \le k - 1 - \frac{t}{2}\sqrt{k-1}\right]$. Using the estimations of (33) we get that this probability is bounded from above by $\frac{1}{e^{t^2/32}}$. Therefore, choosing $t_0 = 10$ we get that the equation (13) is bounded from below by $0.8$. Note that the estimations of the first item are correct for such $t_0$, what finishes the proof of the claim.

$\square$

## G.4 $\ell_q$-norm analysis of the JL transform

In this section we derive bounds on distortion of $\ell_q$-Norm of JL transform.

**Theorem 57.** *Given any $n$-point set $X \subset \ell_2^d$, any integer $k \ge 1$, any $q < k$ and any distribution $\Pi$ over $\binom{X}{2}$, with constant probability the JL transform $f : X \to \ell_2^k$ is such that $distnorm_q^{(\Pi)}(f) = 1 + O\left(\frac{1}{\sqrt{k}} + \frac{q}{k-q}\right)$, for $q < \tilde{c}k$, and $distnorm_q^{(\Pi)}(f) = O\left(\sqrt{\frac{q}{k}}\right)$, for $q \ge \tilde{c}k$.*

*Proof.* For $q \leq \tilde{c}k$ the proof follows as a corollary of Claim 15 (the last inequality) and of Corollary 5. For $q \geq \tilde{c}k$ we use the equality part of Claim 15. The idea is to show that with constant probability it holds that both $\ell_q\text{-}expans^{(\Pi')}(f) = O\left(\sqrt{\frac{q}{k}}\right)$, and $\ell_q\text{-}expans^{(\Pi')}(f) \geq const$, for some positive $const$. The upper bound follows from Lemma 44, while the lower bound is analyzed in the following lemma. The proof of the theorem then follows by Markov's inequality applied on the upper and lower bounds. $\qquad\square$

**Lemma 58.** *Given any $n$-point $X \subset \ell_2^d$, any integer $k \geq 1$, any $q \geq 1$, and $\Pi$, the JL transform of dimension $k$, $f : X \to \ell_2^k$, is such that with probability at least $1/2$, $\ell_q\text{-}expans^{(\Pi)}(f) \geq 1/16$.*

*Proof.* The proof follows from Lemma 10: the random variables $X_i$ in the lemma are random variables $(expans_f(i,j))^q$, the random variable $Y = \left(\ell_q\text{-}expans^{(\Pi')}(f)\right)^q$. From Lemma 52 we derive that for all $i \neq j$,

$$p_{ij} = Pr\left[(expans_f(i,j))^q \leq \frac{1}{C^{q/2}}\right] \leq \left(\frac{e}{C}\right)^{k/2} \leq \frac{1}{4^k},$$

for taking $C = 16e$. Therefore, taking $\beta = 1/2$, by Lemma 10 we obtain that with probability at least $1 - \frac{1}{2^{(2k-1)}}$ it holds that $\left(\ell_q\text{-}expans^{(\Pi')}(f)\right)^q \geq \frac{1}{2(16e)^{q/2}}$, implying $\ell_q\text{-}expans^{(\Pi')}(f) \geq 1/16$. $\quad\square$

## G.5  Additive distortion measures analysis of the JL transform

**Theorem 59.** *For a finite $X \subset \ell_2^d$ and integer $k \geq 2$, let $f : X \to \ell_2^k$ be the JL transform of dimension $k$. For a distribution $\Pi$ over $\binom{X}{2}$, with constant probability, for all $1 \leq r \leq q \leq k - 1$:*

$$REM_q^{(\Pi)}(f), Energy_q^{(\Pi)}(f), \Phi_{\sigma,q,r}^{(\Pi)}(f), Stress_q^{(\Pi)}(f), Stress^{*(\Pi)}_q(f) = O\left(\sqrt{q/k}\right).$$

*Proof.* We first show the bound for the $REM_q$, which immediately implies the upper bounds on $Energy_q$ and $Stress_q$. The bounds on $Stress_q^*$, for all $q$, follow by Claim 21 and Lemma 58. After that, we show the bound on $\sigma$-distortion.

We first prove the theorem holds in expectation for any given value of $q \leq k - 1$, and then show how to conclude the simultaneous guarantees. We analyze the expected value of $\left(REM_q^{(\Pi)}(f)\right)^q$, for a given $1 \leq q \leq k - 1$. Following the lines of considerations of the proof we have

$$E\left[\left(REM_q^{(\Pi)}(f)\right)^q\right] = \int_0^k \left(\sqrt{\frac{k}{x}} - 1\right)^q \frac{x^{\frac{k}{2}-1}}{2^{\frac{k}{2}}\Gamma\left(\frac{k}{2}\right)e^{\frac{x}{2}}}\,dx + \int_k^\infty \left(\sqrt{\frac{x}{k}} - 1\right)^q \frac{x^{\frac{k}{2}-1}}{2^{\frac{k}{2}}\Gamma\left(\frac{k}{2}\right)e^{\frac{x}{2}}}\,dx.$$

(14)

We treat two integrals separately. We start with the second integral. Changing variables $\frac{x}{k} = z^2$ results in

$$\int_k^\infty \left(\sqrt{\frac{x}{k}} - 1\right)^q \frac{x^{\frac{k}{2}-1}}{2^{\frac{k}{2}}\Gamma\left(\frac{k}{2}\right)e^{\frac{x}{2}}}\,dx = 2\int_1^\infty \frac{(z-1)^q z^{k-1}}{e^{\frac{z^2 k}{2}}}\frac{\left(\frac{k}{2}\right)^{k/2}}{\Gamma\left(\frac{k}{2}\right)}\,dz.$$

Additional changing of variables $z - 1 = t$ results in

$$\frac{2\left(\frac{k}{2}\right)^{k/2}}{\Gamma\left(\frac{k}{2}\right)e^{\frac{k}{2}}}\int_0^\infty \frac{t^q(t+1)^{k-1}}{e^{k\left(t+\frac{t^2}{2}\right)}}\,dt.$$

Using the inequality $(1 + t) \leq e^t$ for all real $t$, we have that the above is at most

$$\frac{2\left(\frac{k}{2}\right)^{k/2}}{\Gamma\left(\frac{k}{2}\right)e^{\frac{k}{2}}}\int_0^\infty t^q e^{-t-\frac{kt^2}{2}}\,dt \leq \frac{2\left(\frac{k}{2}\right)^{k/2}}{\Gamma\left(\frac{k}{2}\right)e^{\frac{k}{2}}}\int_0^\infty t^q e^{-\frac{kt^2}{2}}\,dt.$$

Changing variables $u = \frac{t^2 k}{2}$ results in

$$\frac{\left(\frac{k}{2}\right)^{k/2}}{\Gamma\left(\frac{k}{2}\right)e^{\frac{k}{2}}}\left(\frac{2}{k}\right)^{\frac{q+1}{2}}\int_0^\infty \frac{u^{\frac{q+1}{2}-1}}{e^u}\,du = \frac{\left(\frac{k}{2}\right)^{k/2}}{\Gamma\left(\frac{k}{2}\right)e^{\frac{k}{2}}}\left(\frac{2}{k}\right)^{\frac{q+1}{2}}\Gamma\left(\frac{q+1}{2}\right).$$

Using the inequalities of Lemma 11 and Lemma 12 we obtain that the second integral is bounded by

$$\int_k^\infty \left(\sqrt{\frac{x}{k}}-1\right)^q \frac{x^{\frac{k}{2}-1}}{2^{\frac{k}{2}}\Gamma\left(\frac{k}{2}\right)e^{\frac{x}{2}}}\, dx \le \left(\frac{1}{e}\right)^{\frac{q}{2}}\left(\frac{q+1}{k}\right)^{\frac{q}{2}}.$$

Next we bound the first integral of (14). Changing variables $z^2 = \frac{x}{k}$ we get

$$\int_0^k \left(\sqrt{\frac{k}{x}}-1\right)^q \frac{x^{\frac{k}{2}-1}}{2^{\frac{k}{2}}\Gamma\left(\frac{k}{2}\right)e^{\frac{x}{2}}}\, dx = \frac{\left(\frac{k}{2}\right)^{\frac{k}{2}}}{\Gamma\left(\frac{k}{2}\right)}\int_0^1 \frac{(1-z)^q z^{k-1-q}}{e^{\frac{k}{2}z^2}}\, dz.$$

Additional changing of variables $1-z=t$ results in

$$\frac{\left(\frac{k}{2}\right)^{\frac{k}{2}}}{\Gamma\left(\frac{k}{2}\right)e^{\frac{k}{2}}}\int_0^1 \frac{t^q(1-t)^{k-1-q}}{e^{-kt+\frac{kt^2}{2}}}\, dt.$$

Since $q \le k-1$, $k-1-q \ge 0$, therefore, using the inequality $1-t \le e^{-t}$ we get that the above is bounded by

$$\frac{\left(\frac{k}{2}\right)^{\frac{k}{2}}}{\Gamma\left(\frac{k}{2}\right)e^{\frac{k}{2}}}\int_0^1 t^q e^{t(q+1)-\frac{kt^2}{2}}\, dt \le \frac{e^{(q+1)}\left(\frac{k}{2}\right)^{\frac{k}{2}}}{\Gamma\left(\frac{k}{2}\right)e^{\frac{k}{2}}}\int_0^1 t^q e^{-\frac{kt^2}{2}}\, dt \le \frac{e^{(q+1)}\left(\frac{k}{2}\right)^{\frac{k}{2}}}{\Gamma\left(\frac{k}{2}\right)e^{\frac{k}{2}}}\int_0^\infty t^q e^{-\frac{kt^2}{2}}\, dt.$$

Therefore, by the estimations we made for the second integral we obtain that

$$\int_0^k \left(\sqrt{\frac{k}{x}}-1\right)^q \frac{x^{\frac{k}{2}-1}}{2^{\frac{k}{2}}\Gamma\left(\frac{k}{2}\right)e^{\frac{x}{2}}}\, dx \le \frac{1}{2}\cdot e^{q+1}\left(\frac{1}{e}\right)^{\frac{q}{2}}\left(\frac{q+1}{k}\right)^{q/2}.$$

Now, by Markov's inequality, for any integer $q \ge 1$ it holds that

$$Pr\left[\left(REM_q^{(\Pi)}(f)\right)^q \ge 4^q \cdot E\left[REM_q^{(\Pi)}(f)^q\right]\right] \le \frac{1}{4^q}.$$

Therefore, by the union bound, we obtain that with probability at least $\frac{2}{3}$, for all $1 \le q \le k-1$ it holds that $REM_q^{(\Pi)}(f) = O\left(\sqrt{\frac{q}{k}}\right)$, which completes the proof.

We turn to proof the bound on the $\sigma$-distortion of the JL transform. By Corollary 4, Claim 44 and by the above proof, with constant probability the JL transform $f$ into $k$ dimensions is such that $\ell_r\text{-}contr(f) = 1+O\left(\frac{r}{k-r}\right)$, $\ell_r\text{-}expans(f) = 1+O\left(\frac{r}{k}\right)$ and such that $Energy_q^{(\Pi)}(f) = O\left(\sqrt{\frac{q}{k}}\right)$, for any given $r \le q$, and $q \le k-1$. Therefore, by Claim 22, with constant probability:

$$\Phi_{\sigma,q,r}^{(\Pi)}(f) = O\left(Energy_q^{(\Pi)}(f)\right) + O\left(\frac{r}{k-r}\right) = O\left(\sqrt{\frac{q}{k}}\right),$$

which completes the proof. $\qquad\square$

# H   Lower bounds

The following basic lemma turns to be useful in this paper.

**Lemma 60.** *Let $(X,d_X)$ be an $n$-point metric space, and $(Y,d_Y)$ be any metric space. Let $2 \le s < n$ be an integer, and let $\mathcal{S} := \{Z \subseteq X \mid |Z| = s\}$. For a given $q \ge 1$ denote $\mathcal{M}_q = \{\ell_q - dist, Energy_q, Stress_q, Stress_q^*, REM_q\}$. If for all $Z \in \mathcal{S}$ it holds that any embedding $f : Z \to Y$ is such that $M_q(f) \ge \alpha$, for a fixed $M_q \in \mathcal{M}_q$, then it holds that for any $F : X \to Y$, $M_q(F) \ge \alpha$.*

*Proof.* Assume by contradiction there is $F : X \to Y$, such that for some $q$ it holds that $M_q(F) < \alpha$. For a pair $x, y \in X$ denote by $\mu_F(x,y)$ the appropriate distortion measure of the pair (according

to the measure $M_q$ under discussion). For a given subset $Z \in \mathcal{S}$ let $F|Z$ denote the embedding $F$ restricted to $Z$. By definition, for any $Z \in \mathcal{S}$

$$(M_q(F|Z))^q = \frac{1}{\binom{s}{2}} \sum_{x \neq y \in Z} (\mu_F(x,y))^q.$$

Therefore,

$$\sum_{Z \in \mathcal{S}} (M_q(F|Z))^q = \frac{1}{\binom{s}{2}} \left( \binom{n-2}{s-2} \right) \sum_{x \neq y \in X} (\mu_F(x,y))^q,$$

implying

$$\frac{1}{\binom{n}{s}} \sum_{Z \in \mathcal{S}} (M_q(F|Z))^q = \frac{\binom{n-2}{s-2}\binom{n}{2}}{\binom{s}{2}\binom{n}{s}} (M_q(F))^q = (M_q(F))^q.$$

Therefore, there might be at least one $Z \in \mathcal{S}$ such that $M_q(F|Z)^q < (\alpha)^q$, a contradiction.

$\square$

## H.1 Additive distortion measures

The following two lemmas were proved by (8):

**Lemma 61.** *For $A = (\hat{\epsilon}_{ij})$, a $t \times t$ real, symmetric matrix, with all $1$ on diagonal, $\sum_{1 \leq i < j \leq t} \hat{\epsilon}_{ij}^2 \geq \frac{t^2}{2 \cdot rank(A)} - \frac{t}{2}$.*

**Lemma 62.** *For $A = (a_{ij})$, a $t \times t$ real matrix, and for an integer $q \geq 1$ define $A^q$ a $t \times t$ matrix, by $(A^q)_{ij} = a_{ij}^q$. Then $rank(A^q) \leq \binom{q+rank(A)-1}{q}$.*

We first show a lower bound for even values of $q$, which in the next claim will be extended to hold for all values of $q$.

**Claim 63.** *Let $E_n$ denote an $n$-point equilateral space. For any $k \geq 2$, any even $2 \leq q \leq k$, let $n_q = \lceil 4\,(9k/q)^{q/2} \rceil$. For all $n \geq n_q$, for any $f : E_n \to \ell_2^k$, $Energy_q(f) \geq \frac{1}{250}\sqrt{q/k}$.*

*Proof.* By the argument in Lemma 60 it is enough to consider $n = n_q$. For $f : E_{n_q} \to \ell_2^k$, denote $\epsilon_{ij} = |expans_f(x_i, x_j) - 1|$, for all $1 \leq i < j \leq n_q$, so $(Energy_q(f))^q = \sum_{i<j} \epsilon_{ij}^q / \binom{n_q}{2}$. If there are more than $n_q/4$ pairs of points $x_i \neq x_j \in E_{n_q}$ with $dist_f(x_i, x_j) > 2$, then $(Energy_q(f))^q \geq \frac{n_q/4}{\binom{n_q}{2}} \cdot \frac{1}{2^q} \geq \frac{1}{250^q}\left(\frac{q}{k}\right)^{q/2}$,

and we are done. Otherwise, there are at most $n_q/4$ pairs with $dist_f(x_i, x_j) > 2$, implying there is $X \subseteq E_{n_q}$, on $n_q' \geq n_q/2$ points, such that for all $x \neq y \in X$, $dist_f(x,y) \leq 2$. Let $f|X$ denote the embedding $f$ restricted to $X$. Then $(Energy_q(f|X))^q \leq 5(Energy_q(f))^q$, meaning it is enough to lower bound the embedding $f|X$. Denote $X = \{x_i \mid 0 \leq i \leq n_q' - 1\}$, and $V = \{v_i \mid v_i = f(x_i), \forall x_i \in X\}$, w.l.o.g. $v_0 = 0$. For all $x_i \in X$ define $w_q(x_i) = \sum_{j \neq i} (\epsilon_{ij})^q$. W.l.o.g. $x_0$ is such that $w_q(x_0) = \min\{w_q(x_i) \mid x_i \in X\}$. Define a matrix $D_V$ by $D_V[i,j] = \langle v_i/\|v_i\|_2, v_j/\|v_j\|_2 \rangle$, for $0 \leq i \neq j \leq n_q' - 1$. Then $rank(D_V) \leq k$, $D_V[i,i] = 1$ and $\forall\, i \neq j$: $D_V[i,j] = \cos(\alpha_{ij}) = (\|v_i\|_2^2 + \|v_j\|_2^2 - \|v_i - v_j\|_2^2)/(2\|v_i\|_2\|v_j\|_2)$. For all $j$ it holds $\|v_j\|_2 = \|v_j - v_0\|_2$. Since $\epsilon_{ij} = |\,\|v_i - v_j\|_2 - 1|$ and for all $x_i \neq x_j \in X$, $dist_f(x_i, x_j) \leq 2$, it holds that $\epsilon_{ij} \leq 1$. It follows that for each $i,j$: either $dist_f(x_i, x_j) = expans_f(x_i, x_j)$, and thus $\|v_i - v_j\| = 1 + \epsilon_{ij} \leq 2$, or $dist_f(x_i, x_j) = contract_f(x_i, x_j)$, and thus $\|v_i - v_j\| = 1 - \epsilon_{ij} \geq 1/2$. Therefore, it holds that: $\frac{1}{2} - 3(\epsilon_{0i} + \epsilon_{0j} + \epsilon_{ij}) \leq D_V[i,j] \leq \frac{1}{2} + 3(\epsilon_{0i} + \epsilon_{0j} + \epsilon_{ij})$.

Let $A = 2D_V - J$, where $J$ is all $1$ matrix. Then $A$ is symmetric, $A[i,i] = 1$ and $\forall i \neq j, |A[i,j]| \leq 6(\epsilon_{0i} + \epsilon_{0j} + \epsilon_{ij})$, and $rank(A) \leq rank(D_V) + 1 \leq k + 1$. Applying Lemma 62 we get that $rank\left(A^{q/2}\right) \leq n_q'/2$.

Therefore, applying Lemma 61 on the matrix $A^{q/2}$: $\sum_{1 \leq i < j \leq n_q' - 1} 6^q(\epsilon_{0i} + \epsilon_{0j} + \epsilon_{ij})^q \geq$

$$\frac{n_q'^2}{2 \cdot rank(A^{q/2})} - \frac{n_q'}{2} \geq \frac{1}{4} \cdot \frac{n_q'^2}{rank(A^{q/2})} \geq \frac{1}{4} \cdot \frac{n_q'^2}{e^{q/2}} \cdot \left(\frac{q}{q+2k}\right)^{q/2}.$$

On the other hand, since $x_0$ minimizes $w_q(x_i)$ over all $x_i \in X$, we have for $1 \le i, j \le n'_q - 1$:

$$\sum_{i<j} (\epsilon_{0j} + \epsilon_{0i} + \epsilon_{ij})^q \le 2^{q-1} \sum_{i<j} (\epsilon_{0i}^q + \epsilon_{0j}^q + \epsilon_{ij}^q) \le 2^q(n'_q - 1) \sum_j \epsilon_{0j}^q + 2^{q-1} \sum_{i<j} \epsilon_{ij}^q \le$$

$3 \cdot 2^q \sum_{0 \le i < j \le n'_q - 1} \epsilon_{ij}^q$. Therefore, $\sum_{1 \le i < j \le n'_q - 1} 6^q (\epsilon_{0j} + \epsilon_{0i} + \epsilon_{ij})^q \le 3 \cdot 12^q \sum_{0 \le i < j \le n'_q - 1} \epsilon_{ij}^q =$

$3 \cdot 12^q \binom{n'_q}{2} (Energy_q(f|X))^q \le \frac{15 \cdot 12^q}{2} \cdot (n'_q)^2 \cdot (Energy_q(f))^q$. Putting all together we obtain $(Energy_q(f))^q \ge \frac{1}{30 \cdot 36^q} \cdot \left(\frac{q}{k}\right)^{q/2}$, completing the proof.

$\square$

The following claim extends the lower bound for all values of $q \le k$, from the case of lower bound for even values of $q$.

**Claim 64.** *For all $k \ge 2$, $k \ge q \ge 2$, and $n \ge 4 \left(9 \cdot \frac{k}{q}\right)^{q/2}$, for all $f : E_n \to \ell_2^k$: $Energy_q(f) \ge \frac{1}{400} \sqrt{\frac{q}{k}}$.*

*Proof.* If $q$ is even, we are done by Claim 63. Otherwise, let $q' \ge 2$ be the largest even integer such that $q' \le q$. It is easily seen that $N_q \ge N'_q$, for all $2 \le q' \le q \le k$. Therefore, by Claim 63, it holds that for all $n \ge N_q \ge N'_q$

$$Energy_q(f) \ge Energy_{q'}(f) \ge \frac{1}{250} \sqrt{\frac{q'}{k}} \ge \frac{1}{250 \cdot \sqrt{2}} \sqrt{\frac{q}{k}},$$

where the first inequality is due to the monotonicity of Energy measure, and the last inequality due to the fact that $q/q' \le 2$. This finishes the proof of the claim. $\square$

**Corollary 6.** *For all $k \ge 2$, $k \ge q \ge 2$, $r \le q$ and $n \ge 4 \left(9 \cdot \frac{k}{q}\right)^{q/2}$, for all embeddings $f : E_n \to \ell_2^k$, it holds that $\Phi_{\sigma,q,r}(f) \ge \frac{1}{400} \sqrt{\frac{q}{k}}$.*

*Proof.* Let $f : E_n \to \ell_2^k$ be any embedding. For $1 \le r \le q$, consider $\ell_r\text{-}expans(f)$. Let $\hat{f} : E_n \to \ell_2^k$ be the embedding defined by: for all $u \in E_n$, $\hat{f}(u) = \frac{f(u)}{\ell_r\text{-}expans(f)}$. Note that $\ell_r\text{-}expans(\hat{f}) = 1$. Then, since $\sigma$-distortion is a scale invariant measure, it holds that

$$\Phi_{\sigma,q,r}(f) = \Phi_{\sigma,q,r}(\hat{f}) = Eenergy_q(\hat{f}) = \Omega\left(\sqrt{\frac{q}{k}}\right),$$

where the last inequality is true by Claim 64. $\square$

**Claim 65.** *For all $k \ge 1$, $1 \le q < 2$, and $n \ge 18k$, for all $f : E_n \to \ell_2^k$ it holds that $Energy_q(f) = \Omega\left(\frac{1}{k^{1/q}}\right)$.*

*Proof.* In Claim 63 take $q = 2$. Then, $\sum_{1 \le i < j \le N'_2 - 1} 6^2 (\epsilon_{0j} + \epsilon_{0i} + \epsilon_{ij})^2 \le 3 \cdot 12^2 \sum_{0 \le i < j \le N'_2 - 1} \epsilon_{ij}^2$.

Note that for $1 \le q < 2$ it holds that $\epsilon_{ij}^2 \le \epsilon_{ij}^q$, for all $1 \le i < j \le N'_2 - 1$, since for these indexes $\epsilon_{ij} \le 1$.

Therefore, following the arguments of Claim 63, for $1 \le q < 2$ we obtain

$$\sum_{1 \le i < j \le N'_2 - 1} 6^2 (\epsilon_{0j} + \epsilon_{0i} + \epsilon_{ij})^2 \le 3 \cdot 12^2 \sum_{0 \le i < j \le N'_2 - 1} \epsilon_{ij}^2 \le 432 \sum_{0 \le i < j \le N'_2 - 1} \epsilon_{ij}^q = O((N'_2)^2) \cdot (Energy_q(f))^q.$$

Also for $q = 2$, we have for $1 \le q < 2$, $n \ge N_2 = 18k$, for all $f : E_n \to \ell_2^k$, $Energy_q(f) = \Omega\left(\frac{1}{k^{1/q}}\right)$. $\square$

## H.2 Moments of distortions

Let us first to derive lower bounds for $q \leq \sqrt{k}$.

**Corollary 7.** *Given an integer $k \geq 1$, and any $1 \leq q \leq \sqrt{k}$, denote $N = 18k$. For any $n \geq N$, if $f : E_n \rightarrow \ell_2^k$ then $\ell_q\text{-}dist(f) = 1 + \Omega\left(\frac{q}{k}\right)$.*

*Proof.* Let $f : E_n \rightarrow \ell_2^k$ be any embedding. For all $x_i \neq x_j \in E_n$, denote $\epsilon_{ij} = |expans_f(x_i, x_j) - 1|$. Note that if $expans_f(x_i, x_j) \geq 1$, then $dist_f(x_i, x_j) = 1 + \epsilon_{ij}$, and otherwise $dist_f(x_i, x_j) = \frac{1}{1-\epsilon_{ij}} \geq 1 + \epsilon_{ij}$. Therefore, we have

$$(\ell_q\text{-}dist(f))^q = \frac{1}{\binom{n}{2}} \sum_{1 \leq i < j \leq n} (dist_f(x_i, x_j))^q \geq \frac{1}{\binom{n}{2}} \sum_{1 \leq i < j \leq n} (1 + \epsilon_{ij})^q.$$

Consider first the case $q \geq 2$. It can be easily shown (by derivation, for example) that for any $q \geq 2$, and for any $x \geq 0$ it holds that $(1 + x)^q \geq 1 + \frac{q(q-1)}{2}x^2$. Therefore, we obtain

$$\frac{1}{\binom{n}{2}} \sum_{1 \leq i < j \leq n} (1 + \epsilon_{ij})^q \geq \frac{1}{\binom{n}{2}} \sum_{1 \leq i < j \leq n} \left(1 + \frac{q(q-1)}{2}\epsilon_{ij}^2\right) = 1 + \frac{q(q-1)}{2}(Energy_2(f))^2.$$

By Claim 64, we have that for some positive constant $0 < c < 1$,

$$1 + \frac{q(q-1)}{2}(Energy_2(f))^2 \geq 1 + c \cdot \frac{q^2}{k}.$$

Therefore, using the inequality: for all $0 \leq x \leq 1$, for all $0 < r \leq 1$, $(1 + x)^r \geq 1 + \frac{1}{2}xr$, for $x = c \cdot \frac{q^2}{k}$, we conclude that

$$\ell_q\text{-}dist(f) \geq 1 + \Omega\left(\frac{q}{k}\right),$$

completing the proof for all $2 \leq q \leq \sqrt{k}$.

For $1 \leq q < 2$ we apply the following inequality: for all $x \geq 0$ and for all $q \geq 1$, $(1+x)^q \geq 1+q \cdot x$. Therefore, applying Claim 65 we obtain that for some constant $0 < c < 1$,

$$(\ell_q\text{-}dist(f))^q \geq \frac{1}{\binom{n}{2}} \sum_{1 \leq i < j \leq n} (1+\epsilon_{ij})^q \geq \frac{1}{\binom{n}{2}} \sum_{1 \leq i < j \leq n} (1+q \cdot \epsilon_{ij}) \geq 1+q \cdot Energy_1(f) \geq 1+c \cdot \frac{q}{k}.$$

Implying $\ell_q\text{-}dist(f) \geq (1 + cq/k)^{\frac{1}{q}} = 1 + \Omega(1/k)$, which completes the proof. $\qquad \square$

**Theorem 66.** *For all $k \geq 16$, for all $N$ large enough, there exists a metric space $Z \subseteq \ell_2$ on $N$ points, such that for any embedding $F : Z \rightarrow \ell_2^k$ it holds that $\ell_q\text{-}dist(F) \geq 1 + \Omega\left(\frac{q}{k-q}\right)$, for all $\Omega\left(\sqrt{k \log k}\right) \leq q \leq \tilde{c}k$.*

*Proof.* The proof is based on a recent breakthrough result of (49) showing tightness of the JL dimension bound (improving upon (8)). Their result states that for all integer $n \geq 2$, there exists an $n$-point Euclidean metric space $S \subset \ell_2^n$, such that for all $\frac{1}{n^{0.4999}} < \epsilon < 1/8$, it holds that any embedding $f : S \rightarrow \ell_2^d$ with distortion $1+\epsilon$ must have $d \geq \frac{c_{LN} \log n}{\epsilon^2}$, for a constant $0 < c_{LN} < 1/64$. In what follows we assume that $q \leq \tilde{c}k$, for $\tilde{c}$ as defined in Remark 46, and the lower bound for the rest of the range follows from $q = \tilde{c}k$.

Given any $k \geq 16$, and $q$ as in the theorem, let $\epsilon = \frac{c_{LN}}{200} \cdot \frac{q}{k-q}$, and let $n = 2^{\left(2k\epsilon^2\right)/c_{LN}}$. For this definition of $n$ we have that $k = \frac{c_{LN} \log n}{2\epsilon^2} < \frac{c_{LN} \log n}{\epsilon^2}$. In addition, note that for the range of $q$ as in the theorem, it holds that $\epsilon > \frac{1}{n^{0.4999}}$. Moreover, from the definitions of $\epsilon$ and $n$ it holds that $q = 100 \cdot \frac{\log n}{\epsilon} \cdot \frac{k-q}{k}$. Since for all $q$ it holds that $\frac{k-q}{k} < 1$, we have that $q \leq 100 \cdot \frac{\log n}{\epsilon}$. In addition, since $q \leq \tilde{c}k$ it holds that $\frac{k-q}{k} \geq 1 - \tilde{c}$, recalling that $\tilde{c} \leq 0.94$ implies that $\frac{k-q}{k} \geq 3/50$. Therefore, $q \geq 6 \cdot \frac{\log n}{\epsilon}$.

Now we are ready to use the result of Larsen and Nelson. For the above chosen parameters $\epsilon, n, k$, there exists an $n$-point metric space $S \subset \ell_2$, such that for any embedding $f : S \to \ell_2^k$ it holds that $dist(f) \geq 1 + \epsilon$. Therefore, applying Theorem 25 on $S$, we obtain that for our $q$, there exists a metric space $Z$ on $N$ points, for all $N > n$, such that for any embedding $F : Z \to \ell_2^k$ it holds

$$\ell_q\text{-}dist(F) \geq \left(1 + 2\left((1 + \epsilon)^{q/2} - 1\right)/n^2\right)^{1/q}.$$

Note that using the Theorem 25 enables us to choose the metric space $T$ (in the notions of the theorem) to be Euclidean, and such that the composition of $S$ and $T$ will be Euclidean metric space as well. Now, using the bounds on $q$ we have established, we conclude $\ell_q\text{-}dist(F) \geq 1 + \left(\frac{c_{LN}}{2400}\right)\left(\frac{q}{k-q}\right)$. This completes the proof.

$\square$

### H.3   Phase transition: lower bound for $q = k$

**Theorem 67.** *Any embedding $f : E_n \to \ell_2^k$ has $\ell_k\text{-}dist(f) = \Omega((\sqrt{\log n})^{1/k}/k^{1/4})$, for any $k \geq 1$.*

By Claim 24, it is enough to prove that for any non-expansive embedding $\ell_k\text{-}dist(f) = \Omega((\log n)^{1/k}/\sqrt{k})$. We use the *hierarchical separated trees (HST)*, that were introduced in (12).

**Definition 9.** *For $s > 1$, a rooted tree $T$, with labels $\Delta(v) \geq 0$ assigned to each node $v \in V(T)$, is an $s$-HST if the leaves of $T$ have $\Delta(v) = 0$, and for $u \neq v \in V(T)$, if $u$ is a child of $v$ then $\Delta(u) \leq \Delta(v)/s$.*

An $s$-HST induces a metric on the set of its leaves: for all $u \neq v$ leaves in $T$, the metric is define by $d_T(u, v) = \Delta(lca_T(u, v))$. For $s > 1$, $t \geq 1$, $\delta > 0$, let $s\text{-}H_t^\delta$ be the set of all $s$-HST trees on $n$ leaves, with degree at most $2^t$, and with $\Delta(root(T)) = \delta$. Throughout the section we set $s = 2$, and remove it from notation, i.e. $H_t^\delta$ denotes $2\text{-}H_t^\delta$. The proof follows from the following claim:

**Claim 68.** *Any non-expansive embedding $g : E_n \to \ell_\infty^k$ has $\ell_k\text{-}dist(g) = \Omega((\log n)^{1/k})$.*

To prove the claim it is enough to prove the following lemmas:

**Lemma 69.** *Let $g : E_n \to \ell_\infty^k$ be any non-expansive embedding, then there exists a non-expansive embedding $h : g(E_n) \to H_k^1$ such that $\ell_k\text{-}dist(g) \geq \ell_k\text{-}dist(h \circ g)$.*

*Proof.* Since $g$ is non-expansive, $g(E_n) \subset [0, 1]^k$. Note that this cube has edge and diagonal lengths 1. Denote $C_l$ the cube with edge and diagonal lengths $l > 0$. We build an embedding $h : g(E_n) \to T$, $T \in H_k^1$, recursively as follows. The root of the tree $T$ is defined to be $r(T)$ with $\Delta(r(T)) = 1$. Divide the unit cube $[0, 1]^k$ into $2^k$ $C_{1/2}$-sub-cubes, denote these cubes $C_{1/2}^{(j)}$, $1 \leq j \leq 2^k$. For all $C_{1/2}^{(j)}$-sub-cube recursively build the sub-trees $T_j$ on the leaves of points in $g(E_n) \cap C_{1/2}^{(j)}$. Let $\Delta(r(T_j)) = 1/2$. Set the (at most $2^k$) children of $r(T)$ to be $T_j$. If for some $j$, $g(E_n) \cap C_{1/2}^{(j)} = \emptyset$, then $T_j$ is an empty tree. If $|g(E_n) \cap C_{1/2}^{(j)}| = 1$, then the tree $T_j$ contains only the root with label 0.

Let $r_s \neq r_t \in g(E_n)$ be any pair of points. Denote $i \geq 1$ the first step in the recursive construction in which $r_t$, $r_s$ are separated. By construction, the $\ell_\infty$ distance between them is at most $\frac{1}{2^{i-1}}$, while their distance on the constructed tree $T$ is exactly $\frac{1}{2^{i-1}}$, meaning $dist_g(r_s, r_t) \geq 2^{i-1}$, while $dist_{(h \circ g)}(r_s, r_t) = 2^{i-1}$. This completes the proof of the lemma.

$\square$

**Lemma 70.** *For any $k \geq 1$, for any non-expansive $F : E_n \to H_k^1$, $\ell_k\text{-}dist(F) = \Omega((\log n)^{1/k})$.*

*Proof.* For a non-expansive $F : E_n \to H_k^1$, let $T \in H_k^1$ be its image tree. Note that the $\ell_k$-distortion of the embedding is solely defined by the topology of $T$. Define the *k-weight* of $T$ by $w_k(T) = (\ell_k\text{-}dist(F))^k \cdot \binom{n}{2}$. For all $k, n \geq 1$, $\delta \leq 1$, let $S_k(n, \delta) = \min\{w_k(T)|T \in H_k^\delta\}$, $S_k(0, \delta) = 0$. Thus, we have to prove that $S_k(n, 1) \geq \Omega(n^2 \log n/k)$, for all $k \geq 1$, which we show by induction on $n$. We prove that for any $k \geq 1$, for all $n \geq 2$, $S_k(n, 1) \geq n^2 \log n/(4k)$. For $n = 2$,

$S_k(2,1) = 1 \geq \frac{1}{k}$, for all $k \geq 1$. Assume the claim holds for all $n' < n$. Let $D = \{0 \leq n_i \leq n, \ 1 \leq i \leq 2^k, \ \sum_i n_i = n\}$ be the set of constraints. Then, $S_k(n,1) = \min_D\{\sum_i S_k(n_i, 1/2) + \sum_{i \neq j} n_i n_j\}$. Note that for all $n$ and $k$, by the definition $S_k(n, 1/2) = 2^k S_k(n,1)$. Substituting this and applying the induction assumption on each $S_k(n_i, 1)$, we get that $S_k(n, 1) \geq \min_D\{\frac{1}{4k} \cdot 2^k \sum_i n_i^2 \log n_i + \sum_{i \neq j} n_i n_j\}$. We can write $\sum_{i \neq j} n_i n_j = \frac{1}{2}((\sum_i n_i)^2 - \sum_i n_i^2) = \frac{n^2}{2} - \frac{1}{2}\sum_i n_i^2$, therefore $S_k(n,1) \geq \min_D\{\sum_i n_i^2(\frac{2^k}{4k}\log n_i - \frac{1}{2}) + \frac{1}{2}n^2\}$. Consider the real function $f(x) = x^2(\frac{2^k}{4k}\log x - \frac{1}{2})$, on $x \geq 2$. Since $f(x)$ is convex for all $k \geq 1$, by Jensen's inequality the minimum value of the above minimization program is obtained at $n_i = n/2^k$, for all $i$. Thus, $S_k(n,1) \geq \frac{n^2 \log(\frac{n}{2^k})}{4k} + (\frac{1}{2} - \frac{1}{2^{k+1}})n^2 = \frac{n^2 \log n}{4k} + (\frac{1}{2} - \frac{1}{4} - \frac{1}{2^{k+1}})n^2 \geq \frac{n^2 \log n}{4k}$, for all $k \geq 1$. $\square$

**Lemma 71.** *Let* $F : E_n \to H_k^1$ *be any non-expansive embedding. Then* $\ell_q\text{-}dist(F) \geq \Omega\left(\left(\frac{1}{k-q}\right)^{1/q}\right)$, *for any* $k - 1 \leq q \leq k^-$.

*Proof.* The proof of this lemma is similar to the proof of Lemma 70 itself. Namely, we show by induction on $n$ that $S_q(n,1) \geq n^2 \cdot \alpha(n)$, where this time $\alpha(n) = \frac{1}{4(1-2^{(q-k)})} \cdot \left(1 - n^{\frac{q-k}{k}}\right)$ (note that in the Lemma 70, $\alpha(n) = \log n$). Let us first estimate the lower bound on $\alpha(n)$. For all $n \geq 2^{\frac{k}{k-q}}$, and $q, k$ such that $q > k - 1$, it holds that

$$\alpha(n) \geq \frac{1}{8 \ln 2(k-q)}.$$

To see this, note that $1 - n^{\frac{q-k}{k}} \geq 1/2$, and $1 - 2^{(q-k)} \leq (k-q)\ln 2$ for the chosen values of $n, q, k$. Also note, that from the above constrain on $n$ it follows that $q \leq k\left(1 - \frac{1}{\log n + 1}\right) = k^-$.

Now we replicate the inductive proof of Lemma 70.

**Inductive Claim.** Given any $k - 1 \leq q \leq k^-$, for all $n \geq 2$, it holds that $S_q(n,1) \geq n^2 \alpha(n)$.

**Base Case:** $n = 2$. $S_q(2,1) = 1 \geq 4 \cdot \frac{1}{4} = 1$.

**Induction's Assumption.** Given any $k - 1 \leq q \leq q^*$, assume that for all $n' < n$, $S_q(n', 1) \geq (n')^2 \alpha(n')$.

**Induction's Step.** We have to prove that $S_q(n,1) \geq n^2 \alpha(n)$. Following the lines of the proof of Lemma 70, it is enough to show that

$$\min_{\substack{0 \leq n_i \leq n, \\ 1 \leq i \leq 2^k, \\ \sum_i n_i = n}} \left\{ 2^q \sum_i n_i^2 \alpha(n_i) + \sum_{1 \leq i < j \leq 2^k} n_i n_j \right\} \geq n^2 \alpha(n),$$

substituting $\alpha(n_i)$ it is equivalent to showing that

$$\min_{\substack{0 \leq n_i \leq n, \\ 1 \leq i \leq 2^k, \\ \sum_i n_i = n}} \left\{ 2^q \sum_i n_i^2 \left(1 - n_i^{\left(\frac{q-k}{k}\right)}\right) + 4\left(1 - 2^{(q-k)}\right) \sum_{1 \leq i < j \leq 2^k} n_i n_j \right\} \geq n^2 \left(1 - n^{\left(\frac{q-k}{k}\right)}\right).$$

Applying similar consideration to this of Lemma 70, we conclude that the minimum is obtained for $n_i = n/2^k$, for all $i$. Therefore, substituting it in the formula and doing some math, we conclude that we have to show that

$$2^{(q-k)}n^2 - n^{\left(\frac{q-k}{k}\right)}n^2 + \left(1 - 2^{(q-k)}\right)n^2 \geq n^2\left(1 - n^{\left(\frac{q-k}{k}\right)}\right),$$

which is true for any $k, q, n \geq 1$. This completes the proof of the lemma, and of the claim. $\square$

**Claim 72.** *Any $f : E_n \to \ell_2^k$ has $\ell_q\text{-}dist(f) \geq \Omega\left(\frac{1}{k^{1/4}}\left(\frac{1}{k-q}\right)^{1/2q}\right)$, for all $n$, $k$ large enough, and for all $k-1 \leq q \leq k^-$.*

The proof of this claim follows the lines of the proof of Theorem 6. Specifically, Lemma 70 is replaced by Lemma 71. The proof follows by induction argument showing that $S_q(n,1) \geq n^2 \cdot \alpha(n)$, where $\alpha(n) = \frac{1}{4(1-2^{(q-k)})} \cdot \left(1 - n^{\frac{q-k}{k}}\right)$ (instead of $\alpha(n) = \log n$).

**Claim 73.** *There exists an $n$-point metric space $X \subset \ell_2$ such that for any embedding $f : X \to \ell_2^k$, for all $k \geq 1$, for all $q > k$ it holds that $\ell_q\text{-}dist(f) = \Omega\left(n^{\left(\frac{1}{2\lceil k/2 \rceil} - \frac{2}{q}\right)}\right)$.*

*Proof.* We base the proof on the worst case lower bound of (59), who showed that there exists an $n$-point metric space $X \subset \ell_2^{k+1}$ such that any embedding of it into $\ell_2^k$ requires distortion at least $\Omega\left(n^{2/k}\right)$ for all even $k \geq 2$, and at least $\Omega\left(n^{2/(k+1)}\right)$, for all odd $k \geq 3$. The proof immediately follows from the standard norm relations: for all $q \geq 1$, $\ell_q\text{-}dist(f) \geq \frac{\ell_\infty\text{-}dist(f)}{n^{2/q}}$. In addition, by the definition $\ell_\infty\text{-}dist(f) \geq \sqrt{dist(f)}$. $\qquad\square$

Another technique applied to the equilateral space provides us with the following lower bounds.

**Claim 74.** *For all $k \geq 1$ and for all $q > k$, any embedding $f$ of an $n$ point equilateral metric space $E_n$ into $\ell_2^k$ has $\ell_q\text{-}dist(f) \geq \Omega\left(n^{\frac{1}{2k} - \frac{1}{2q}}\right)$.*

*Proof.* By Claim 24 it's enough to show a lower bound on non-expansive embeddings. Let $f : E_n \to \ell_2^k$ be any non-expansive embedding, it is enough to show that for any $q > k$ it holds that $\ell_q\text{-}dist(f) \geq \Omega\left(n^{\frac{1}{k} - \frac{1}{q}}\right)$. The idea is to use a volume argument to show that there is a linear number of pairs with small enough distance. Let us formulate this. First, w.l.o.g. assume that $f(E_n)$ resides in the Euclidean unit ball. For any $0 < \epsilon < 1$ the set of points $N \subset \ell_2$ is called $\epsilon$-separated, if for any two points $n_1 \neq n_2 \in N$ it holds that $\|n_1 - n_2\|_2 \geq \epsilon$. The basic volume argument implies for any $\epsilon$-separated set $N$ contained in a unit ball of any in $k$-dimensional normed space:

$$|N| \leq \left(\frac{2}{\epsilon} + 1\right)^k \leq \left(\frac{4}{\epsilon}\right)^k.$$

If for at least $n/2$ points of $f(E_n)$ there exists at least one point $y \in f(E_n)$, such that their distance is at most $\frac{100}{n^{1/k}}$, then there are at least $n/4$ pairs with distance at most $\frac{100}{n^{1/k}}$. Therefore, we have that in such case

$$\ell_q\text{-}dist(f) \geq \left(\frac{n/4 \cdot \left(\frac{n^{1/k}}{100}\right)^q}{n^2}\right)^{1/q} \geq \Omega\left(n^{\frac{1}{k} - \frac{1}{q}}\right).$$

Otherwise, there is a subset of $f(E_n)$ of size at least $n/2$ that is $\frac{100}{n^{1/k}}$- separated. Therefore, we get that $n/2 \leq \left(\frac{1}{25}\right)^k \cdot n$, contradiction. $\qquad\square$

# I    Lower bounds for all $q$ simultaneously

**Claim 75.** *Let $k \geq 1$. For any $n \geq 18k$, let $f : E_n \to \ell_2^k$ be an embedding such that for some $1 \leq q \leq \sqrt{k}$, it holds that $\ell_q\text{-}dist(f) \leq 1 + \frac{c}{q}$, for some $c > 0$. Then, $\ell_1\text{-}dist(f) \geq 1 + \frac{q}{k} \cdot \Omega\left(\frac{1}{c + \ln(k/q^2)}\right)$.*

*Proof.* First, note that for $q = o(\log k)$, the statement in the claim is not interesting, since for such values of $q$ it guarantees a weaker lower bound than we have proven in Corollary 1. Therefore, we will prove the claim for larger values of $q$. Let us define several parameters for which we will prove the claim. Let $\tau = \frac{1}{400^2}$, let $A = 4c + 4\ln\left(\tau \cdot \frac{k}{q^2}\right)$. Observe that since $q \leq \sqrt{k}$ and $k \geq 1$, and

$c > 0$, it holds that $A \geq 4 \ln(400^2) \geq 8$. We will show that for any $q \geq 2c + \log(\tau k)$ it holds that $\ell_1\text{-}dist(f) \geq 1 + \frac{1}{\tau} \cdot \frac{q}{k} \cdot \frac{1}{A}$.

Denote $S \subseteq \binom{n}{2}$ the set of pairs such that $dist_f(i,j) \leq 1 + \frac{A}{q}$, for all $(i,j) \in S$. Denote $L = \binom{n}{2} \setminus S$. Recall the definition of $\epsilon_{ij} = |expans_f(i,j) - 1|$. Note that for any pair $(i,j) \in S$ it holds that $\epsilon_{ij} \leq \frac{A}{q}$.

The proof of the claim follows from the following inequality (which we will prove in the sequel): $(Energy_2(f|S))^2 \geq \frac{1}{\tau} \cdot \frac{1}{k}$, where as before, $f|S$ denotes the embedding $f$ restricted to the subset $S$.

Assume the inequality is correct, then we have

$$\frac{1}{\tau} \cdot \frac{1}{k} \leq (Energy_2(f|S))^2 \leq \max_{(i,j) \in S} \{\epsilon_{ij}\} \cdot Energy_1(f|S) \leq \frac{A}{q} \cdot Energy_1(f),$$

implying $Energy_1(f) \geq \frac{1}{\tau} \cdot \frac{q}{k} \cdot \frac{1}{A}$. Therefore, by Claim 19 we obtain

$$\ell_1\text{-}dist(f) \geq 1 + REM_1(f) \geq 1 + Energy_1(f) \geq 1 + \frac{1}{\tau} \cdot \frac{q}{k} \cdot \frac{1}{A},$$

as required. Next we prove the inequality itself.

Recall that $n \geq 18k$. Therefore, from Claim 64 it follows that $(Energy_2(f))^2 \geq \frac{2}{400^2} \cdot \frac{1}{k} = \frac{2}{\tau} \cdot \frac{1}{k}$. The proof proceeds by showing that $(Energy_2(f|L))^2$ has small contribution to the total energy. Specifically, in the next computations we show that for the chosen parameters, $(Energy_2(f|L)^2 \leq \frac{1}{\tau} \cdot \frac{1}{k}$, which will finish the proof of the claim.

For all $\alpha \geq 1$, denote by $P_\alpha$ the fraction of the pairs in $L$ that have distortion bigger than $1 + \alpha \cdot \frac{A}{q}$. By assumption, $(\ell_q\text{-}dist(f))^q \leq \left(1 + \frac{c}{q}\right)^q$. Then by Markov's inequality, we have

$$P_\alpha \leq \frac{\left(1 + \frac{c}{q}\right)^q}{\left(1 + \alpha \cdot \frac{A}{q}\right)^q}.$$

Therefore,

$$(Energy_2(f|L))^2 \leq 2\left(\frac{A}{q}\right)^2 \cdot \int_1^\infty \alpha P_\alpha \, d\alpha \leq 2\left(\frac{A}{q}\right)^2 \int_1^\infty \frac{\left(1 + \frac{c}{q}\right)^q}{\left(1 + \alpha \cdot \frac{A}{q}\right)^q} \cdot \alpha \, d\alpha \leq 2e^c \cdot \left(\frac{A}{q}\right)^2 \cdot \int_1^\infty \frac{\alpha}{\left(1 + \alpha \cdot \frac{A}{q}\right)^q} \, d\alpha.$$

We compute the integral by dividing the integration interval into two sub intervals, and we show that for the chosen parameters, each is bounded by $\frac{1}{2\tau k}$. We start with the first integral:

$$\int_1^{\frac{q}{A}} \frac{\alpha}{\left(1 + \alpha \cdot \frac{A}{q}\right)^q} \, d\alpha \leq \int_1^{\frac{q}{A}} \frac{\alpha}{e^{\frac{\alpha A}{2}}} \, d\alpha,$$

where the last inequality holds since for all $0 \leq x \leq 1$, $1 + x \geq e^{x/2}$. Substituting $z = \frac{\alpha A}{2}$ we obtain

$$= \left(\frac{2}{A}\right)^2 \cdot \int_{\frac{A}{2}}^\infty \frac{z}{e^z} = \left(\frac{2}{A}\right)^2 \cdot \Gamma\left(2, \frac{A}{2}\right) = \left(\frac{2}{A}\right)^2 \cdot \left((A/2 + 1) \cdot e^{-\frac{A}{2}}\right),$$

using the exact evaluation of $\Gamma(2, A/2)$ Therefore, putting all together, for the first integral we obtain

$$2e^c \cdot \left(\frac{A}{q}\right)^2 \cdot \int_1^{\frac{q}{A}} \frac{\alpha}{\left(1 + \alpha \cdot \frac{A}{q}\right)^q} \, d\alpha \leq \frac{4e^c}{q^2} \cdot (A + 2) \cdot e^{-\frac{A}{2}}.$$

Substituting the value of $A$ we obtain

$$\frac{4e^c}{q^2} \cdot \frac{A+2}{e^{2c} \cdot \tau^2 \cdot \left(\frac{k}{q^2}\right)^2} = \frac{4}{e^c} \cdot \frac{q^2}{\tau k} \cdot (A+2) \cdot \frac{1}{\tau k}.$$

We should verify that for the chosen value of $A$

$$\frac{4}{e^c} \cdot \frac{q^2}{\tau k} \cdot (A+2) \le 1,$$

Substituting the value of $A$, and using $k \ge q^2$, and out choice of $\tau$, we have:

$$A + 2 \le 4 \ln\left(\frac{e^c \tau k}{q^2}\right) + 2 \le \frac{e^c \tau k}{4q^2},$$

as required. Now, it remains to bound the second integral taken over the second interval:

$$2e^c \cdot \left(\frac{A}{q}\right)^2 \cdot \int_{\frac{q}{A}}^{\infty} \frac{\alpha}{\left(1 + \alpha \cdot \frac{A}{q}\right)^q} \, d\alpha = 2e^c \int_{2}^{\infty} \frac{z-1}{z^q} \, dz \le 2e^c \cdot \frac{1}{2^{(q-2)} \cdot (q-2)},$$

where the second step is obtain by substituting $z = 1 + \frac{\alpha A}{q}$, and noting that we may assume $q > 2$. Recalling that $A \ge 8$, and that $q \ge 2c + \log(\tau k)$, we conclude

$$2e^c \cdot \frac{1}{2^{(q-2)} \cdot (q-2)} \le e^c \cdot \frac{1}{2} \cdot \frac{1}{4^c k \tau} \le \frac{1}{2\tau k},$$

as required. This completes the proof of the claim. $\square$

**Claim 76.** *Let $k \ge 1$ be any integer, and let $n \ge 18k$. Let $f : E_n \to \ell_2^k$ be an embedding such that for some $\sqrt{k} \ge q > 2$, it holds that $Energy_q(f) \le c \cdot \sqrt{\frac{q}{k}}$, or some $c \ge 1$. Then, $Energy_1(f) \ge \Omega\left(\frac{1}{\left(\frac{q}{q-2}c\right)^{(q/(q-2))}} \cdot \frac{1}{\sqrt{q}\sqrt{k}}\right).$*

*Proof.* The proof closely follows the lines of the proof of Claim 75. Let $\tau = 400^2$, and $A = \left(2\frac{q}{q-2}\tau c^q\right)^{\frac{1}{q-2}}$. We will show that $Energy_1(f) \ge \frac{1}{\tau} \cdot \frac{1}{A} \cdot \frac{1}{\sqrt{k}\sqrt{q}}$. Denote by $S \subseteq \binom{n}{2}$ the set of the pairs with $\epsilon_{ij} \le A \cdot \sqrt{\frac{q}{k}}$. The proof of the claim will follow from the following inequality: $(Energy_2(f|S))^2 \ge \frac{1}{\tau} \cdot \frac{1}{k}$.

Assuming the inequality is correct, we obtain

$$\frac{1}{\tau} \cdot \frac{1}{k} \le (Energy_2(f|S))^2 \le \max_{(i,j) \in S}\{\epsilon_{ij}\} \cdot Energy_1(f|S) \le A \cdot \sqrt{\frac{q}{k}} \cdot Energy_1(f),$$

which implies the claim. It remains to prove the inequality itself.

For any $\alpha \ge 1$, denote $P_\alpha$ the fraction of the pairs of $L$ with $\epsilon_{ij} \ge \alpha \cdot A \cdot \sqrt{\frac{q}{k}}$. Therefore, by Markov's inequality

$$P_\alpha \le \frac{c^q \left(\frac{q}{k}\right)^{q/2}}{\alpha^q \cdot A^q \left(\frac{q}{k}\right)^{q/2}} = \frac{c^q}{A^q} \frac{1}{\alpha^q}.$$

Therefore, the contribution of the pairs of the set $L$ to the total sum is bounded by

$$(Energy_2(f|L))^2 \le 2\left(A \cdot \sqrt{\frac{q}{k}}\right)^2 \cdot \int_1^\infty \alpha P_\alpha \, d\alpha \le 2\frac{c^q}{A^q} \cdot A^2 \cdot \frac{q}{k} \cdot \int_1^\infty \frac{1}{\alpha^{q-1}} \, d\alpha = 2\frac{c^q}{A^{q-2}} \cdot \frac{q}{q-2} \cdot \frac{1}{k}$$

By our choice of $A$ we have that $\frac{c^q}{A^{q-2}} \cdot \frac{q}{q-2} \le \frac{1}{2\tau}$, completing the proof of the claim. $\square$

## I.1 Limit of simultaneous embeddings for large $q$

In the following theorem we show that any embedding that achieves $\ell_1\text{-}dist = O(1)$, cannot obtain for large $q \geq k - 1$, the best bounds that could be achieved by non-simultaneous embeddings (essentially it must forfeit the square root gain). We achieve the following results as corollaries of Claim 24(second argument in max) composed with Claim 72 (for non-expansive embeddings), and Claim 74.

**Theorem 77.** *Let $E_n$ be an equilateral metric space, for $n$ large enough. Let $f : E_n \to \ell_2^k$ be an embedding such that $\ell_1\text{-}dist(f) = O(1)$. Then the following simultaneous guarantees on $\ell_q\text{-}dist(f)$ hold*[14]:

| $k - 1 \leq q \leq k^-$ | $q = k$ | $k < q \leq \log n$ |
|---|---|---|
| $\Omega\left(\frac{1}{\sqrt{k}(k-q)^{1/q}}\right)$ | $\Omega\left(\frac{(\log n)^{1/k}}{\sqrt{k}}\right)$ | $\Omega\left(n^{\frac{1}{k}-\frac{1}{q}}\right)$ |

# J  Approximate optimal embedding of general metrics

We present here the tight relation of the values of $Stress_q$ and $Stress_q^*$. We formulate this relation in the following claim.

**Claim 78.** *Let $(X, d_X)$ be any finite metric space, and $(Y, d_Y)$ be any scalable metric space. For a given $q \geq 1$ and a distribution $\Pi$ over pairs of $X$, the following two statements hold.*

1. *For any $f : X \to Y$ there exists an embedding $F : X \to Y$ such that $Stress^{*(\Pi)}_q(F) \leq 2 \cdot 2^{1/q} Stress^{(\Pi)}_q(f)$.*

2. *For any $g : X \to Y$ there exists an embedding $G : X \to Y$ such that $Stress^{(\Pi)}_q(G) \leq 2 \cdot 2^{1/q} Stress^{*(\Pi)}_q(g)$.*

*Proof.* We show the proof for the first item of the claim. The second item is obtained similarly. Denote by $H^* : X \to Y$ the trivial embedding for $Stress^{*(\Pi)}_q$. Recall that the trivial embedding is just some embedding into $Y$ scaled such that its $Stress^{*(\Pi)}_q$ is at most 1. We have two cases.

If $Stress^{(\Pi)}_q(f) \geq \frac{1}{2 \cdot 2^{1/q}}$, then taking $F := H^*$ we obtain $Stress^{*(\Pi)}_q(H) \leq 1 \leq 2 \cdot 2^{1/q} \cdot Stress^{(\Pi)}_q(f)$. Otherwise, if $Stress^{(\Pi)}_q(f) \leq \frac{1}{2 \cdot 2^{1/q}}$, taking $F := f$ we obtain that we have to show that

$$\left(Stress^{(\Pi)}_q(f)\right)^q \geq \frac{1}{2^q \cdot 2} \cdot \left(Stress^{*(\Pi)}_q(f)\right)^q.$$

Recall that it holds that

$$\left(Stress^{(\Pi)}_q(f)\right)^q = \left(Stress^{*(\Pi)}_q(f)\right)^q \cdot \frac{\sum\limits_{1 \leq i \neq j \leq n} \Pi(i,j)\left(\hat{d}_{ij}\right)^q}{\sum\limits_{1 \leq i \neq j \leq n} \Pi(i,j)(d_{ij})^q},$$

where $d_{ij} = d_X(i,j)$, and $\hat{d}_{ij} = d_Y(f(i), f(f))$, for all $i \neq j \in X$. Therefore, it is enough to show that

$$\frac{\sum\limits_{1 \leq i \neq j \leq n} \Pi(i,j)\left(\hat{d}_{ij}\right)^q}{\sum\limits_{1 \leq i \neq j \leq n} \Pi(i,j)(d_{ij})^q} \geq \frac{1}{2^q \cdot 2}.$$

Therefore, it holds that

$$2^q \left( \frac{\sum\limits_{1 \le i \ne j \le n} \Pi(i,j) \left( \hat{d}_{ij} \right)^q}{\sum\limits_{1 \le i \ne j \le n} \Pi(i,j)(d_{ij})^q} + \frac{\sum\limits_{1 \le i \ne j \le n} \Pi(i,j) \left| d_{ij} - \hat{d}_{ij} \right|^q}{\sum\limits_{1 \le i \ne j \le n} \Pi(i,j)(d_{ij})^q} \right) =$$

$$2^q \left( \frac{\sum\limits_{\hat{d}_{ij} \ge d_{ij}} \Pi(i,j) \left( \hat{d}_{ij} \right)^q}{\sum\limits_{1 \le i \ne j \le n} \Pi(i,j)(d_{ij})^q} + \frac{\sum\limits_{\hat{d}_{ij} < d_{ij}} \Pi(i,j) \left( \hat{d}_{ij} \right)^q}{\sum\limits_{1 \le i \ne j \le n} \Pi(i,j)(d_{ij})^q} + \frac{\sum\limits_{1 \le i \ne j \le n} \Pi(i,j) \left| d_{ij} - \hat{d}_{ij} \right|^q}{\sum\limits_{1 \le i \ne j \le n} \Pi(i,j)(d_{ij})^q} \right) \ge$$

$$\frac{\sum\limits_{\hat{d}_{ij} \ge d_{ij}} \Pi(i,j) \left( d_{ij} \right)^q}{\sum\limits_{1 \le i \ne j \le n} \Pi(i,j)(d_{ij})^q} + \frac{\sum\limits_{\hat{d}_{ij} < d_{ij}} \Pi(i,j) \left( d_{ij} \right)^q}{\sum\limits_{1 \le i \ne j \le n} \Pi(i,j)(d_{ij})^q} = 1.$$

From this inequality we obtain the required. □

The following corollary immediately follows from the claim.

**Corollary 8.** *Let* $(X, d_X)$ *be an* $n$-*point metric space and* $\Pi$ *be any distribution. Then for any* $q \ge 2$ *there exists a polynomial time algorithm that computes an embedding* $F : X \to \ell_2^n$ *such that* $Stress^*{}_q^{(\Pi)}(F) \le 4 \cdot 4^{1/q} OPT^*$, *where* $OPT^* = \inf\limits_{f : X \to \ell_2^n} \{Stress^*{}_q^{(\Pi)}(f)\}$.

*Proof.* In Claim 78, take $f : X \to \ell_2^n$ to be an embedding that brings $Stress_q^{(\Pi)}$ to its minimum, and take $g : X \to Y$ to be an embedding that brings $Stress^*{}_q^{(\Pi)}$ to its minimum. By Theorem 8 $f$ can be computed in polynomial time. Therefore, $F$ would be either equal to $H^*$ or to $f$ itself, and this can be decided in polynomial time. Applying Claim 78 to this choice results the corollary.

□

**Remark 79.** *We note that for the special case of* $q = 2$ *(and for any distribution* $\Pi$ *over the pairs), it can be easily shown that* $OPT = OPT^*$, *and furthermore, scaling the embedding that brings* $Stress_2^{(\Pi)}$ *to the minimum by factor* $\frac{\sum \Pi(i,j)(d_{ij})^2}{\sum \Pi(i,j)(\hat{d}_{ij})^2}$, *results in the embedding that brings* $Stress^*{}_2^{(\Pi)}$ *to the minimum. Note that this observation implies approximation factor* 1 *for the above corollary.*

**Theorem 80.** *Let* $(X, d_X)$ *be an* $n$-*point metric space and* $\Pi$ *be any distribution. Then for any* $q \ge 2$ *and for* $Obj_q^{(\Pi)} \ne Stress^*{}_q^{(\Pi)}, \Phi_{\sigma,q,2}^{(\Pi)}$ *there exists a polynomial time algorithm that computes an embedding* $f : X \to \ell_2^n$ *such that* $Obj_q^{(\Pi)}(f)$ *approximates* $OPT^{(n)}$ *to within any level of precision. For* $Obj_q^{(\Pi)} = Stress^*{}_q^{(\Pi)}$ *there exists a polynomial time algorithm that computes an embedding* $f : X \to \ell_2^n$ *with* $Stress^*{}_q^{(\Pi)}(f) = O\left(OPT^{(n)}\right)$.

*Proof.* For $X = \{x_i\}_{i=0}^{n-1}$ denote $d_{ij} = d_X(x_i, x_j)$. For all $0 \le i < j \le n-1$ let variable $z_{ij}$ represent the square of the distance between the images of the pair $(x_i, x_j)$. Let $g_{ij} = \frac{1}{2}(z_{0i} + z_{0j} - z_{ij})$, then $\sqrt{z_{ij}}$ describe Euclidean distances iff the matrix $G[i, j] = (g_{ij})$ is PSD. For all the objective measures, the optimization program is:

$$\min \sum_{0 \le i < j < n} \Pi(i,j) \left( dist_{Obj}(i,j) \right)^q; \quad \text{s.t.} \ \ g_{ij} = (1/2)(z_{0i} + z_{0j} - z_{ij}), \ \ G[i,j] = (g_{ij}) \succeq 0, \ \ z_{ij} \ge 0; \ \ \forall i \ne j.$$

Where for $Obj = \ell_q\text{-}dist$, $(dist_{Obj}(i,j))^q = (\max\{\frac{z_{ij}}{(d_{ij})^2}, \frac{(d_{ij})^2}{z_{ij}}\})^{q/2}$; for $Obj = Energy$, $(dist_{Obj}(i,j))^q = ((\sqrt{z_{ij}}/d_{ij} - 1)^2)^{q/2}$; for $Obj = Stress$, $(dist_{Obj}(i,j))^q = ((\sqrt{z_{ij}} - d_{ij})^2)^{q/2}$; for $Obj = REM$, $(dist_{Obj}(i,j))^q = \max\left\{ ((\max\{\frac{d_{ij}}{\sqrt{z_{ij}}} - 1, 0\})^2)^{q/2}, ((\frac{\sqrt{z_{ij}}}{d} - 1)^2)^{q/2} \right\}$. It is easily checked that all are convex functions, for $q \ge 2$. □

In the next claim we show that optimizing for $\sigma$-distortion can be reformulated in terms of optimizing for Energy measure.

**Claim 81.** *Let $(X, d_X)$ be an $n$-point metric, and $\Pi$ be a distribution over $\binom{X}{2}$. For a given $q \geq 2$, there is a polynomial time algorithm that finds an embedding $f : X \to \ell_2^n$, such that $\Phi_{\sigma,q,2}^{(\Pi)}$ approximates $OPT^n$, with any level of precision.*

*Proof.* Similarly to the proof of Theorem 80, the embedding $f$ is given by the solution to the following optimization problem:

$$minimize \quad \sum_{1 \leq i \neq j \leq n} \Pi(i,j) \left| \frac{\sqrt{z_{ij}}/d_{ij}}{\sqrt{\binom{n}{2}^{-1} \sum_{1 \leq i \neq j \leq n} z_{ij}/d_{ij}^2}} - 1 \right|^q ,$$

subject to $z_{ij} \geq 0$, $g_{ij} = (1/2)(z_{0i} + z_{0j} - z_{ij})$, and $G[i,j] = (g_{ij}) \succeq 0$. Since $\sigma$-distortion is scalable, this minimization problem is equivalent to the following:

$$minimize \quad \sum_{1 \leq i \neq j \leq n} \Pi(i,j) \left| \frac{\sqrt{z_{ij}}/d_{ij}}{\sqrt{\binom{n}{2}^{-1} \sum_{1 \leq i \neq j \leq n} z_{ij}/d_{ij}^2}} - 1 \right|^q ,$$

subject to $\binom{n}{2}^{-1} \cdot \sum_{1 \leq i \neq j \leq n} z_{ij}/d_{ij}^2 = 1$ and subject to all the rest previous constraints. Namely, the optimization problem we have to solve is $minimize \quad \sum_{1 \leq i \neq j \leq n} \Pi(i,j) \left| \sqrt{z_{ij}}/d_{ij} - 1 \right|^q$, s.t. $z_{ij} \geq 0$, $g_{ij} = (1/2)(z_{0i} + z_{0j} - z_{ij})$, $G[i,j] = (g_{ij}) \succeq 0$, and $\binom{n}{2}^{-1} \cdot \sum_{1 \leq i \neq j \leq n} z_{ij}/d_{ij}^2 = 1$. Note that the objective function is the objective of $Energy_q$ measure, which we showed to be convex. In addition, all the constraints are convex. Thus, the minimization problem is a convex minimization, which can be solved in polynomial time with any level of precision. This completes the proof. □

**Theorem 82.** *Let $(X, d_X)$ be a finite metric space, $\Pi$ a distribution over $\binom{X}{2}$, $k \geq 3$ and $2 \leq q \leq k-1$. There exists a randomized polynomial time algorithm that finds an embedding $F : X \to \ell_2^k$, such that with high probability each one of the following holds:[15]*

1. $\ell_q\text{-}dist^{(\Pi)}(F) = \left(1 + O\left(\frac{1}{\sqrt{k}} + \frac{q}{k-q}\right)\right) OPT.$

2. $Obj_q^{(\Pi)}(F) = O(OPT) + O\left(\sqrt{\frac{q}{k}}\right), \quad for \quad Obj_q^{(\Pi)} \in \left\{REM_q^{(\Pi)}, Energy_q^{(\Pi)}, \Phi_{\sigma,q,2}^{(\Pi)}, Stress_q^{(\Pi)}, Stress^{*(\Pi)}_q\right\}.$

*Proof.* The embedding $F : X \to \ell_2^k$ is defined to be the composition of the embedding $OPT^{(n)} : X \to \ell_2^n$, and the JL transform $g : \ell_2^n \to \ell_2^k$. In Appendix D.2 we provide results on the composition of embeddings for each of the objective functions $Obj_q^{(\Pi)}$, except $\Phi_{\sigma,q,2}^{(\Pi)}$, which we treat in a sequel. In particular, we bound $E\left[Obj_q^{(\Pi)}(F)\right]$ in terms of $OPT^{(n)}$ and $\left(E\left[(Obj_q^{(\Pi)}(g))^q\right]\right)^{1/q}$. For the first item we use Claim 29 to obtain $E\left[\ell_q\text{-}dist^{(\Pi)}(F)\right] = \left(1 + O\left(\frac{1}{\sqrt{k}} + \frac{q}{k-q}\right)\right) OPT$. To get the constant probability of success we apply Markov's inequity on the random variable $\ell_q\text{-}dist^{(\Pi)}(F) - OPT$. The second item follows by composition claims: Claims 30, 31, 32, 33 and by bounds that are developed in the proof of Theorem 2. To get the constant probability of success we apply Markov's inequality. A slightly more involved argument yields the high probability result.

For the $\sigma$-distortion measure, let $f : X \to \ell_2^n$ be the embedding of Claim 81. Namely, $\Phi_{\sigma,q,2}^{(\Pi)}(f)$ has optimal value for $\Phi_{\sigma,q,2}^{(\Pi)}$ objective, and also $Energy^{(\Pi)}(f)$ has optimal value for $Energy_q^{(\Pi)}$ objective, under constraint $\ell_2\text{-}expans^{(\Pi)}(f) = 1$.

Let $g : \ell_2^n \to \ell_2^k$ be a JL transform into $k$ dimensions, and let $F : X \to \ell_2^k$ be the composition $g \circ f$. Then, by the composition property of $Energy_q$ (Claim 31) and by the analysis of the JL transform

for the additive measures:

$$E\left[Energy_q^{(\Pi)}(F)\right] = O\left(Energy_q^{(\Pi)}(f)\right) + O\left(\sqrt{\frac{q}{k}}\right) = O\left(OPT\right) + O\left(\sqrt{\frac{q}{k}}\right),$$

where $OPT$ is the value of the optimal (for $\Phi_{\sigma,q,2}^{(\Pi)}$) embedding into $k$ dimensions. In addition, by the composition properties of the $\ell_q$-expansion (Claim 29) and by bounds of Claim 44:

$$E\left[(\ell_2\text{-}expans(F))^2\right] \le (\ell_2\text{-}expans(f))^2 \cdot \left(1 + \frac{2}{k}\right).$$

In addition,

$$E\left[\frac{1}{(\ell_2\text{-}expans(g \circ f))^2}\right] = E\left[\frac{1}{\sum_{1 \le i \ne j \le n} \binom{n}{2}^{-1} \cdot (expans_f(i,j))^2 \cdot (expans_g(i,j))^2}\right]$$

$$\le E\left[\sum_{1 \le i \ne j \le n} \binom{n}{2}^{-1} \cdot (expans_f(i,j))^2 \cdot \frac{1}{(expans_g(i,j))^2}\right],$$

where the last inequity holds by Jensen's inequality, since $\sum_{1 \le i \ne j \le n} \binom{n}{2}^{-1} \cdot (expans_f(i,j))^2 = 1$ (recall that $f$ is such that $\ell_2\text{-}expans(f) = 1$). Then, since $1/(expans_g(i,j)) = contract_g(i,j)$:

$$E\left[\frac{1}{(\ell_2\text{-}expans(g \circ f))^2}\right] \le E\left[\sum_{1 \le i \ne j \le n} \binom{n}{2}^{-1} \cdot (expans_f(i,j))^2 \cdot (contract_g(i,j))^2\right]$$

$$= \sum_{1 \le i \ne j \le n} \binom{n}{2}^{-1} \cdot (expans_f(i,j))^2 \cdot E\left[(contract_g(i,j))^2\right] \le 1 + \frac{2}{k-1},$$

where the last inequality is due to Claim 43.

Thus, by Markov's inequality and union bound we conclude that with constant probability there exists an embedding $F : X \to \ell_2^k$, such that $\frac{1}{1+16/k} \le \ell_2\text{-}expans(F) \le 1 + 16/k$, and $Eneregy_q^{(\Pi)}(F) = O\left(OPT\right) + O\left(\sqrt{\frac{q}{k}}\right)$. Therefore, by Claim 22, we conclude that with constant probability there is an embedding $F : X \to \ell_2^k$ such that $\Phi_{\sigma,q,2}(F) = O\left(OPT\right) + O\left(\sqrt{\frac{q}{k}}\right)$, which completes the proof. $\square$

## K  On moments of embedding general metrics

In this section we consider a general question of embedding a finite metric space into low dimensional normed spaces.

In (3) it was shown that every finite metric embeds in Euclidean space with $\ell_q$-distortion $O(q)$ in $O(\log n)$ dimension. One immediate consequence of our bounds on the $\ell_q$-distortion is a substantial improvement of the dimension in this theorem to $O(q)$. An immediate application are *linear size-constant query time* distance oracles with average case guarantees.

**Theorem 83** (Theorem 9, (3)). *Let $(X, d_X)$ be an $n$ point metric space, and let $1 \le p \le \infty$. There exists a non-contractive embedding $f : X \to \ell_p^d$, with $d = O_p(\log n)$, and such that for every $1 \le q \le \infty$ it holds that $\ell_q\text{-}dist(f) = O\left(\min\{q, \log n\}/p\right)$.*

Assuming $q = O(\log n)$, we obtain the following corollary result:

**Corollary 9** (General metric space into $k$-dimensional Euclidean space). *Given any $n$ point metric space $(X, d_X)$, integer $k \ge 1$, any $q \ge 1$, there exists an embedding $F : X \to \ell_2^k$ with the following guarantees[16] on $\ell_q\text{-}dist(F)$:*

| $1 \leq q < \tilde{c}k$ | $\tilde{c}k \leq q \leq k^-$ | $q = k$ | $k < q$ |
|---|---|---|---|
| $O\left(\sqrt{q}\right)$ | $O\left(\sqrt{q}\left(\frac{k}{k-q}\right)^{1/2q}\right)$ | $O\left(\sqrt{k}(\sqrt{\log n})^{1/k}\right)$ | $O\left(q^{3/4}\frac{n^{(1/k-1/q)}}{k^{1/4}(q-k)^{1/2q}}\right)$ |

*Proof.* Denote the embedding into Euclidean space of Theorem 83 by $f_{ABN}$, and the JL embedding into $k$ dimensions of Theorem 45 by $f_{JL}$. Embed $X$ as follows: first apply $f_{ABN}$ to Euclidean space, and then apply $f_{JL}$ to reduce the dimension. To analyze the $\ell_q\text{-}dist(f_{JL} \circ f_{ABN})$ we invoke the second part of Claim 27, with the following parameters: $f = f_{ABN}$, $g = f_{JL}$, $s_1 = t_1 = 2$, and $s_2 = \infty$, $t_2 = 1$. Since $f_{ABN}$ is non-contractive it holds that $\ell_\infty\text{-}contr(f_{ABN}) \leq 1$, and for any $q \geq 1$ it holds that $\ell_{2q}\text{-}expans(f_{ABN}) = \ell_{2q}\text{-}dist(f_{ABN}) = O(q)$. The bounds on $\ell_q$-expansion and $\ell_q$-contraction of the JL transform (for a given value of $q$) are estimated in Lemma 43 ($\ell_q$-contraction for $1 \leq q < k$), in Lemma 53 ($\ell_q$-contraction for $q \geq k$), and in Lemma 44 ($\ell_q$-expansion for all $k, q \geq 1$). $\qquad\square$

Yet stronger result follows by composition of $f_{ABN}$ with the embedding of Theorem G.2 (note that both embeddings provide guarantees for all $q$ simultaneously), and by the first part of Claim 27.

**Corollary 10.** *Given any $n$-point metric $(X, d)$ and any integer $k \geq 1$ there exists an embedding $F : X \to \ell_2^k$ such that for all $q \geq 1$ simultaneously the following bounds on $\ell_q\text{-}dist(f)$ hold:*

| $1 \leq q < \tilde{c}k$ | $\tilde{c}k \leq q \leq k^-$ | $q = k$ | $k < q$ |
|---|---|---|---|
| $O(q)$ | $O\left(q + \left(\frac{k}{k-q}\right)^{1/q}\right)$ | $O\left(k + (\log n)^{1/k}\right)$ | $O\left(q\sqrt{q}k + \frac{n^{(2/k-2/q)}}{(q-k)^{1/q}}\right)$ |

# L   Application to hyper-sketching

We propose a generalization of the standard sketching model (45; 43) for pairs, and consider sketches that provide a good approximation for the weighted sum of moments of distances of a set of points $S \subseteq X$.

**Definition 10 (Hyper-Sketching).** *Let $(X, d)$ be a finite metric space, and let $\ell \in \mathbb{N}$, $\alpha \geq 1$ and $0 < \delta < 1$. We say that $X$ admits an $(\ell, \alpha, \delta)$-hyper-sketching if there exists a distribution over maps $\sigma$, mapping points of $X$ into words of length $\ell$, and an algorithm $\mathcal{H}$ such that given a query $(S, w, q)$, where $S \subseteq X$, $w$ is a nonnegative weight function over pairs of $S$ and $q \geq 1$, computes a function $\mathcal{H}(\sigma(S))$, that with probability at least $1 - \delta$ returns an $\alpha$ approximation to $\left(\sum_{(x,y)\in\binom{S}{2}} w(x,y)(d(x,y))^q\right)^{1/q}$. The query time is the running time of $\mathcal{H}$.*

We may focus on $\delta = 2/3$ and omit it from the notation. The results extend by the standard median estimate method at a cost of $O(\log(1/\delta))$ factor in all parameters. The query time[17] depends on $m_w = |support(w)|$, which is bounded by $\binom{|S|}{2}$. In what follows we focus on Euclidean subsets, whereas the case of the general metric sketching analyzed in the next subsection.

**Theorem 84.** *Let $X \subset \ell_2^d$ be any $n$-point set, and let $0 < \epsilon < 1$. Then $X$ admits $\left(1/\epsilon^2, \alpha(\epsilon, q)\right)$-hyper-sketching, where $\alpha(\epsilon, q) = 1 + O\left(\epsilon + q\epsilon^2\right)$ if $q < 1/\epsilon^2$, and $\alpha(\epsilon, q) = O\left(\epsilon\sqrt{q}\right)$ if $q \geq 1/\epsilon^2$. The query time is $O\left(m_w/\epsilon^2\right) = O\left((s/\epsilon)^2\right)$, and the preprocessing time is $O\left(\frac{nd}{\epsilon^2}\right)$.*

*Proof.* In the preprocessing stage we apply the JL map $f : X \to \ell_2^{(1/\epsilon^2)}$. Given any $S \subseteq X$ and nonnegative weight function $w$ over pairs of $S$, let $W = \sum_{(u,v)\in\binom{S}{2}} w(u,v)$. We can extend $w$ into a distribution $\Pi$ over all pairs of $X$ by defining $\Pi(x,y) = w(x,y)/W$ for all $x \neq y \in S$, and $\Pi(x,y) = 0$, for all the rest pairs. It therefore holds that $\left(\sum_{(x,y)\in\binom{S}{2}} w(x,y)\|x-y\|_2^q\right)^{\frac{1}{q}} = W \cdot \left(\sum_{(x,y)\in\binom{S}{2}} \Pi(x,y)\|x-y\|_2^q\right)^{\frac{1}{q}} = W \cdot \left(\sum_{(x,y)\in\binom{X}{2}} \Pi(x,y)\|x-y\|_2^q\right)^{\frac{1}{q}}$. To answer the query,

the algorithm computes the quantity: $W \cdot \left( \sum_{(x,y) \in \binom{S}{2}} \Pi(x,y)(\|f(x) - f(y)\|_2^q)\right)^{\frac{1}{q}}$. The theorem follows immediately from our bounds on the distortion of the $l_q$-norm in Theorem 57. $\square$

**Improved Query Time for Sum of Distances**. Focusing on the basic case of $q = 1$ and unit weight function $w$, we improve the query time to linear in $|S|$ while keeping asymptotically the same bound on the distortion. We use ideas of (26) in a non-black manner:

**Theorem 85.** *Let $X \subset \ell_2^d$ be any $n$-point set, and let $0 < \epsilon < 1$. Then $X$ admits $(O\left(1/\epsilon^2\right), 1+\epsilon)$-hyper-sketching that answers queries for $q = 1$ and uniform weight function $w$, with query time $O(s + 1/\epsilon^4)$, and with the preprocessing time $O\left(\frac{nd}{\epsilon^2}\right)$.*

*Proof.* We shall present here a simple construction that provides query time $O\left(s + \frac{\sqrt{s}}{\epsilon^4}\right) = O(s + 1/\epsilon^8)$. A more involved argument allows us to improve the query time (details will appear in the full version of the paper).

In the preprocessing phase we apply two JL embeddings of dimension $k$, $f : X \to \ell_2^k$ where $k = O\left(1/\epsilon^2\right)$, and $g : \ell_2^k \to \ell_2^8$. For $x_i \neq x_j \in X$ denote by $f_{ij} = \|f(x_i) - f(x_j)\|_2$, and $g_{ij} = \|g(x_i) - g(x_j)\|_2$. Given a query set $S \subseteq X$ of size $|S| = s$, let $D = \sum_{(x_i,x_j) \in \binom{S}{2}} \|x_i - x_j\|_2$. We argue that with probability at least $3/4$ all of the following events hold

1. $\sum_{(x_i,x_j) \in \binom{S}{2}} f_{ij} \in (1 \pm \epsilon)D$ ,

2. $\sum_{(x_i,x_j) \in \binom{S}{2}} g_{ij} = \Theta\left(D\right)$ ,

3. There exists a positive constant $c > 0$ such that for all $(x_i, x_j) \in \binom{S}{2}$ it holds that $\frac{f_{ij}}{g_{ij}} \leq c \cdot s^{1/4}$.

It is enough to show that each of these events holds with probability at least $11/12$. The first two items hold by Theorem 57. The third item holds by Lemma 52.

From now on we condition on the event that all these three events hold. To answer the query, apply the first phase of (26) on $g(f(S))$ with the distances given by the $g_{ij}$, to obtain the probabilities $\tilde{p}_{ij}$. By the above observations, there exists a positive constant $\tilde{c}$ such that $\tilde{p}_{ij} \geq \frac{g_{ij}}{\tilde{c}D}$. This involves $O(s + 1/\epsilon^2)$ distance computations, where computation takes a constant time (it is done in $\mathbb{R}^8$). Next, sample the set $U$ of samples of pairs of $f(S)$, $|U| = u$, according to the probabilities $\tilde{p}_{ij}$ (we will choose $u$ later), in time $O(s + u)$, and compute the estimator $\hat{D}$ from the samples. It holds that

$$E\left[\hat{D}\right] = \frac{1}{u} \cdot E\left[\sum_{(x_i,x_j) \in U} f_{ij}/\tilde{p}_{ij}\right] = \sum_{(x_i,x_j) \in \binom{S}{2}} f_{ij}.$$

Denote $D_f = \sum_{(x_i,x_j) \in \binom{S}{2}} f_{ij}$, and note that for all $i \neq j$ it holds that $f_{ij}/\tilde{p}_{ij} \leq \frac{f_{ij}}{g_{ij}} \cdot \tilde{c}D \leq c's^{1/4}D$, for some positive $c'$. Therefore, by Hoeffding's inequality it holds that

$$\Pr\left[|\hat{D} - D_f| \geq \epsilon D_f\right] \leq 2e^{-\frac{2u\epsilon^2(D_f)^2}{(c')^2\sqrt{s}D^2}}.$$

Since $D_f/D \geq C$, for some $C > 0$, it is enough to choose $u = \hat{C}\frac{\sqrt{u}}{\epsilon^2}$, for constant $\hat{C} > 0$ big enough, to have the desired approximation with constant probability. Therefore, the time required to compute $\hat{D}$ on the set of samples $U$ is $O\left(\frac{\sqrt{s}}{\epsilon^4}\right) = O\left(s + \frac{1}{\epsilon^8}\right)$ (using the inequality $a \cdot b \leq a^2 + b^2$ for any $a, b \in \mathbb{R}$), which finishes the proof. $\square$

## L.1 Hyper Sketching for General Metrics

Combining our JL moment analysis results with some previous results on embeddings of arbitrary metric spaces into Euclidean space we can obtain the following result. For a finite metric space $X$ and a set $S$ let $\lambda(S) = \min_{x \in X, r > 0}\{|B(x, 3r)| \mid S \subseteq B(x, r)\}$. Applying Theorem 57 together with the scaling local embedding Theorem 3 of (2) we get:

**Corollary 11.** *Let $X$ be an $n$-point metric space. Then for every $k \geq 1$ and $q \geq 1$, $X$ admits $\left(k, (1 + \sqrt{\frac{q}{k}}) \cdot \tilde{O}(\log(\lambda(S)))\right)$-hyper-sketching.*[18] *The query time is $O\left(m_w k\right) = O\left(s^2 k\right)$, with $O(n^2)$ preprocessing time.*

Also in the special case of $q = 1$ and uniform weights we can compose the scaling local embedding of (2) with Theorem 85 to obtain the improved query time:

**Corollary 12.** *Let $X$ be an $n$-point metric space. Then for every $k \geq 1$ and $q \geq 1$, $X$ admits $\left(O(1), \tilde{O}(\log(\lambda(S)))\right)$-hyper-sketching.*[19] *The query time is $O\left(s\right)$, with $O(n^2)$ preprocessing time.*