[Reviews · NeurIPS 2019]

Reviewer 1



Originality As far as I can tell, the authors' claim that this is the first such work is correct. Previous work has been done is describing heuristics or empirical understandings of such behaviour, but the work is nonetheless original in proving a theoretical basis for this. Quality The authors' exposition of the problem and the solution is well thought out and expertly laid out in a logical and convincing form. However, the excellent technical contribution is somewhat lacking in discussion, particularly given the authors aim to bridge the gap between theory and practice ; such claims as "This new consequence may serve an important guide for practical considerations" warrant a standalone discussion section which is not provided. Further, the results predicted in theory could have been compared to empirical experiments to show tightness in practice, and phase transitions could be shown in experiments as a demonstration. Clarity The work is well written and the notations are easily understood. There are a moderate number of typographical errors, these are listed at the end of this review. Significance The authors correctly identify that metric dimensionality reduction is a crucial part of most modern machine learning pipelines. Better theoretical understanding of the average case performance is a highly significant contribution to a vast array of applications. Further, the authors' framework for collapsing existing distortion measures to just two generalized forms will facilitate further theoretical analysis of this ensemble of measures. Detailed comments : Intro Laundry list of use cases : maybe better separating citations by subfield so these are more useful to the reader Background Page 3, reference to Hriser (998a) should be (1998a) Energy+REM definition should be display math as it takes up a whole line anyway Section 2 "in what follows we will omit pi from the notation" - there is an extra space after the paranthesis. Really, this should be a separate sentence and not a paranthesis. Section 3 Footnote citation should be citep not citet Section 4 JL citation should be citep not citet Missing space after comma at Rademacher entries matrix -------------------------------- Update after author response -------------------------------- The authors' rebuttal was succinct and clear and provided ample empirical results supporting the theory presented in the original submission. These empirical results significantly strengthen the manuscript and I argue for acceptance as a result.

Reviewer 2



This paper reads well with clear math notations and has sufficient originality. The paper provides the first comprehensive analysis of metric dimensionality reduction under various distortion measurement criteria and fills the gap between the theoretical perspective and practical viewpoint on metric dimensionality reduction. The paper provides a clean line of related works and highlights the contributions of the paper in a reasonable and fair way. This paper as it states might provide useful practical guidance on metric dimensionality reduction.

Reviewer 3



The authors benchmark various average case distortion criteria. This task is valuable in itself, as dimensionality reduction plays a central role in many key applications of machine learning. However, the authors do not demonstrate enough evidence to support this contribution. For example, on pg 5. the authors state "is is easy to see all (adapted versions of) distortion criteria discussed obey all properties." The reader did not find it easy to see, and after re-reading the previous 4 pages did not find either the adaptation in question well specified or support for the claim that the criteria are fulfilled. If this support is in the appendix, the relevant sections should be well specified for the reader. This paper could benefit greatly from having a more clearly delineated structure. The flow of the paper is obtuse, due in part to having unintuitive names for sections and subsections. Referencing sections that do not exist (on page 3) combined with the lack of a conclusion, makes this work feel like one that is in progress rather than complete. It is also hard to decouple at times what is the authors contribution vs previous work. For example in section 2), the motivated properties are largely the same as that proposed by Chennuru Vankadara and von Luxburg (2018). It is unclear exactly what has been generalized. In addition, one of the properties defined by Chennuru Vankadara and von Luxburg (2018) "translation invariance" is not discussed here, although the authors imply they benchmark against all the properties listed. The authors do not benchmark their proposed new method (the square root of the variance distortion proposed by Chennuru Vankadara and von Luxburg (2018)) using a simulation framework that would help evidence the utility of their contribution. The lack of empirical simulation to support the theoretical discussion makes placing the value of this work within the literature difficult. Overall, the poor structure of the writing makes the overall contribution of the authors hard to discern. This may be a symptom of this being a work in progress, and not being able to appropriately clarify or support claims to the reader.

[Author Response · NeurIPS 2019]

We first address all referees' request to include experimental validation of the theoretical results in the paper.

**Empirical Experiments.** In all the experiments, we have followed VL [NIPS'18] using similar distributions for
sampling random Euclidean input spaces, tests were made for a large range of parameters, averaging over at least 10
independent tests. The results are consistent for all settings, and measures, and will be provided in full in the final paper.

*Tightness of the bounds, phase transition phenomenon, and superiority of JL.* In our paper we proved theoretical bounds
for the distortion measures of the JL transform into $k \geq 1$ dimensions. In particular, we showed that for $q < k$ the
$\ell_q$-distortion is bounded by $1 + O(1/\sqrt{k}) + O(q/k)$, and all the rest measures are bounded by $O(\sqrt{q/k})$. Particularly,
the bounds are independent of $n$ - the size and dimension $d$ of the input data set. In addition, we proved that for the
$\ell_q$-distortion and $\mathrm{REM}_q$ measures a phase transition must occur at $q \sim k$ for any dim. reduction method, where the
bounds dramatically increase from being bounded by some constant to grow with $n$, in particular as $\mathrm{poly}(n)$ for $q > k$.

The graphs in Fig. 1 and Fig.2a describe the following setting: A random $X$ of a fixed size and dimension $n = 800$ was
embedded into $k \in [4, 30]$ dimensions, by the JL/PCA/Isomap methods; the value of $q = 10$. We stress that we run
many more experiments a wide range of parameter values of $n \in [100, 3000]$, $k \in [2, 100]$, $q \in [1, 10]$, and obtained
essentially identical qualitative behavior. In Fig. 1a, the $\ell_q$-distortion as a function of $k$ of the JL embedding is shown
for $q = 8, 10, 12$. The phase transitions are seen at around $k \sim q$ as predicted by our theorems. In Fig. 1b the bounds
and the phase transitions of the PCA and Isomap methods are presented for the same setting ($d = 800, q = 10$), as
predicted by our lower bounds. In Fig. 1c, $\ell_q$-distortion bounds are shown for increasing values of $k > q$. Note that
the $\ell_q$-distortion of the JL is a *small constant close to* 1, as predicted, compared to values significantly $> 2$ for the
compared heuristics. Overall, Fig. 1 clearly shows that JL dramatically outperforms the other methods for all the
range of values of $k$. Below is **Fig. 1: Validating $\ell_q$-distortion behavior**. The same conclusions as above hold for

| (a) Phase transition of JL | (b) Phase transition of PCA/Isomap | (c) Comparing $\ell_q$-dists for $k > q$ |

20
$\sigma$-distortion as well, as shown in Fig. 2a, on the same sample data set. In the last experiment shown in Fig. 2b, we tested
the behavior of the $\sigma$-distortion as a function of $d$-the dimension of the input data set, similarly to that of VL[18](Fig.
2), and tests are shown for embedding dimension $k = 20$ and $q = 2$. According to our theorems, the $\sigma$-dist of the JL
transform is $O(\sqrt{q/k})$, which is bounded by constant for $q < k$. It is seen that the $\sigma$-dist is growing as $d$ increases for
both PCA/ISOMAP, whereas it is a *constant* for JL, as predicted. Moreover, JL obtains a *significantly smaller value* of
$\sigma$-distotion. Below is **Fig. 2: Validating $\sigma$-distortion behavior**. In the final paper, we will include further experiments

| (a) sigma-distortion | (b) sigma-distortion as a function of dimension $d$ |

26
on the JL-based approximation algorithms, which are expected to show similar to more dramatic qualitative behavior.

**Discussion/Conclusion Section.** Two of the referees #2 and #5 rightfully requested the inclusion of such a section,
discussing consequences of the work for practical considerations. We note that a shortened version implicitly appears in
the last 3 paragraphs prior to section 1.1 in the supp. material. The discussion section will greatly expand on these.

**Further improvement suggestions.** We thank referee #2 for his detailed comments that we'll happily incorporate.
We shall adopt referee #4's suggestion to use numeric citations (we didn't realize it was possible). Referee #5 asks to
improve clarity and writing, in contrast to the others who seem impressed by it. He mentioned "sections that do not exist
(on page 3)" - can be found in supp. material. We realize that NIPS has a wide range of audience and we will make an
effort to rewrite in a way that will be clear for all. The referee also criticizes the theoretical methodology of the paper,
yet the paper contains very detailed proofs for all theorems. The *only* exception mentioned by the referee: proofs of the
properties in page 5 will be included in the full paper. In particular, the "translation invariance" property mentioned by
the referee, trivially holds for any distance based measure in any metric space *by definition*. We note that we do not
"propose a new distortion measure" but new dim. reduction methods, based on JL, which we have addressed above.

[Meta-Review · NeurIPS 2019]

This is a very interesting paper, which presents a comprehensive theoretical analysis of metric dimensionality reduction. It describe existing distortion measures in terms of moments of distortions and give an average case performance guarantee for these moments of distortion. Also, an approximate algorithm with provable guarantees on metric dimensionality reduction is introduced. The main objection on this paper was the absence of empirical evidence to support the claims. The authors have conducted additional experiments in the rebuttal phase but there are missing details regarding the experiments. The authors are advised to improve the quality of their paper in light of the reviewers' comments and incorporate their recommended changes.